# A reversible state of hypometabolism in a human cellular model of sporadic Parkinson's disease

Sebastian Schmidt [1,2] ✉, Constantin Stautner[1], Duc Tung Vu [3], Alexander Heinz[4], Martin Regensburger [5], Ozge Karayel [3], Dietrich Trümbach [1,6], Anna Artati[7], Sabine Kaltenhäuser[4], Mohamed Zakaria Nassef[4], Sina Hembach[1], Letyfee Steinert[1], Beate Winner[8], Winkler Jürgen [5], Martin Jastroch [9], Malte D. Luecken [10], Fabian J. Theis [10,11], Gil Gregor Westmeyer[2,12], Jerzy Adamski [13,14,15], Matthias Mann [3,16], Karsten Hiller [4], Florian Giesert [1], Daniela M. Vogt Weisenhorn [1] & Wolfgang Wurst [1,17,18,19] ✉

Sporadic Parkinson's Disease (sPD) is a progressive neurodegenerative disorder caused by multiple genetic and environmental factors. Mitochondrial dysfunction is one contributing factor, but its role at different stages of disease progression is not fully understood. Here, we showed that neural precursor cells and dopaminergic neurons derived from induced pluripotent stem cells (hiPSCs) from sPD patients exhibited a hypometabolism. Further analysis based on transcriptomics, proteomics, and metabolomics identified the citric acid cycle, specifically the α-ketoglutarate dehydrogenase complex (OGDHC), as bottleneck in sPD metabolism. A follow-up study of the patients approximately 10 years after initial biopsy demonstrated a correlation between OGDHC activity in our cellular model and the disease progression. In addition, the alterations in cellular metabolism observed in our cellular model were restored by interfering with the enhanced SHH signal transduction in sPD. Thus, inhibiting overactive SHH signaling may have potential as neuroprotective therapy during early stages of sPD.

Parkinson's disease (PD) is a neurodegenerative disorder characterized in advanced stages by motor disabilities and the loss of dopaminergic neurons (DAns) specifically in the substantia nigra pars compacta[1]. The vast majority of PD cases are sporadic, induced by a combination of genetic and environmental risk factors. Only about 15% are associated with heritable familial mutations[2]. The degenerative processes in PD patients start and develop long before the characteristic motor symptoms occur, which are needed to finalize the diagnosis. This prodromal phase can last up to 20 years[3,4]. Understanding this early phase of PD at the molecular level is of the highest importance for the development of disease-modifying or neuroprotective therapies,

which require intervention at the earliest stages of disease in order to prevent progressive neurodegeneration.

To investigate disease-causing molecular and cellular mechanisms, model systems carrying PD-associated mutations are widely used which allowed identifying molecular and cellular dysfunctions associated with PD such as mitochondrial impairment, autophagy, protein aggregation, proteasomal degradation, and primary cilia dysfunction[4,5]. With the majority of PD-associated genes including *PARK7* (*DJ1*), *PINK1*, *PRKN*, *SNCA*, and *LRRK2* directly affecting mitochondria, mitochondrial dysfunction has emerged as a central factor contributing to PD etiology. Overall, these genes are involved in

mitochondrial homeostatic control as well as basic functions such as energy production and oxidative stress which may crosstalk with other PD-associated pathways. Also in postmortem studies analyzing brain tissue of sporadic Parkinson's disease (sPD) patients, mitochondrial alterations could be identified amongst others varying degrees of complex I and complex II deficiency (from ~30 to 60%)[6–8].

Still, the relevance of these alterations for the etiology of sporadic Parkinson's disease (sPD) remains largely elusive due to the lack of suitable human model systems[9,10]. The ascent of human induced pluripotent stem cells (hiPSCs) allowed the use of patient-specific stem cells as model systems to expand our knowledge of human physiology at the cellular level[11]. During the reprogramming of patient-derived fibroblasts into hiPSCs these cells are rejuvenated regarding their epigenetic state, transcriptome, telomeres, and mitochondrial function[12–15]. Thus, in contrast to analyses of postmortem material from sPD patients, hiPSCs and their derivatives are thought to recapitulate early disease events.

In order to establish a human cellular model of early sPD, we analyzed sPD patient-derived hiPSCs as well as their differentiation products—neural precursor cells (hNPCs) and DAns—for metabolic and mitochondrial alterations. We could show that sPD neural cells develop a state of hypometabolism also due to a bottleneck within the citric acid cycle at the level of the α-ketoglutarate dehydrogenase complex (OGDHC). Thereby, alterations in sPD metabolism were introduced by enhanced primary cilia (PC)-mediated sonic hedgehog (SHH) signal transduction, as alterations in cellular metabolism could be rescued in our cellular model of sPD by interfering with SHH signaling. Thus, dysfunctional PC signaling pathways, especially SHH signaling, induce major metabolic rearrangements associated with sPD. This suggests that metabolic dysfunction modifiable by SHH signaling via PC is a central factor contributing to the pathoetiology of sPD implying a novel therapeutic option.

## Results

In this study, we used hiPSCs derived from fibroblasts from 7 sPD patients and 5 age- and sex-matched Ctrl individuals, which were cultivated in vitro for approximately 60 passages. sPD patients were clinically examined and screened for the absence of known PD-causing familial mutations (PARK1-18)[16]. hiPSCs were repeatedly characterized for copy number variations and their differentiation potential[17]. To establish a model system for sPD, hiPSCs were differentiated into human neural precursor cells (hNPCs) and further into dopaminergic neurons (DAns), which are vulnerable to degeneration in PD[1]. Differentiation stages were confirmed using immunohistochemical staining for characteristic markers such as the precursor markers NESTIN, SOX2, and SOX1 (Supplementary Fig. 1a) or the DAn marker TUBB3, RBFOX3 (synonym: NeuN), and TH[17]. Expression of these differentiation markers was not affected in sPD hNPCs (Supplementary Fig. 1b–d), nor was the abundance and morphology of DAns derived thereof[17]. This indicates that the DAn differentiation process assessed at various stages was comparable between Ctrl and sPD cells.

Previous studies using these hiPSC-derived neuronal cells indicated that mitochondrial functionality regarding the cellular respiration as well as complex I deficiency are impaired in sPD[17]. The present study aims to decipher the molecular mechanisms underlying these PD-associated changes.

### Mitochondrial dysfunction in neural cells derived from sPD patients

Mitochondrial function was assessed in hiPSCs, as well as their derivatives—hNPCs and DAns—using Seahorse XF analysis. It allows the assessment of multiple parameters, like basal respiration, proton leak, ATP-linked respiration, and maximal respiratory capacity. Furthermore, different energy substrates (glucose or pyruvate) allow to investigate the contribution of specific metabolic pathways to cellular respiration and can be used to determine possible bottlenecks.

Using glucose as an energy substrate, we observed no significant differences in mitochondrial respiration in sPD hiPSC and DAns. However, the basal, ATP-linked, and maximal respiratory capacity of sPD hNPCs was decreased (trend) (Fig. 1). This might indicate an sPD-specific defect in cellular respiration possibly in glucose uptake, glycolysis, the citric acid cycle, or oxidative phosphorylation.

To discriminate limitations within glycolysis from mitochondrial ones, pyruvate was supplied as a substrate that can directly enter the citric acid cycle. Using pyruvate as an energy substrate, the basal, ATP-linked, and maximal mitochondrial respiration of sPD hNPCs and DAns was significantly reduced. In contrast, the original hiPSCs were still not affected (Fig. 1).

Since cellular respiration was even more impaired when pyruvate was used as an energy substrate, this indicated a defect downstream of glycolysis in sPD hNPCs and DAns. Thus, we expected the defect either in the substrate delivery for the respiratory chain by the citric acid cycle or the respiratory chain itself.

To get a more comprehensive overview of cellular metabolism, the glycolytic flux based on the extracellular acidification rate (ECAR) was also assessed. The glycolytic flux was only analyzed in cells supplied with glucose as an energy substrate. The oxidation of glucose during glycolysis depends on ATP hydrolysis and results in the production of protons, pyruvate, and often lactate. ECAR mainly correlates with lactate/$H^+$ secretion and can be masked by various other cellular processes leading to acidification. Thus, it only offers a rough overview of glycolytic rates and has to be interpreted with caution[18].

The glycolytic flux analysis did not show significant differences in the ECAR between sPD and Ctrl hiPSCs, hNPCs, and DAns (Supplementary Fig. 2a–c), supporting our hypothesis of a defect downstream of glycolysis as indicated by the OCR measurements using pyruvate as a substrate. It might, however, also indicate that reduced mitochondrial ATP production in sPD is not compensated by an increased glycolytic flux and conversion of pyruvate to lactate.

To further elucidate the impact of glycolytic flux and mitochondrial respiration to energy production, we calculated the ECAR to OCR ratio and visualized the total levels by plotting the ECAR against the OCR (Supplementary Fig. 2d–f). The higher the ratio, the lower the proportion of mitochondrial respiration for energy production should be. Indicative for the well-known glycolytic switch during neuronal differentiation, the ECAR to OCR ratio is declining during this process with the highest values in hiPSCs and the lowest ones in DAns. Furthermore, the ratio is significantly increased in sPD hNPCs and tends to be increased in hiPSC and DAns. This indicates a decreased proportion of mitochondrial respiration to total energy production in sPD.

### Mitochondrial health is not affected in neural cells derived from sPD patients

Aiming to identify the underlying causes of reduced mitochondrial respiration, we first investigated mitochondrial mass and morphology. Alterations in both have been previously described in various PD models[19–22].

The amount of total mitochondrial numbers and morphology was determined using two different independent stainings. On one hand, we stained the ATP synthase with an antibody against ATP5F1A in hNPCs (Fig. 2a) and DAns derived thereof (Fig. 2b). To specifically visualize functional/active mitochondria, we used MitoTracker which accumulates specifically in mitochondria with intact membrane potential (Fig. 2a, b). In total, five characteristics of mitochondrial morphology were assessed: The number of mitochondria per cell, the mean area and fluorescence intensity of the mitochondria per cell, as well as the mean length of their morphological skeleton, and its number of branch points per cell. However, neither in hNPCs (Fig. 2c) nor in DAns (Fig. 2d) derived thereof any significant differences

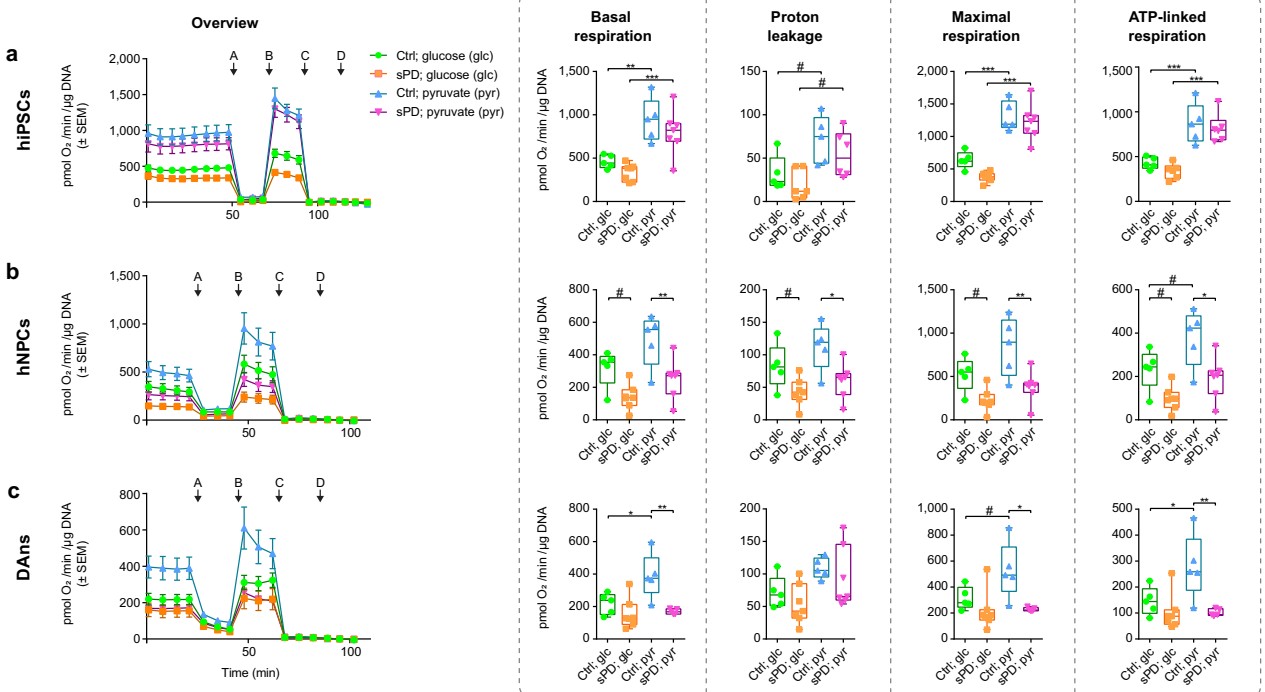

**Fig. 1 | Alterations in cellular respiration in sPD patient-derived cells.**
**a** Mitochondrial stress test performed in hiPSCs, **b** thereof differentiated hNPCs, and **c** DAns using a Seahorse XFe96 Extracellular Flux Analyzer. Cells were measured in Seahorse XF assay medium supplemented with 25 mM glucose or 5 mM pyruvate (as shown in ref. 17). Injected were (A) Oligomycin (1 μg/ml), (B) carbonyl cyanide p-trifluoro-methoxyphenyl hydrazone (FCCP; 0.5 μM), (C) Rotenone (5 μM)/Antimycin A (2 μM), and (D) 2-deoxyglucose (2-DG; 100 mM). Measurement progression is shown with means ± standard error of the mean (SEM). Boxplots display the median and range from the 25th to 75th percentile. Whiskers extend from the min to max value. Each dot represents one patient. $n = 5$ Ctrl and 7 sPD patient-derived cell clones, in triplicates. $p$-values were determined by one-way ANOVA with Sidak's post hoc test. $^{\#}p < 0.1$; $^{*}p < 0.05$; $^{**}p < 0.01$; $^{***}p < 0.001$. Source data including $p$-values are provided as a Source Data file.

between Ctrl and sPD clones could be observed, indicating that mitochondrial content or general health is not impaired in neural cells derived from sPD hiPSCs.

In addition, we characterized the abundance and post-translational modifications of the mitochondrial fusion and fission machinery which is essential for mitochondrial quality control and functioning. These processes allow cells to adapt the mitochondrial morphology to cellular metabolic demands and substrate supply[23] and is a measure for mitochondrial stress. Fusion of the outer and inner mitochondrial membrane is mainly facilitated by the mitofusions (MFN1 and MFN2), as well as OPA1, respectively, and mitochondrial fission is mainly mediated by DRP1[23]. As fusion is regulated by the expression of the corresponding components, expression levels of *MFN1*, *MFN2*, and *OPA1* were assessed on mRNA level using RT-qPCR. Contrary, fission is mainly regulated by post-translational modifications which were quantified on protein level using western blots. For example, phosphorylation of Serine at position 616 is thought to activate fission activity[23,24]. As mitochondrial function highly relies on substrate availability, cells were supplied with glucose or pyruvate as during the mitochondrial respiration analysis to unmask possible deficits in adjusting mitochondrial morphology to cellular demands in sPD clones. However, no significant differences in the expression of *MFN1*, *MFN2*, or *OPA1* (Fig. 2e) as well as in the abundance of DRP1, or DRP1-pSerine[616] (Fig. 2f, g) could be identified in sPD hNPCs neither with glucose nor with pyruvate as energy substrate. Upon changing the substrate supply from glucose to pyruvate, a significant upregulation in the expression of *MFN2* and thus the fusion machinery could be observed for both Ctrl and sPD clones, possibly as a reaction to the loss of glycolytic energy production. These results further validate our previous results and exclude a decrease in mitochondrial mass or alterations in mitochondrial dynamics as an explanation for the respiratory deficiency that was observed in the Seahorse analysis.

**Transcriptome analysis reveals differential expression of genes associated with the electron transport chain in sPD**
To get insights into the molecular underpinnings for the observed respiratory deficiency in sPD hNPCs—which showed the most pronounced phenotype—we first performed a pathway enrichment analysis using our recently published single-cell transcriptome data (bulk-like) of these cells (Supplementary Data 2)[17]. On the pathway level, mainly processes associated with the respiratory chain were dysregulated in sPD (Fig. 3a). Remarkably, the expression of almost every gene encoding a subunit of the electron transport chain was affected in sPD, and in almost every case the expression on mRNA level was downregulated (Fig. 3b). This was also true for all 13 mitochondrial-encoded subunits, which were downregulated to an even higher extent than the remaining nuclear-encoded subunits. However, mitochondrial-encoded tRNAs were not differentially expressed in sPD indicating that there exists not a general problem with the mitochondrial genome and transcriptional processes (Supplementary Data 5 of ref. 17).

Consequently, the abundance of the mitochondrial complexes I - V was determined on the protein level using western blots for labile subunits of each complex. Contrary to our expectations, sPD and Ctrl hNPCs (Fig. 3c, d) and DAns (Fig. 3e) expressed the labile subunits in similar amounts. Thus, it can be concluded that the total abundance of the respiratory chain complexes was not altered in sPD on the protein level.

Although the total abundance of mitochondrial complexes on the protein level was not affected in sPD, their activity may be altered due to differences in complex assembly and structure[25]. Indeed, a reduced complex I activity by ~30% has been described for these sPD hNPCs[17].

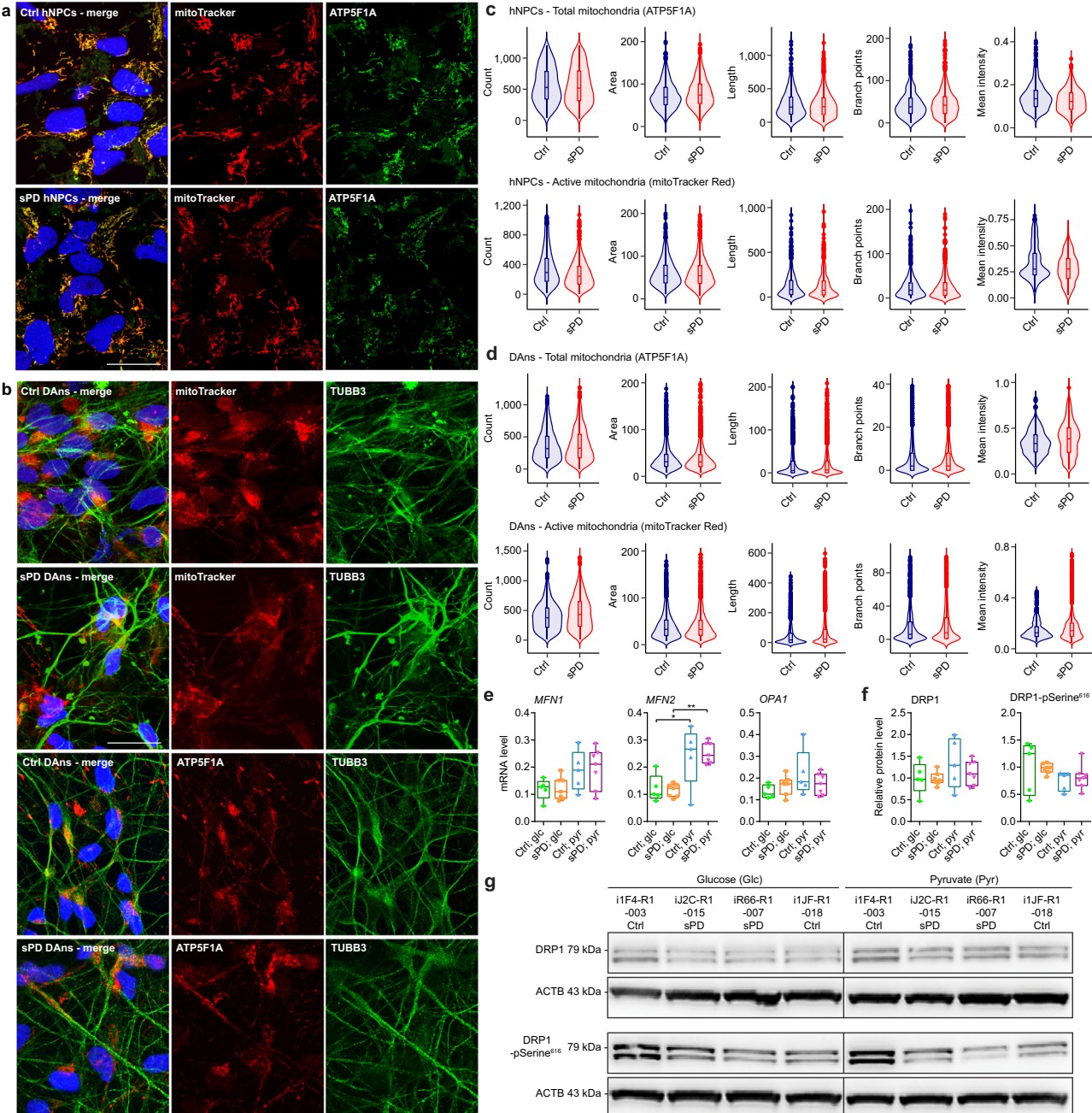

**Fig. 2 | Mitochondrial abundance and morphologies are not altered in patient-derived cells. a** Quantification of total mitochondrial mass and functional mitochondria in hNPCs or **b** DAns. Total mitochondrial mass was visualized using immunostainings with an antibody against the ATP synthase (ATP5F1A). Functional mitochondria with an active membrane potential were visualized using a Mito-Tracker probe (200 nM for 20 min). Images are exemplarily shown for hNPCs of clone i1E4-R1-003 (Ctrl), and iR66-R1-007 (sPD); for DAns of clone i1E4-R1-003 (mitoTracker−Ctrl), iR66-R1-007 (mitoTracker−sPD), i1JF-R1-018 (ATP5F1A−Ctrl), and iJ2C-R1-015 (ATP5F1A−sPD). Scale bar = 20 μm. **c** Violin plots highlight some morphological characteristics of mitochondria in hNPCs or **d** DAns: number of mitochondria (count), mitochondrial area, skeleton length, number of branch points, and mean intensity. On average, 112 (hNPCs−ATP5F1A), 160 (hNPCs−mitoTracker), 370 (DAns−ATP5F1A), 360 (DAns−mitoTracker) cells per clone were analyzed. **e** To assess alterations in the mitochondrial fusion machinery, expression of the mitofusions *MFN1*, *MFN2*, and *OPA1* was quantified in hNPCs by RT-qPCR. Cells were cultivated on the energy substrates used in the Seahorse XF analysis (25 mM glucose or 5 mM pyruvate). **f** To assess alterations in the mitochondrial fission machinery, expression of DRP1 and its phosphorylation on Ser[616] was quantified in hNPCs by western blot. Protein levels were normalized to ACTB. **g** Western blots are exemplary shown for some hNPC clones. Boxplots display the median and range from the 25th to 75th percentile. Whiskers extend from the min to max value or to the most extreme data point which is no more than 1.5 times the interquartile range (**c**, **d**). Each dot represents one patient. *n* = 5 Ctrl and 7 sPD patient-derived cell clones, in triplicates. *p*-values were determined by linear mixed effects model (**c**, **d**); one-way ANOVA with Sidak's Post hoc test (*p*-values are provided together with the source data) (**e**, **f**). *$p < 0.05$; **$p < 0.01$; ***$p < 0.001$. Source data are provided as a Source Data file.

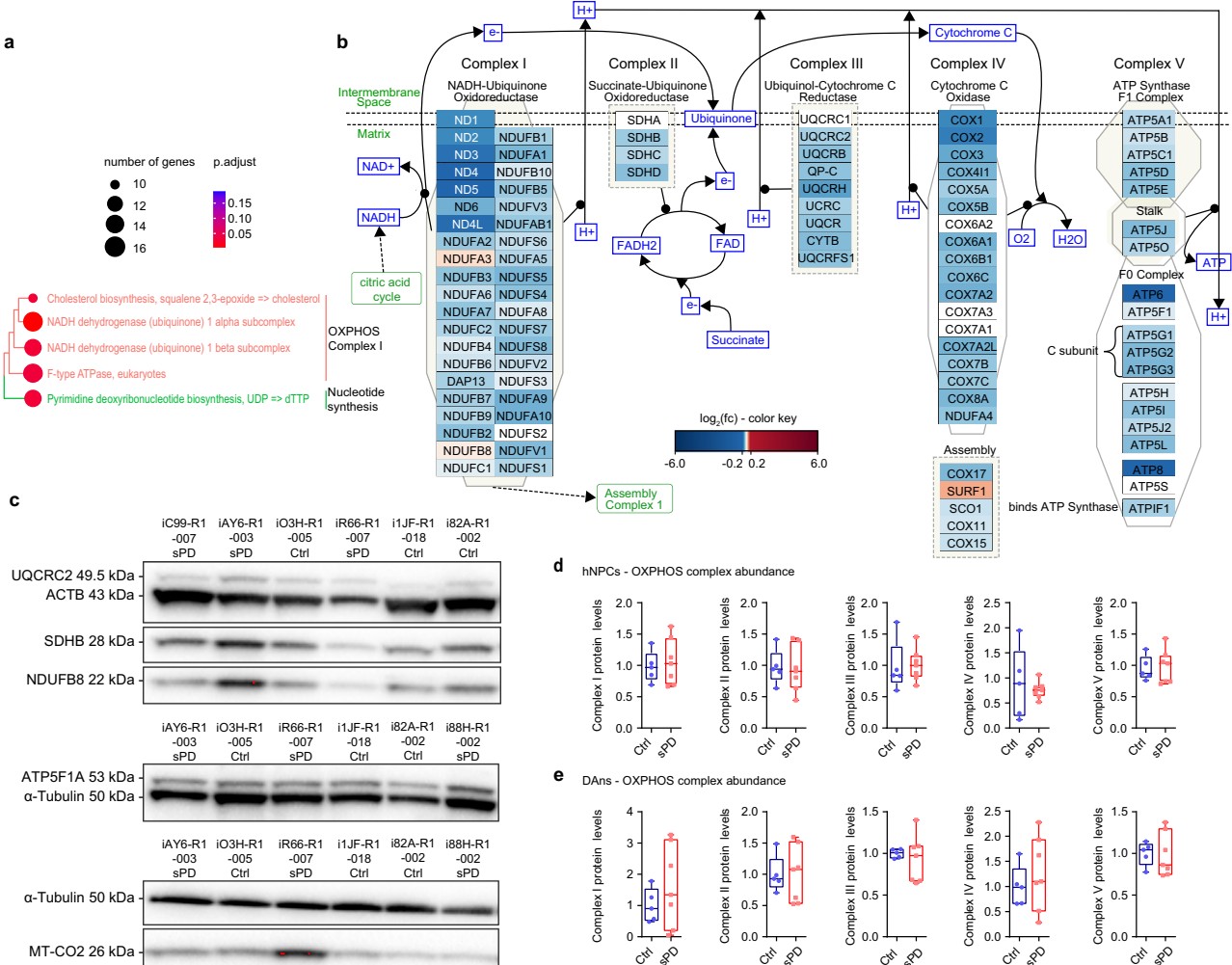

**Fig. 3 | Abundance of the electron transport chain complexes is not affected in sPD. a** Metabolic KEGG pathways enriched in sPD hNPCs. Single-cell transcriptome data (bulk-like) were previously published[17]. FDR-corrected *p*-values are represented by *q*-values. **b** Visualization of the 'Electron Transport Chain (OXPHOS system in mitochondria)' pathway based on the single-cell transcriptome data with manual annotations. Color intensities for up- (red) and downregulated (blue) genes are proportional to the fold change. Not significantly altered genes are colored in white. **c** The abundance of mitochondrial complexes I – V was quantified by western blot with antibodies against the labile subunits NDUFB8 (complex I), SDHB (complex II), UQCRC2 (Complex III), MT-CO2 (Complex IV), and ATP5F1A (Complex V).

Expression levels were normalized to ACTB or α-Tubulin levels. Western blots are exemplarily shown for some hNPC clones. Quantifications of protein levels are shown for **d** hNPCs and **e** DAns. *n* = 5 Ctrl and 7 sPD patient-derived cell clones, in triplicates. Boxplots display the median and range from the 25th to 75th percentile. Whiskers extend from the min to max value. Each dot represents one patient. *p*-values were determined by one-sided hypergeometric tests (**a**), two-sided *t*-test (**d**) (complex I: *p* = 0.69; complex II: *p* = 0.86; complex III: *p* = 0.89; complex IV: *p* = 0.66; complex V: *p* = 0.83), **e** (complex I: *p* = 0.35; complex II: *p* = 0.99; complex III: *p* = 0.62; complex IV: *p* = 0.50; complex V: *p* = 0.90). **p* < 0.05; ***p* < 0.01; ****p* < 0.001. Source data are provided as a Source Data file.

This corresponds well with literature describing complex I deficiency in different PD models as well as in postmortem brain tissues of PD patients[26,27].

Since hNPCs behaved like DAns and recapitulated the main findings of sPD-specific alterations in cellular respiration, these hNPCs are perfect for further investigations to unravel the underlying pathological mechanisms and represent—as published earlier[17]—a suitable model for sPD.

**Proteome analysis reveals dysregulation of pathways associated with mitochondrial function in sPD**
To further characterize dysfunctional cellular processes, we performed a proteome analysis in these hNPCs (Supplementary Data 3). Principal Component Analysis (PCA), as well as correlation analysis and hierarchical clustering, were used to visualize the variability within the samples to detect possible outliers (Fig. 4a and Supplementary Fig. 3a). Two human-derived cell lines (sPD−iR66-R1-007 and Ctrl−i1JF-R1-018) seemed to cluster separately and thus were removed as outliers for

downstream analysis. In total 7943 proteins were quantified (Fig. 4b and Supplementary Data 4) and out of these 1667 were significantly dysregulated (differentially expressed proteins−DEPs) in sPD based on a negative binomial generalized linear model and Wald tests (*q*-value < 0.05). A heatmap visualizing expression levels of all DEPs shows the similarity between patients and highlights differences in sPD (Fig. 4c). Similar to the transcriptome[17], levels of most DEPs were only slightly altered. However, up- and downregulation of DEPs was more balanced (Fig. 4b) compared to the transcriptome level where ~88% of differentially expressed genes (DEGs) were downregulated.

To determine disease-associated cellular processes, we performed a pathway enrichment analysis based on all DEPs using multiple pathway databases (KEGG, Reactome, and WikiPathways)[28]. Significantly enriched pathways (*q*-value < 0.05) could be grouped into the categories metabolism (citric acid cycle), primary cilia (PC), the CCT/TriC complex, RHO/RAC GTPase cycle, growth factor signaling, and cell cycle (Fig. 4d, Supplementary Fig. 3b, c, and Supplementary Data 5).

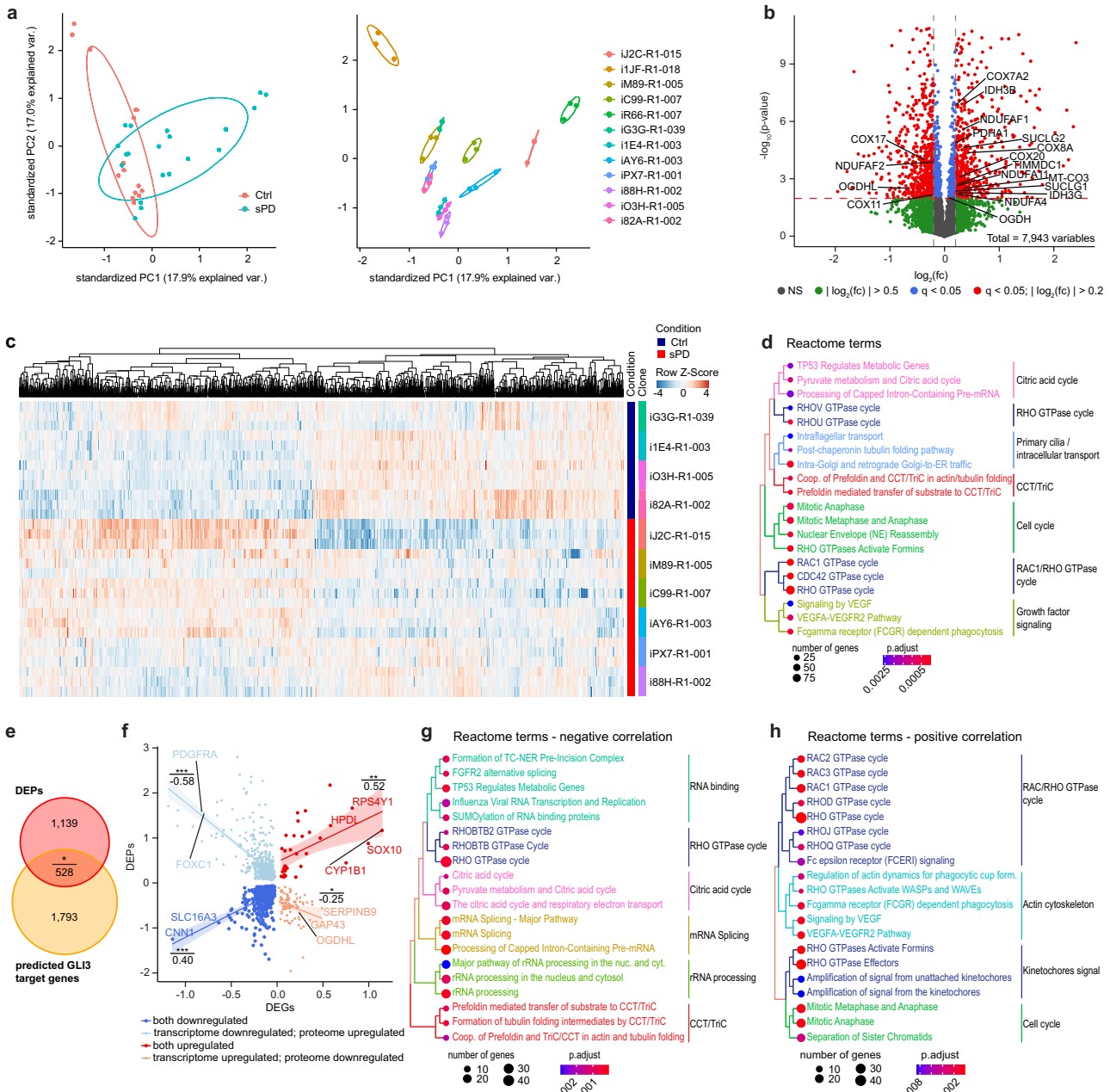

**Fig. 4 | Proteome analysis points towards primary cilia and citric acid cycle defects in sPD. a** Principal component analysis visualizes the variability within technical replicates and conditions. **b** Proteome analysis identified 1667 (out of 7943 proteins) which were dysregulated in sPD hNPCs. Dotted line indicates the significance threshold ($q < 0.05$). **c** Heatmap showing log2-transformed fold changes (FC) with columns scaled by z-score for differentially expressed proteins (DEPs). **d** Enriched Reactome terms based on all DEPs. **e** Overlap of genome-wide predicted GLI3 target genes[17] with DEPs. **f** Correlation between DEGs and DEPs. FC for DEGs is plotted on the x-axis, FC of DEPs on the y-axis. Linear regressions between upregulated DEGs and DEPs (red), downregulated DEGs and DEPs (blue), upregulated DEGs and downregulated DEPs (orange), downregulated DEGs and upregulated

DEPs (light blue) are displayed together with their respective Pearson correlation coefficients and p-values. **g** Enriched Reactome terms for DEG-DEP pairs with a negative correlation (downregulated DEG and upregulated DEP; upregulated DEG and downregulated DEP) between transcriptome and proteome level as well as a **h** positive correlation (downregulated DEG and DEP; upregulated DEG and DEP). $n = 5$ Ctrl and 7 sPD patient-derived cell clones, in triplicates. p-values were determined by one-sided hypergeometric tests (**d, g, h**); one-sided Fisher's Exact test (**e**) ($p = 0.014$); two-sided t-test (**f**). p-values corrected for multiplicity are represented by q-values. *$p < 0.05$; **$p < 0.01$; ***$p < 0.001$. See also Supplementary Fig. 3 and Supplementary Data 4, 5, and 6.

Two of these categories thereby directly translate from the transcriptome to the proteome namely cell cycle and cellular processes related to PC function. Alterations within cell cycle-related processes are not surprising due to the cycling nature of neural precursor cells. Still, alterations within these pathways did not affect cellular proliferation[17]. In addition, the intraflagellar transport (IFT) pathway associated with PC has been reported on the transcriptome level

before[17]. PC are hair-like organelles that extrude from the cell surface and function as cellular antennas that are thought to mediate the transduction of a myriad of external signaling events including SHH signaling[29,30]. The consequences of a dysfunctional IFT system or its interaction with the sorting system, especially the BBSome, are alterations in signal transduction[17,31]. This has been validated for PC-mediated alterations in SHH signaling, which was enhanced in sPD

hNPCs[17]. For SHH signaling, three transcription factors are known to mediate its signal transduction namely GLI1, GLI2, and GLI3[32]. Especially the latter one is of particular interest, as it is thought that SHH signaling during brain development is mainly to restrict the repressing effect of GLI3[33,34]. Thus, a significant enrichment of GLI3 target genes within DEGs (-22.5%)[17] and DEPs (-31.7%; Fisher's Exact test $p = 0.014$) (Fig. 4e), as well as an enrichment of misexpressed IFT components also on protein level (Fig. 4d) further strengthens the relevance of altered PC function in sPD etiology.

Interestingly, the massive downregulation of subunits of the electron transport chain on the transcriptome level was not observed on the proteome level. Here mainly components of complex IV seemed to be affected, rather than of complex I. Out of 18 complex IV subunits, levels of MT-CO3 (COX3), COX7A2, COX8A, and NDUFA4 were altered next to the assembly factors COX20, COX17, and COX11[35]. Although complex I is the largest complex of the electron transport chain consisting of 14 central subunits that are involved in energy conservation and roughly 30 accessory subunits[36], not a single complex I central subunit was misexpressed on the protein level. Instead, the assembly factors[36] NDUFAF1, NDUFAF2, and TIMMDC1 as well as one accessory subunit namely NDUFA11 were dysregulated. This is consistent with the unchanged total complex I abundance in sPD hNPCs analyzed by quantifying the labile subunit NDUFB8 by western blot (Fig. 3d, e). Thus, alterations in the assembly of complex I rather than dysfunctional central subunits might contribute to its observed reduced activity in sPD. Also, the abundance of the other labile subunits SDHB (complex II), UQCRC2 (Complex III), MT-CO2 (Complex IV), and ATP5F1A (Complex V) was not altered in the proteome analysis confirming the western blot results (Fig. 3d, e) as well as the quantification of total mitochondrial abundance (Fig. 2).

A strong pattern of dysregulation on the proteome level was evident for the citric acid cycle, a pathway significantly enriched when using KEGG, Reactome, and WikiPathway terms (Fig. 4d and Supplementary Fig. 3b, c). The citric acid cycle is the central common pathway for the oxidation of fuel molecules (carbohydrates, amino acids e.g. glutamine, and fatty acids) and includes a series of redox reactions resulting in the oxidation of substrates into $CO_2$, yielding ATP, NADH, or $FADH_2$[37,38]. If dysfunctional, it can create bottlenecks in the NADH/$FADH_2$ production rates that fuel the electron transport chain or in the production of intermediates for fatty acid and amino acid anabolism.

Surprisingly, expression of the citric acid cycle enzymes seemed to be mainly downregulated on the transcriptome level (bulk-like DEGs) but upregulated on the proteome level. In total, 1250 or 75.0% of the DEPs were also detected as DEGs in the transcriptome. Out of these, 586 (46.9%) DEP-DEG pairs showed a positive (downregulated DEG and downregulated DEP or upregulated DEG and upregulated DEP) and 664 (53.1%) a negative correlation (downregulated DEG and upregulated DEP or upregulated DEG and downregulated DEP) (Fig. 4f). Pathway enrichment analysis based on DEP-DEG pairs with a negative (Fig. 4g) or positive (Fig. 4h, Supplementary Fig. 3d, and Supplementary Data 6) correlation, respectively, confirmed the previous observation of the citric acid cycle enzymes following a negative correlation between transcriptome and proteome.

Taken together, the proteome analysis further validated PC and especially IFT dysfunction in sPD. It identified sPD-specific alterations in basal metabolism regarding the citric acid cycle. In order to determine the consequences of these changes in pathways affecting basal metabolism a metabolome analysis is warranted.

## Non-targeted metabolome analysis highlights the citric acid cycle as a bottleneck in sPD metabolism

To gain a better understanding of metabolic alterations under basal conditions also referring to possible alterations within the citric acid cycle, we performed a non-targeted metabolomics analysis in hNPCs. PCA was used to visualize the variability within the samples to detect possible outliers (Fig. 5a). The observed variation between technical replicates seemed to be lower than the variation between samples from different patients. No cell line clustered separately and thus no sample was removed as an outlier. In total 223 metabolites passed quality control and 45 metabolites were significantly dysregulated in sPD based on Student's $t$-tests. Interestingly, most of the metabolites detected and all significantly dysregulated metabolites were downregulated in sPD indicating a state of hypometabolism in sPD hNPCs (Fig. 5b and Supplementary Data 7).

An integrated analysis of metabolome data together with the transcriptome (bulk-like DEGs) (Fig. 5c and Supplementary Data 8) or proteome data (Fig. 5d and Supplementary Data 8) allowed refining pathway enrichment analysis of metabolic pathways affected in sPD. This analysis pointed toward significantly altered processes associated with "pyruvate metabolism" and the "one carbon cycle" (Fig. 5e), as well as towards the "citric acid cycle". Regarding the latter, the abundance of three consecutive intermediate metabolites, namely succinate, fumarate, and malate (Fig. 5f), as well as the expression levels of most citric acid cycle associated genes were reduced in sPD (Fig. 5e). Also on the protein level, the levels of many citric acid cycle associated enzymes were altered in sPD (Fig. 5e).

As levels of the metabolites aconitate (cis and trans), and α-ketoglutarate were not affected (Fig. 5e), we hypothesized that the conversion of α-ketoglutarate to succinate may be a bottleneck in sPD metabolism causing a reduced citric acid cycle flux and thus results in a reduced abundance of citric acid cycle intermediates. Changes in the citric acid cycle may directly affect the flux through the electron transport chain and thus mitochondrial respiration. In line with this, the abundance of the electron transport chain substrates NADH and succinate, produced within the citric acid cycle, were reduced in sPD (Fig. 5e, f). Interestingly, NAD$^+$ levels were not affected in sPD thus being not the limiting factor within the NAD$^+$/NADH balance (Fig. 5f). Both NADH and NAD$^+$ levels could also be validated independently (Fig. 5g). These reduced levels of electron transport chain substrates are further supported by a decreased mitochondrial ATP production rate (calculated from the basal mitochondrial respiration) in sPD hNPCs (Figs. 5h and 1b (pyr)), whereas the total ATP levels (Fig. 5i) were not affected.

Our assumption of an α-ketoglutarate dehydrogenase complex (OGDHC) deficiency in sPD clones was supported by the misregulation of its rate-limiting subunits OGDH and the brain-specific isoform OGDHL (Fig. 5e, j) on both transcriptome and proteome level.

Taken together, the non-targeted metabolomics data highlight a state of hypometabolism in neural cells derived from sPD patients. Furthermore, they strengthen the assumption of metabolic defects in the citric acid cycle in sPD, possibly at the level of the α-ketoglutarate dehydrogenase complex.

## Reduced glucose uptake in neural cells derived from sPD patients is not caused by alterations in glucose transporters

Based on the reduced mitochondrial respiration and ATP production, as well as on the massive reduction of metabolites we hypothesized that an sPD-specific state of hypometabolism develops in neural cells derived from sPD hiPSCs. To further investigate this state of hypometabolism, uptake and secretion rates of the main carbon sources glucose, lactate, glutamine, and glutamate within a 24 h period were analyzed in hNPCs (Fig. 6a). The glucose and glutamine uptake rates were significantly reduced in sPD hNPCs indicating a reduced total uptake of carbon sources for biomass and energy production. These measurements nicely correlated with clinical observations from sPD patients[39–41]. The reduced glucose uptake rate was further validated by using different methods again in hNPCs (Fig. 6b) but also in DAns derived thereof (Fig. 6c). This further strengthens our hypothesis of hypometabolism observed in neuronal cells (hNPCs and DAns) derived from sPD hiPSCs.

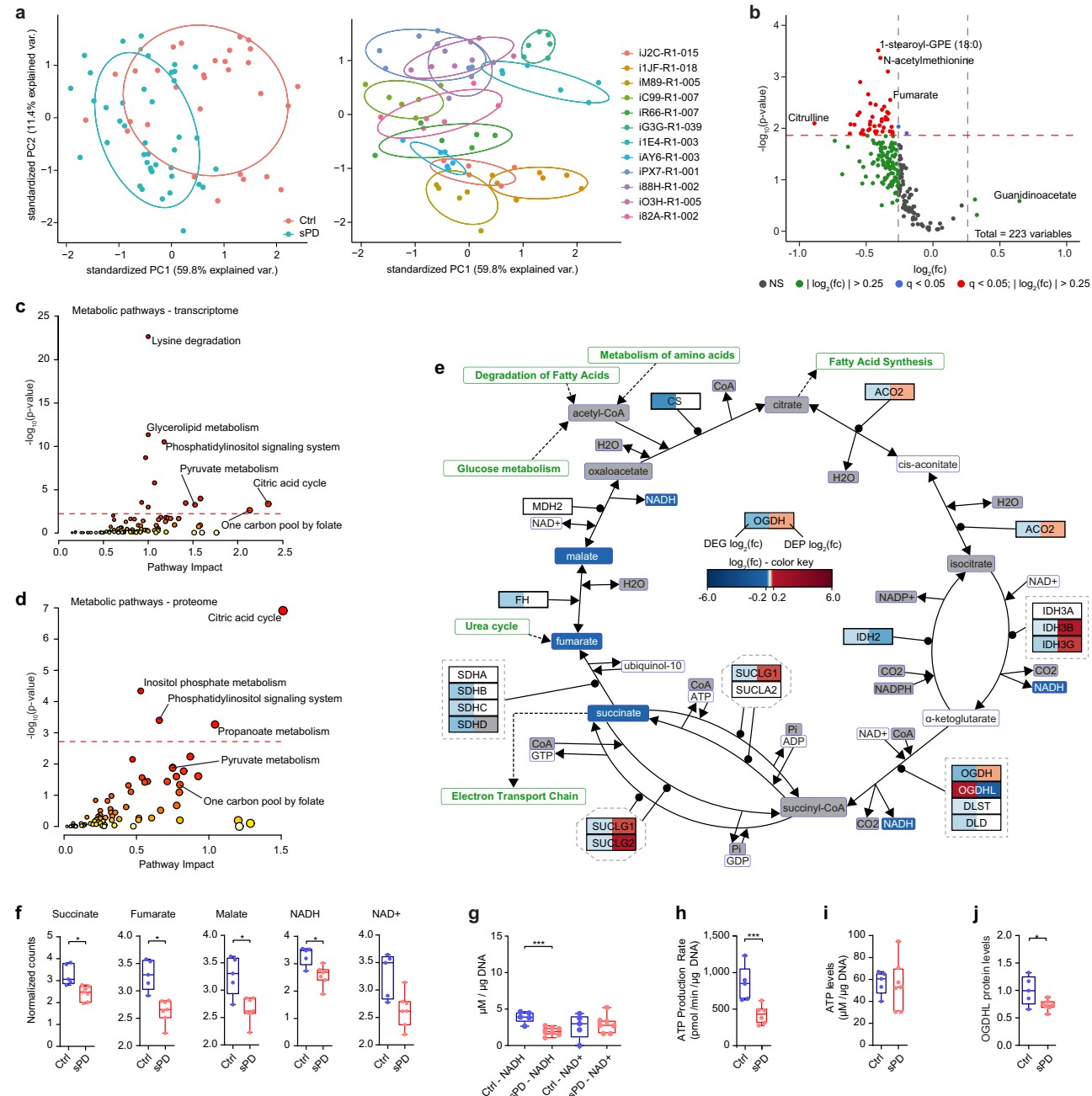

**Fig. 5 | Metabolic alterations propose a citric acid cycle bottleneck in sPD.**
**a** Principal component analysis visualizes the variability within technical replicates and conditions. **b** Non-targeted metabolomic analysis identified 45 metabolites (out of 223) that were significantly dysregulated in sPD hNPCs. Dotted line indicates the significance threshold ($q < 0.05$). **c** Integrative analysis of transcriptome and metabolome data **d** or proteome and metabolome data identified metabolic pathways enriched in sPD hNPCs. Node color is based on $p$-values and node radius on the pathway impact values. Dotted line indicates the significance threshold ($q < 0.05$). **e** Visualization of the "citric acid cycle" pathway with manual annotations. Color intensities for up- (red) and downregulated (blue) differentially expressed genes (DEG)/proteins (DEP) or metabolites are proportional to the fold change. Unchanged genes/proteins/metabolites are colored in white, not quantified metabolites or proteins are colored in gray. **f** Based on the non-targeted metabolomic analysis, levels of the citric acid cycle metabolites succinate,

fumarate, malate, and NADH were reduced in sPD. NAD+ levels were not affected. **g** Quantification of NADH and NAD+ levels in sPD hNPCs. **h** The ATP production rate was calculated from the basal mitochondrial respiration linked to ATP production measured in Seahorse XF analysis from hNPCs (Fig. 1b). **i** Cellular ATP levels (in μM) were analyzed in hNPCs and normalized to the genomic DNA content. **j** The abundance of OGDHL was quantified in hNPCs by western blot. Expression levels were normalized to GAPDH. Boxplots display the median and range from the 25th to 75th percentile. Whiskers extend from the min to max value. Each dot represents one patient. $n = 5$ Ctrl and seven sPD patient-derived cell clones, in triplicates or six replicates for the non-targeted metabolome analysis. $p$-values were determined by one-sided hypergeometric tests (**c**, **d**); two-sided $t$-test (**f**) (see Supplementary Data 7), **g** (NADH: $p = 0.0006$; NAD+: $p = 0.72$), **h** ($p = 0.005$), **i** ($p = 0.88$), **j** ($p = 0.030$). *$p < 0.05$; **$p < 0.01$; ***$p < 0.001$. See also Supplementary Fig. 7 and Supplementary Data 7 and 8. Source data are provided as a Source Data file.

To identify the mechanism underlying the hypometabolism observed in sPD, we first analyzed the expression and function of glucose transporters[42,43]. Expression levels of the most abundant glucose transporters *SLC2A3* and *SLC2A1* on mRNA level were not altered

in sPD hNPCs (Fig. 6d). Expression of *SLC2A4* was significantly decreased in sPD, however, as *SLC2A4* and *SLC2A2* were only very weakly expressed, their relevance in neural cells is questionable. Also on protein level, the expression of the most abundant glucose

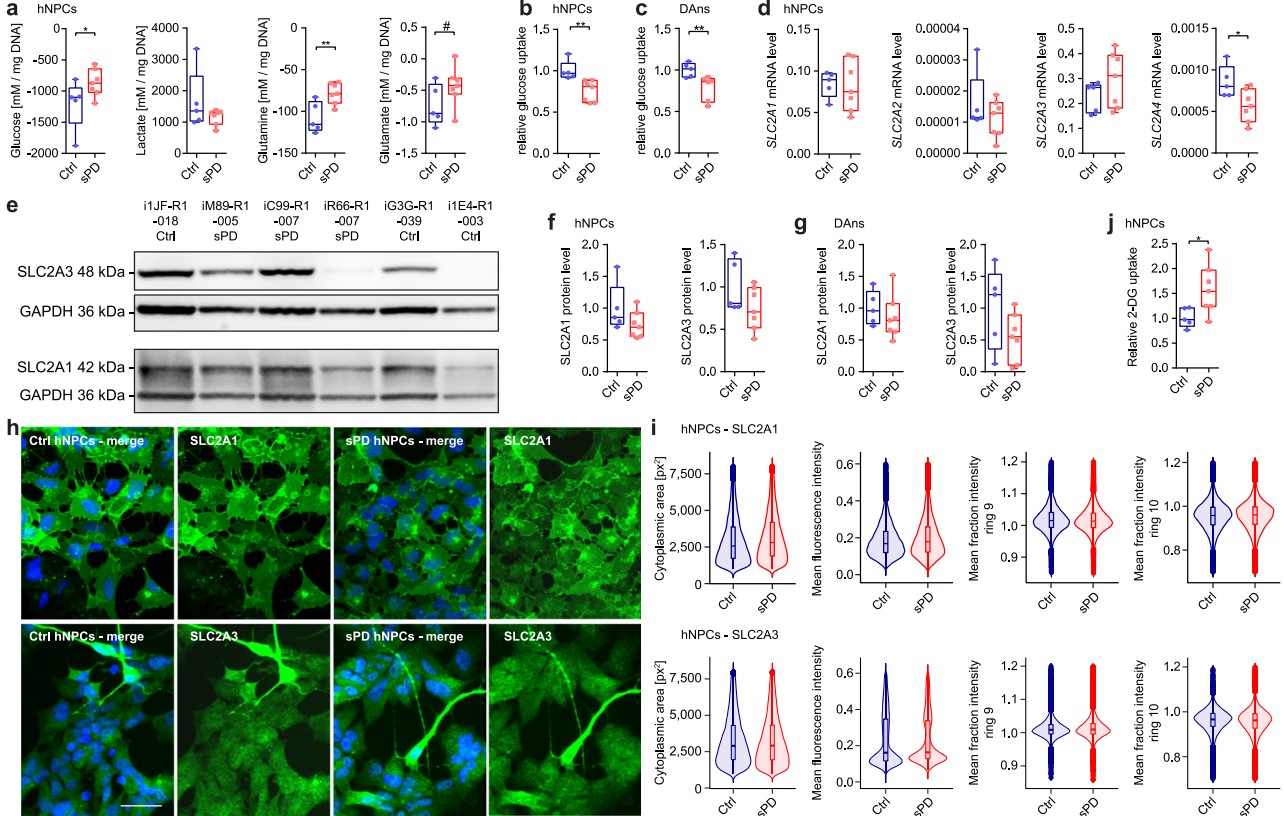

**Fig. 6 | Metabolite uptake and secretion rates indicate a state of hypometabolism in sPD. a** Uptake/secretion rates of main exchange metabolites of hNPCs measured in the medium after 24 h using a YSI biochemistry analyzer. **b** Glucose uptake rates were validated independently in hNPCs and **c** DAns. Values were subtracted from blanks and normalized to mean levels of Ctrl clones. **d** Normalized gene expression levels of glucose transporters analyzed on mRNA level by RT-qPCR in hNPCs. **e** The abundance of glucose transporters on the protein level was quantified by western blot. Western blots are shown exemplarily for some hNPC clones. Quantification of protein levels is shown for **f** hNPCs and **g** DAns. **h** The cellular localization of glucose transporters was visualized using immunostainings. Immunostainings are exemplarily shown for hNPCs of clone i1JF-R1-018 (SLC2A1−Ctrl), iM89-R1-005 (SLC2A1−sPD), iO3H-R1-005 (SLC2A3−Ctrl), and iAY6-R1-003 (SLC2A3−sPD). Scale bar = 20 μm. **i** To quantify the radial distribution of glucose transporters in hNPCs, the cytoplasmic area was separated into ten rings around

the center of the corresponding nuclei, with ring ten being the outermost. On average, 3960 (SLC2A1), 4062 (SLC2A3) cells per clone were analyzed.
**j** Quantification of cellular 2-deoxyglucose (2-DG) levels of hNPCs after a 15 min incubation period. Boxplots display the median and range from the 25th to 75th percentile. Whiskers extend from the min to max value or to the most extreme data point which is no more than 1.5 times the interquartile range (**i**). Each dot represents one patient. $n = 5$ Ctrl and seven sPD patient-derived cell clones, in triplicates or six replicates (**a**). $p$-values were determined by linear mixed effects model (**a**) (glucose: $p = 0.038$; lactate: $p = 0.14$; glutamine: $p = 2.7 \times 10^{-4}$; glutamate: $p = 0.075$) (**i**); two-sided Mann−Whitney $U$ test (**b**) ($p = 0.003$), **c** ($p = 0.009$), **d** (*SLC2A2*: $p = 0.76$); two-sided $t$-test (**d**) (*SLC2A1*: $p = 0.88$; *SLC2A3*: $p = 0.24$; *SLC2A4*: $p = 0.031$), **f** (SLC2A1: $p = 0.16$; SLC2A3: $p = 0.14$), **g** (SLC2A1: $p = 0.46$; SLC2A3: $p = 0.15$), **j** ($p = 0.038$). *$p < 0.05$; **$p < 0.01$; ***$p < 0.001$. Source data are provided as a Source Data file.

transporters SLC2A3 and SLC2A1 was not affected, neither in sPD hNPCs (Fig. 6e, f) nor in DAns (Fig. 6g). In addition, we quantified the cellular and membrane localization of these glucose transporters using immunohistochemical staining in hNPCs (Fig. 6h). Again, the mean fluorescence intensity as well as the radial distribution of SLC2A1 and SLC2A3 was not altered in sPD hNPCs (Fig. 6i).

Still, to assess the overall function of glucose transporters and hexokinase-1 (HK1) which mediates the first step of glycolysis to capture glucose within cells, the uptake and phosphorylation of the glucose analog 2-deoxyglucose was quantified. 2-deoxyglucose is phosphorylated by HK1, but inhibits further glycolytic processing and thus accumulates within cells[44]. Interestingly, within a short period of 15 min, sPD hNPCs consumed significantly more 2-deoxyglucose than Ctrl cells (Fig. 6j), suggesting that the glucose uptake and fixation per se is not affected in sPD and thus, is not the limiting factor responsible for the observed hypometabolism.

### Metabolic flux analysis confirms a bottleneck within the citric acid cycle at the level of the OGDHC in sPD

Based on our assumption of metabolic defects in the citric acid cycle in sPD hNPCs, we performed stable-isotope tracing with either [U-¹³C]

Glutamine or [U-¹³C]Glucose followed by GC-MS measurements of mainly citric acid cycle related metabolites (Supplementary Fig. 4 and Supplementary Data 9). The contribution of ¹³C to metabolites of the central carbon metabolism showed that roughly 20% of carbon within the citric acid cycle was derived from glucose, whereas ~60% was derived from glutamine, highlighting the relevance of also glutamine metabolism for basal metabolic function and energy production (Fig. 7a).

If using a [U-¹³C]Glucose tracer, the entrance of carbon via Acetyl-CoA into the citric acid cycle produced M2-citrate isotopologues or M4-citrate if M2-citrate already passed the citric acid cycle once (Fig. 7b). Thus, the unaltered ratio of M4/M2-citrate between sPD hNPCs and Ctrl indicated a normal flux—also in patient-derived hNPCs—through the citric acid cycle of carbons derived from [U-¹³C]Glucose (Fig. 7c).

Alternatively, [U-¹³C]Glutamine can enter the citric acid cycle via oxidative or reductive metabolism producing M4-citrate or M5-citrate isotopologues, respectively (Fig. 7b). At this level, there seemed to be a rerouting of carbons through the citric acid cycle in sPD, as oxidative glutamine flux was significantly decreased (Fig. 7d) whereas the reductive flux was significantly increased (Fig. 7e). Most likely due to a

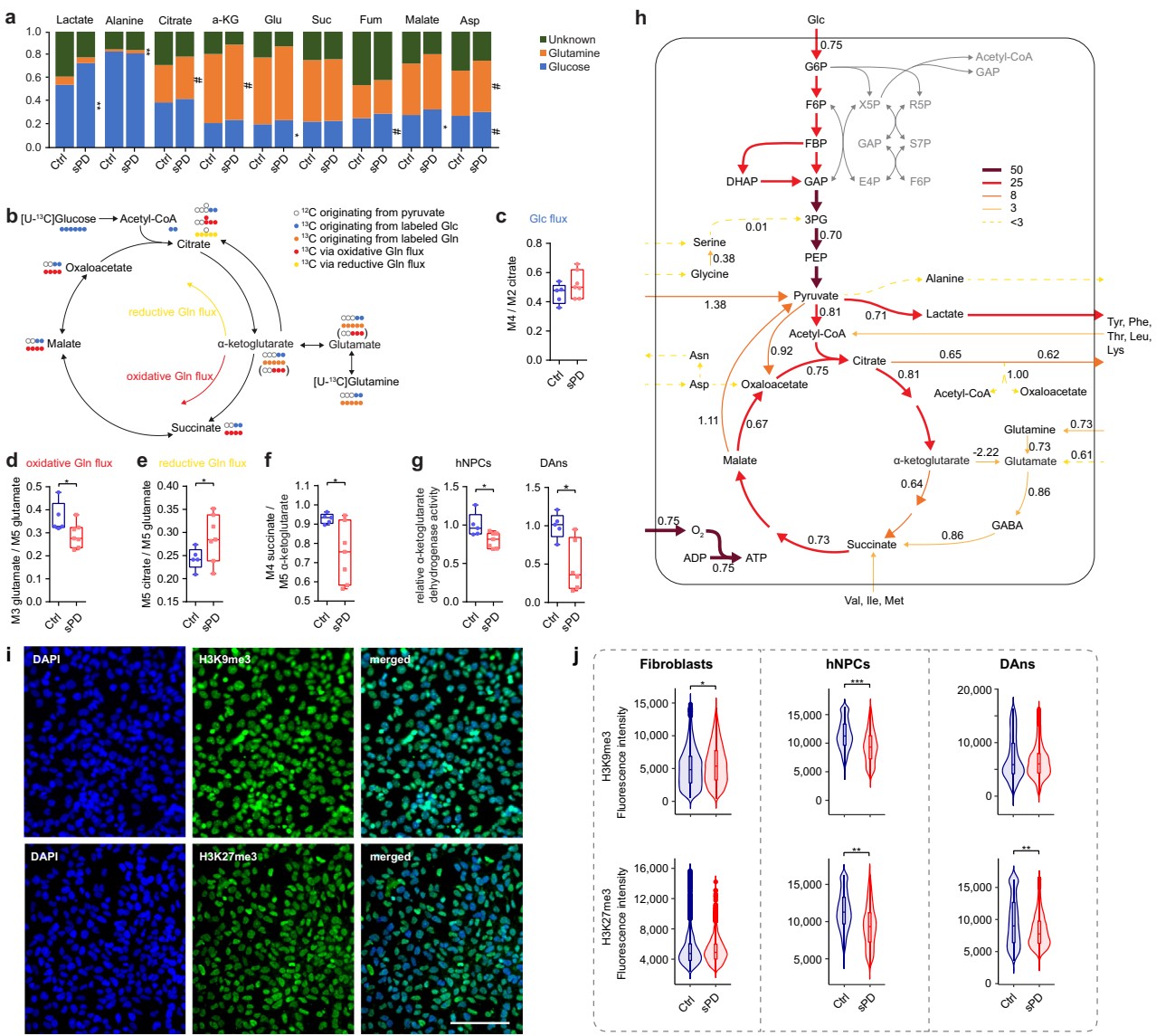

**Fig. 7 | Metabolic flux analysis validates a bottleneck within the citric acid cycle in sPD. a** Fractional contribution of [U-$^{13}$C]Glutamine (Gln) and [U-$^{13}$C]Glucose (Glc) metabolism to metabolite abundance. Calculated from MIDs determined by isotopic tracing after a 24 h incubation period in hNPCs. **b** Overview of carbon transitions for [U-$^{13}$C]Glutamine and [U-$^{13}$C]Glucose metabolism. **c** Cycling of carbons originating from [U-$^{13}$C]Glucose through the citric acid cycle. Ratio of M4- over M2-citrate. **d** Quantification of the oxidative glutamine flux originating from [U-$^{13}$C]Glutamine. Ratio of M3- over M5-glutamate. **e** Quantification of the reductive glutamine flux originating from [U-$^{13}$C]Glutamine. Ratio of M5-citrate over M5-glutamate. **f** Quantification of the α-ketoglutarate dehydrogenase complex activity for carbons originating from [U-$^{13}$C]Glutamine. Ratio of M4-succinate over M5-α-ketoglutarate. **g** Activity of the α-ketoglutarate dehydrogenase complex of (left) hNPCs or (right) DAns. Values were normalized to mean levels of Ctrl hNPCs or DAns, respectively. **h** Metabolic flux map for sPD hNPCs. Arrow thickness and

colors reflect flux values in units of mM/ h/ mg DNA. Fold changes (sPD / Ctrl hNPCs) are noted next to the respective fluxes. Fluxes colored in gray could not be determined with sufficient reliability. **i** Immunostainings for aging-associated histone markers H3K9me3 and H3K27me3. Immunostainings are exemplarily shown for hNPCs of clone iO3H-R1-003. Scale bar = 100 μm. **j** Violin plots show the fluorescence intensity per condition. In total, 600 cells per clone were analyzed. Boxplots display the median and range from the 25th to 75th percentile. Whiskers extend from the min to max value or to the most extreme data point which is no more than 1.5 times the interquartile range (**j**). Each dot represents one patient. $n = 5$ Ctrl and 7 sPD patient-derived cell clones, in triplicates. *p*-values were determined by linear mixed effects model (**a, c**) ($p = 0.15$), **d** ($p = 0.014$), **e** ($p = 0.047$), **f** ($p = 0.046$), **j** two-sided *t*-test (**g**) (hNPCs: $p = 0.024$; DAns: $p = 0.011$). *$p < 0.05$; **$p < 0.01$; ***$p < 0.001$. See also Supplementary Figs. 4–6 and Supplementary Data 9 and 10. Source data are provided as a Source Data file.

reduced activity of the α-ketoglutarate dehydrogenase complex (OGDHC) in sPD, based on the ratio of M4-succinate to M5-α-ketoglutarate (Fig. 7f). This was indeed validated by analyzing the OGDHC activity in cell lysates of these hNPCs as well as DAns differentiated thereof (Fig. 7g) and correlates with the OGDHC activity measured in postmortem PD patients[45]. As only three members of the family of α-ketoacid dehydrogenase multienzyme complexes have been described so far[46], we also assessed the activity of the pyruvate dehydrogenase complex (PDC). The PDC activity, however, was similar in Ctrl and sPD hNPCs (Supplementary Fig. 7b), indicating that it is not

a general dysfunction of this enzyme class per se but is rather specifically affecting the OGDHC activity.

For a better comparison of metabolic differences between Ctrl and sPD, we performed a $^{13}$C-metabolic flux analysis (Supplementary Data 10) by integrating the mass isotopomer distributions yielded from the [U-$^{13}$C]Glucose and [U-$^{13}$C]Glutamine labeling experiments (Supplementary Data 9) with the extracellular uptake/secretion rates (Fig. 6a) into a metabolic network model[47,48]. A simplified metabolic flux map for sPD is shown in Fig. 7h (direct comparison of Ctrl and sPD maps is shown in Supplementary Fig. 5). The fold changes relative to

Ctrl hNPCs are indicated next to the respective fluxes. Generally, sPD hNPCs displayed a reduced metabolic flux (0.75-fold) within glycolysis correlating with a reduced uptake and metabolization of glucose (Fig. 6a). Consequently, also the citric acid cycle flux was reduced in sPD, however, to various extents. The initial steps from Acetyl-CoA fixation to α-ketoglutarate oxidation were reduced by 19%, while the conversion of α-ketoglutarate to succinate was reduced by 36%. Most likely due to a reduced activity of the OGDHC (Fig. 7f, g) as well as an increased reductive glutamine metabolism (Fig. 7e). Instead, the net flux from α-ketoglutarate to glutamate was inverted (−2.22 fold) in sPD hNPCs compared to Ctrl. Thus, indicating a net carbon efflux from the citric acid cycle also supplying the GABA shunt, which is thought to partially compensate for the measured reduction in OGDHC activity. A similar metabolic pattern has been predicted using transcriptome data from substantia nigra of sPD patients[49]. The reduced citric acid cycle flux resulted in a reduced NADH production rate and consequently a reduced $O_2$ consumption and mitochondrial ATP production rate (0.77-fold). Similar values for NADH (Fig. 5e, f, g) and ATP (Fig. 5h) production as well as $O_2$ consumption (Fig. 1b) have been measured in experiments included in this manuscript and further validate the proposed metabolic model for sPD.

Another pathway that seemed to be heavily affected in sPD was the serine-glycine-one-carbon metabolism, which is essential for providing methyl groups for DNA, lipid, and protein modifications. Both the exchange flux from serine to glycine producing M3-serine, M2-glycine, and a methyl group (Supplementary Fig. 4b), as well as the total flux (0.38 fold, Fig. 7h) feeding glycine and serine into the central carbon metabolism and producing M0- or M2-serine, was strongly reduced in sPD. A reduced supply of methyl groups by the one-carbon cycle may further affect methylation patterns in sPD. Indeed, methylation levels of histones shown for H3K9me3 and H3K27me3 were significantly reduced in sPD hNPCs and DAns (Fig. 7i, j and Supplementary Fig. 6). An interference with chromatin organization in sPD may also explain the pattern of global gene dysregulation observed in sPD hNPCs, with most genes being slightly downregulated[17]. Interestingly, alterations in histone modifications were not present in the original patient-derived fibroblasts (Fig. 7j). Both a reduced mitochondrial function (Fig. 1), as well as reduced levels of the histone marks H3K9me3 and H3K27me3 are also well-characterized hallmarks of cellular aging[50], with aging being the greatest risk factor for developing PD[51].

Taken together, the [13]C labeling experiments allowed us to identify the citric acid cycle, specifically at the step of the OGDHC, as a bottleneck in sPD metabolism. Consequently, glucose uptake and flux were reduced in sPD hNPCs, mitochondrial respiration was reduced, as well as fluxes through the one-carbon cycle.

## The state of hypometabolism in sPD is introduced by alterations in SHH signal transduction

We previously reported a connection between the reduced basal mitochondrial respiration (Fig. 1) and an enhanced SHH signal transduction mediated by primary cilia (PC) dysfunction[17]. PC are hair-like organelles that extrude from the cell surface and function as cellular antennas that are thought to mediate the transduction of external signaling events, among which SHH signaling is very prominent[29,30]. SHH binding to its receptor PTCH1 results in its removal from PC, which allows the translocation of SMO into the PC. Subsequently, SMO interferes with the post-translational modification of the respective transcription factors, the GLIs, resulting in distinct transcriptional changes. Using SMO interactors such as cyclopamine (cyc), a cell-permeable steroidal alkaloid, it is possible to repress SMO activity and thus SHH signal transduction[52,53]. By repressing the overactive SHH signal transduction in sPD hNPCs to similar levels as in Ctrl, we previously showed that mitochondrial respiration analyzed by Seahorse XF was restored in sPD to levels similar as in Ctrl[17].

This might be mediated by a direct regulation of OGDHC abundance and thus activity through SHH signaling, as the limiting OGDHC subunit OGDHL is a predicted target gene of the SHH transcription factors GLI2 and GLI3, as well as FOXA2 (Fig. 8a). In line with this, expression levels of OGDHL were misregulated on the transcriptome and proteome level (Figs. 5e, j and 8b), and the reduced OGDHL protein levels could be restored to Ctrl levels upon cyc treatment (Fig. 8b and Supplementary Fig. 7a). Thus, the bottleneck within the citric acid cycle, the OGDHC activity, seemed to be introduced by alterations in PC-dependent SHH signaling in sPD, as also OGDHC activity was rescued after OGDHL expression was normalized by cyc treatment (Fig. 8c). By treating hNPCs with cyc, also the reduced glucose uptake and thus metabolization was rescued in sPD to similar levels as in Ctrls (Fig. 8d). This indicates that the observed state of hypometabolism, the reduced glucose uptake and flux through the central carbon metabolism, and the reduced mitochondrial respiration were introduced specifically in sPD by altered SHH signal transduction. The increased metabolic rate in sPD hNPCs following cyc treatment seemingly also resulted in a sufficient supply of methyl groups for, e.g. post-translational modifications, as also homocysteine levels (Fig. 8e), as well as levels of the histone marks H3K9me3 (Fig. 8f) and H3K27me3 (Fig. 8g) were restored in sPD to similar levels as in Ctrl.

Interestingly, these SHH-dependent changes in basal and also maximal mitochondrial respiration occurred independently of the reduced activity of the complex I of the electron transport chain. Following cyc treatment, complex I activity was still reduced by ~30% in sPD hNPCs (Fig. 8h), indicating that the observed complex I deficiency is not reflected in the mitochondrial respiration measured by Seahorse XF, nor is it affecting basal cellular metabolism.

Thus, complex I deficiency is a pathological hallmark independent of alterations in SHH signaling and OGDHC deficiency. However, it is not yet a limiting factor contributing to the more severe metabolic alterations associated with sPD that have been observed in hNPCs and DAns.

## Multiple-factor analysis based on the characterization of hNPCs and DAns allows to stratify patients

In a next step, the experimental data were used to stratify the patients that donated the skin fibroblasts used for hiPSC generation according to the severity of their molecular alterations. To do so, a multiple-factor analysis (MFA) was performed including detailed information gained from the scRNA-seq, proteome analysis, and non-targeted metabolomics analysis. In total, five groups of variables were used: the top 100 DEGs, the top 100 DEPs, the significantly altered metabolites (45 metabolites), as well as the key metabolic and ciliary parameters (61 functional parameters), and as a supplementary group, the information about the disease state (Ctrl or sPD—Table 1).

The results show that dimensions 1 and 2 explain together about 42% of the variability observed in patient-derived cells (Fig. 9a). The first dimension represents mainly the functional parameters and altered metabolites, whereas DEGs and DEPs mainly contribute to dimension 2 (Fig. 9b). Of particular interest is the separation of Ctrl and sPD patients by dimension 1 (Fig. 9a). The larger the distance between a sPD patient and the average of Ctrl patients, the larger is the molecular deviation which might reflect the severity of the disease state.

To validate this hypothesis, the sPD patients were clinically examined at the timepoint of skin biopsy and were assessed again after a mean of 10.7 years (range 9–12 years). Both at baseline and at follow-up, all PD patients fulfilled the clinical diagnostic criteria for PD[54] and were defined as having sPD by the absence of known PD-causing familial mutations (PARK1–18) and a negative family history of PD[16]. Disease progression was assessed by monitoring changes in PD-associated scores determined according to the Hoehn & Yahr scale (H&Y), part III (motor symptoms) of the Unified Parkinson's Disease

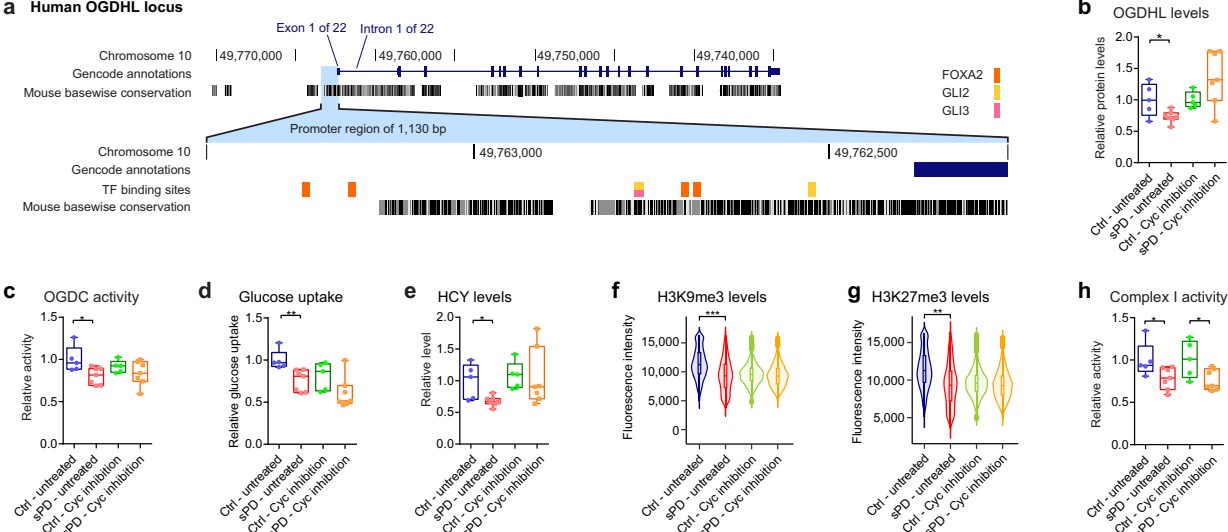

**Fig. 8 | Alterations in SHH signal transduction underly the hypometabolism observed in sPD. a** Visualization of the human OGDHL locus and its alignment to the mouse genome. The transcription factor bindings sites of GLI1, GLI2, GLI3, and FOXA2 are highlighted. **b** The abundance of OGDHL was quantified by western blot. Expression levels were normalized to GAPDH. **c** Activity of the α-ketoglutarate dehydrogenase complex. Values were normalized to mean levels of Ctrl hNPCs. **d** Glucose uptake rates. Quantified were glucose levels in the growth medium after a 24 h incubation period. Values were subtracted from blank values and normalized to mean levels of Ctrl hNPCs. **e** Cellular homocysteine (HCY) levels. Values were normalized to mean levels of Ctrl hNPCs. **f** Quantification of histone marks H3K9me3 and **g** H3K27me3. Analyzed were 600 cells per clone. **h** Analysis of relative complex I activity. Values were normalized to mean levels of Ctrl hNPCs.

Boxplots display the median and range from the 25th to 75th percentile. Whiskers extend from the min to max value or to the most extreme data point which is no more than 1.5 times the interquartile range (**f**, **g**). Each dot represents one patient. Experiments were performed in hNPCs (DMSO ctrl and cyc treated−10 μM for 4 days). $n = 5$ Ctrl and 7 sPD patient-derived cell clones, in triplicates. p-values were determined by linear mixed effects model (**f**) (untreated: $p = 0.56$; cyc: $p = 5.2 \times 10^{-5}$), (**g**) (untreated: $p = 0.004$; cyc: $p = 0.85$); two-sided $t$-test (**b**) (untreated: $p = 0.030$; cyc: $p = 0.10$), (**c**) (untreated: $p = 0.024$; cyc: $p = 0.30$), (**e**) (untreated: $p = 0.017$; cyc: $p = 0.88$), (**h**) (untreated: $p = 0.048$; cyc: $p = 0.032$); two-sided Mann–Whitney-$U$ test (**d**) (untreated: $p = 0.003$; cyc: $p = 0.11$). *$p < 0.05$; **$p < 0.01$; ***$p < 0.001$. See also Supplementary Fig. 7. Source data are provided as a Source Data file.

Rating Scale (UPDRS), as well as the activities of daily living (ADL) scale (Table 1).

Indeed, dimension 1 can also be used to separate subgroups of sPD patients with slow versus fast disease progression (see also Table 1). As patient R66, who was lost to follow-up, clustered together with the fast progression group, it may be possible that R66 also exhibited a faster disease progression.

## OGDHC deficiency and metabolic alterations correlate with disease progression in sPD patients

In a further translational effort, we correlated our metabolic in vitro findings with disease progression of the sPD patients that donated the

## Table 1 | Disease progression in sPD patients within 10 years after biopsy

| Time of biopsy | | | Clinical changes ~10 years after biopsy | | | |
|---|---|---|---|---|---|---|
| Patient ID | Gender | Years of illness | Δ H&Y | Δ UPDRS III [points] | Δ L-Dopa equivalent [mg] | Δ ADL |
| J2C | m | 3 | 3 | 24 | 400 | −0.6 |
| M89 | m | 3 | 3 | 53 | 1563 | −0.6 |
| C99 | m | 7 | 2.5 | 36 | 610 | −0.5 |
| R66 | m | 3 | Follow-up not available | | | |
| AY6 | M | 4 | 1 | −1 | 800 | −0.2 |
| PX7 | M | 1 | 1 | 0 | 1900 | −0.1 |
| 88H | F | 6 | 2 | 7 | 640 | −0.3 |

Left: Description of sPD patients at the timepoint of skin biopsy (Gender, and time in years between sPD diagnosis and tissue biopsy). Right: Long-term history of sPD patients. Changes in the Hoehn&Yahr scale (ΔH&Y), in motor examinations (Part III) according to the Unified Parkinson Disease Rating Scale (ΔUPDRS III), medication requirement (ΔL-Dopa equivalent), and activities of daily living (ΔADL) monitored within 9–12 years after skin biopsy.

skin fibroblasts used for hiPSCs generation. A subset of variables is displayed in Fig. 9c or as scatter plots in Fig. 9d.

In line with our observations regarding the impact of OGDHC deficiency in sPD, a strong link existed between the rate of disease progression and OGDHC activity. In both, hNPCs and DAns, OGDHC activity correlated with disease progression (ΔH&Y (hNPCs – $R = -0.93$; $p = 0.006$) (DAns – $R = -0.96$; $p = 0.009$), ΔADL (hNPCs – $R = 0.95$; $p = 0.004$) (DAns – $R = 0.89$; $p = 0.046$), and in parts ΔUPDRS III (hNPCs – $R = -0.89$; $p = 0.017$)). Furthermore, in patients but not in Ctrls OGDHC activity correlated with cellular glucose consumption indicating a link between reduced glucose uptake and the reduced activity of the enzyme complex. Complementary, also the cellular glucose uptake was linked to disease progression (e.g. hNPCs – $R = 0.9$; $p = 0.015$).

In addition, OGDHC deficiency in sPD remained stable during the differentiation from hNPCs to DAns ($R = 0.82$; $p = 0.043$). This again validates the usability of hNPCs for disease modeling.

Interestingly, the ciliary capacity to transduce SHH signaling correlated with metabolic parameters. The capacity was assessed by monitoring the nuclear levels of GLI3-full length (GLI3-FL) which is thought to function as a weak transcriptional enhancer and the truncated GLI3-repressor (GLI3-R) form. An increase of the GLI3-FL/GLI3-R (enhancer/repressor) ratio positively correlated with the protein abundance of the predicted SHH target OGDHL in hNPCs ($R = 0.98$; $p- = 0.00047$). OGDHL protein levels thereby positively correlated with overall OGDHC activity ($R = 0.73$; $p = 0.098$). These findings together with the correlations of disease progression with OGDHC activity and OGDHL protein levels (ΔH&Y ($R = -0.82$; $p = 0.046$) further strengthen the observed impact of ciliary-mediated SHH signaling on sPD onset and progression.

A similar separation of patients versus controls as by dimension 1 (Fig. 9a) could be achieved by only plotting the OGDHC activity

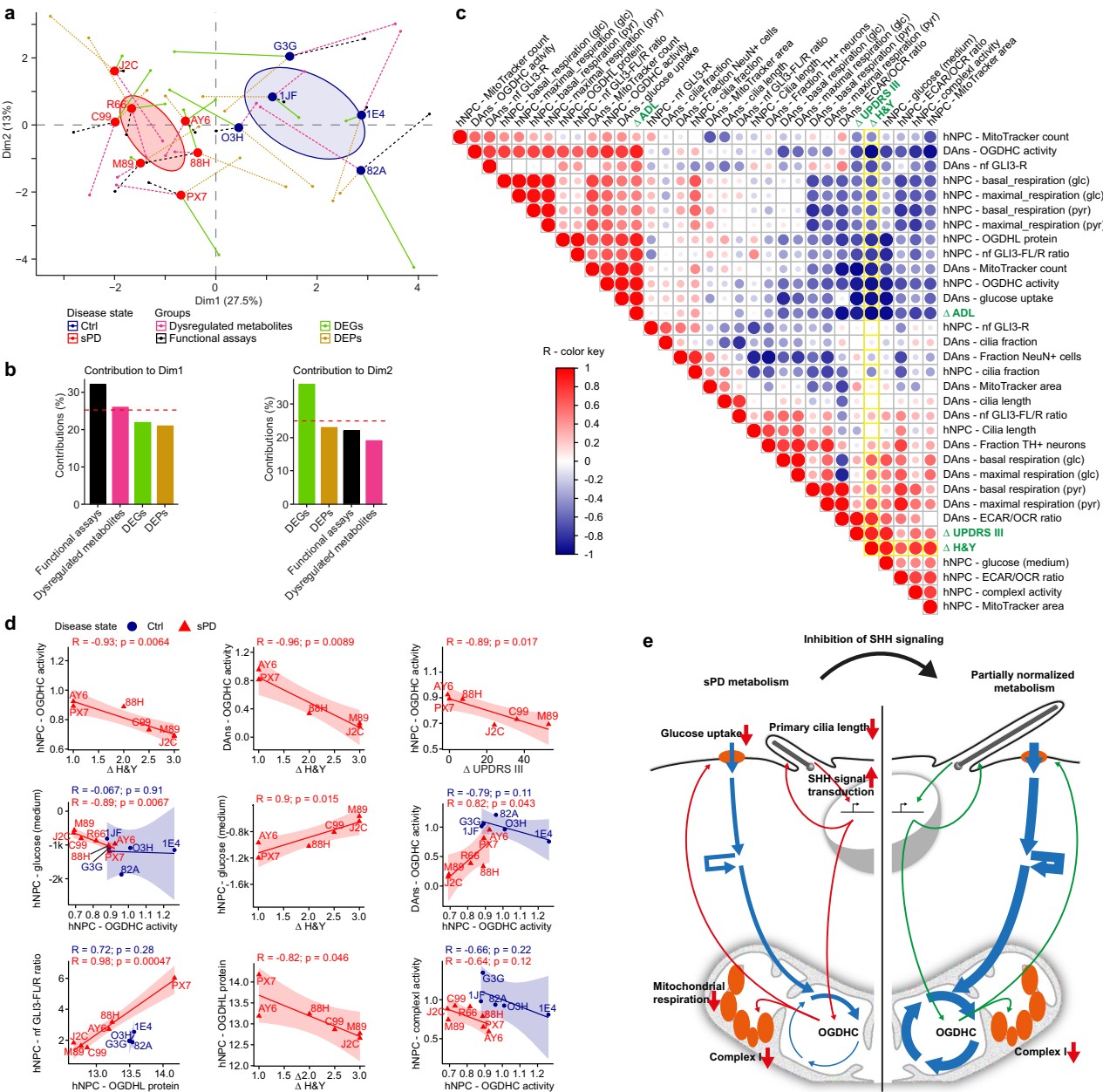

**Fig. 9 | OGDHC activity and metabolic alterations correlate with disease progression in sPD patients. a** Multiple-factor analysis using differentially expressed genes (DEGs; bulk-like; top 100) (Fig. 3 and ref. 17), differentially expressed proteins (DEPs; top 100) (Fig. 4), significantly altered metabolites (45 metabolites) (Fig. 5), as well as the key experimental variables displayed in Figs. 1–7 and ref. 17 describing alterations in sPD (61). The group points and lines represent the patient coordinates conditioned by the corresponding group variables. **b** Contributions of quantitative variables to the dimension 1 and 2 of the multiple-factor analysis. **c** Correlogram visualizing the Pearson correlation coefficients of variables measured in sPD hNPCs and DANs, as well as parameters associated with disease progression in the sPD patients over a period of 9–12 years (Table 1). Disease progression markers are highlighted in green. Correlations with the ΔH&Y scale are highlighted in yellow. Colors and dot sizes are proportional to the correlation coefficients. Negative correlations are colored in blue, positive correlations in red. **d** Visualization of interesting dependencies within measured variables and parameters of disease progression in sPD patients. Each point represents one patient. Linear regressions were calculated for each group and are displayed with a 95% confidence interval. The Pearson correlation coefficient (R) is displayed next to the corresponding $p$-value ($p$) (two-sided). Correlations with $p < 0.1$ were considered significant. **e** Graphical overview summarizing the main findings.

measured in hNPCs against the respective complex I activity (Fig. 9d). Although activity levels of both enzymes did not significantly correlate in sPD ($R = -0.64$; $p = 0.12$) or Ctrl ($R = -0.66$; $p = 0.22$) samples, they clearly separated the individuals by the disease state. Furthermore, the separation was clearer in this case than if parameters would have been considered alone (Fig. 7g and ref. 17).

In sum, we present here a human cellular model system that combines—and allows to model— most of the known sporadic PD-associated metabolic alterations. Based on our findings, we propose a mechanism in which PC dysfunction underlies the onset of most of these metabolic alterations (Fig. 9e). Dysfunctional PC thereby affect SHH signal transduction which in turn results in altered gene expression patterns amongst others of the OGDHC. This creates a bottleneck within the citric acid cycle and thus reduces the flux through the main metabolic routes of glycolysis, citric acid cycle, and OXPHOS resulting in a sPD-specific state of hypometabolism. Contrary, complex I deficiency seems to evolve independently of the PC-mediated metabolic alterations. Thus, we present a model in which complex I deficiency

and the SHH-mediated hypometabolism develop as two independent hits that negatively impact cellular energy supply and could be used to predict disease progression in sPD patients.

## Discussion

The patient-derived hiPSC-based system described in this study to model sporadic Parkinson's disease is the first that integrates most patient-associated metabolic phenotypes into one comprehensive molecular and mechanistic model. Focusing on mitochondrial dysfunction, we showed that neuronal cells—hNPCs and DAns—derived from sPD patients progress into a symptomatic state with severe mitochondrial alterations in e.g. mitochondrial ATP production and complex I functionality. Although these cells exhibit an sPD-specific mitochondrial phenotype, they do not degenerate[17]. Thus, this model system can provide unique insights into cellular processes and molecular mechanisms early during sporadic PD etiology before the onset of neurodegeneration. Processes that can be targeted to slow down, halt, or even prevent disease progression before the brain is irreversibly damaged.

To unravel these mechanisms, we performed a multilayered omics analysis based on transcriptomics, proteomics, and metabolomics using neuronal cells of the dopaminergic lineage. The multilayered omics analysis allowed us to identify a crucial bottleneck in sPD metabolism within the citric acid cycle, which is the most important metabolic pathway for the cellular energy supply, especially within the brain. Furthermore, the citric acid cycle is thought to be the central wheel connecting almost all individual metabolic pathways so that a blockage within the cycle can affect the abundance of metabolites involved in most metabolic processes[37,55,56]. Especially the α-ketoglutarate dehydrogenase complex (OGDHC) activity was significantly reduced in sPD hNPCs and DAns derived thereof, which is the rate-limiting step in the citric acid cycle. This reduced activity might be caused by the altered abundance of OGDHC subunits. Interestingly, the abundance of the rate-limiting subunit OGDH and its brain-specific isoform OGDHL was affected, both of which are essential for normal OGDHC function in neuronal cells. On the protein level, the ubiquitously expressed OGDH is upregulated in sPD, and contrary, the brain-specific OGDHL[57] isoform is downregulated.

Interestingly, alterations in OGDHC activity have been implicated in multiple neurodegenerative diseases such as Huntington's disease, Alzheimer's disease (reduction of ~30–90%[58]), and Parkinson's disease (reduction of ~50%[45,59]) to various degrees. Also, OGDHC abundance is reduced in postmortem brain regions of PD patients[60]. A central role of OGDHC activity for the progression of sPD is also implied by the present correlation analysis revealing that OGDHC activity is tightly correlated with clinical markers of disease progression over a long-term follow-up.

As a consequence of altered OGDHC function in sPD, also several OGDHC-dependent cellular processes could be affected as well, such as α-ketoglutarate metabolization or cellular reactive oxygen species (ROS) signaling.

Concerning the α-ketoglutarate metabolization, the OGDHC activity is thought to determine if the citric acid cycle acts as an oxidative pathway to produce energy in form of reducing equivalents or as a reductive pathway to produce intermediates for biosynthetic processes. A reduced OGDHC activity due to, e.g. OGDHL mutations, has been shown to result in reduced mitochondrial respiration and ATP production[61], an depletion of fumarate, and malate[61], as well as a reduced cortical glucose consumption[62,63]. In fact, this is in line with our data concerning the metabolic alterations in sPD. In sPD hNPCs and DAns, a reduced OGDHC activity is combined with a reduced mitochondrial ATP production rate, reduced levels of both fumarate, and malate, as well as reduced glucose consumption. Furthermore, α-ketoglutarate is rerouted at the step of the OGDHC[58] instead of being processed to succinate. The oxidative glutamine flux was significantly

decreased, whereas the reductive flux was significantly increased. This resulted in sPD hNPCs in an efflux of carbon out of the citric acid cycle at the level of α-ketoglutarate, diminishing the production of reducing equivalents. This is in line with metabolomic flux analysis that revealed enhanced reductive carboxylation upon genetic deletion of OGDHC subunits[64]. Thus overall, the alterations in OGDHC activity seemed to introduce an sPD-specific state of hypometabolism marked by reduced glucose and glutamine metabolization, a reduced mitochondrial ATP production rate, as well as a reduced abundance of most quantified metabolites. Alterations in glucose metabolization have also been previously linked to sPD using positron emission tomography (PET) imaging[39–41]. Basal ganglia neurons, DAns, and also cholinergic neurons receiving the dopaminergic inputs, might be especially vulnerable as they heavily rely on the citric acid cycle and electron transport chain due to their high energetic burden and thus exhibit high levels of OGDHC[58].

Second, the OGDHC is also thought to function as a redox sensor to reduce the NADH output of the citric acid cycle to the electron transport chain in situations of oxidative stress[65]. Depending on ROS levels, the OGDHC activity is reversibly reduced. This might be particularly crucial for ROS homeostasis and thus cellular health in combination with an already reduced OGDHC activity, as well as a misassembled and dysfunctional complex I as observed in our cellular model[17] and in postmortem material of sPD patients[26,66]. Furthermore, cellular antioxidant capacities may also be limited due to a reduced NAD(P)H production rate as a consequence of the observed hypometabolism/metabolic flux, since NADPH powers the majority of ROS-detoxifying enzymes such as the glutathione reductases or thioredoxin reductases[67]. Besides being sensitive to ROS, the OGDHC is also a prominent source of superoxide or $H_2O_2$ production exceeding the respective levels of complex I by about eight times[68].

In sum, we present here a human cellular system that combines—and allows to model—most of the known sporadic PD-associated metabolic alterations such as reduced glucose metabolization, reduced mitochondrial respiration, and thus ATP production, as well as an OGDHC and complex I deficiency. Thus, it can provide unique insights into the molecular mechanisms underlying the metabolic alterations early during sporadic PD etiology that can be targeted for therapeutic purposes. It further allows to investigate causalities between these PD-associated alterations.

Possibly of therapeutic potential is the finding that most of the described metabolic alterations, especially the observed hypometabolism were evoked by altered SHH signal transduction in sPD hNPCs. We previously reported that these sPD hNPCs and DAns developed alterations in primary cilia function which manifest in an overactive SHH signal transduction in sPD[17]. Upon inhibiting the enhanced SHH signal transduction in sPD, glucose uptake and the activity of the OGDHC could be restored. Interestingly, the misexpressed OGDHL subunit which is thought to result in altered OGDHC activity is also a predicted SHH/GLI3 target gene. Thus, primary cilia dysfunction and alterations in SHH signal transduction are the underlying causes of OGDHC dysfunction which in turn creates a bottleneck within the citric acid cycle causing an sPD-specific state of hypometabolism. Therefore, inhibiting the overactive SHH signaling may be one part of a potential neuroprotective therapy during early stages of sPD to attenuate the metabolic alterations. Independently of the SHH signaling mediated alterations in metabolism seems to develop the complex I deficiency in sPD hNPCs, displaying a second hit that further contributes to the severe metabolic alterations.

Astonishing, however, is that external stimuli that are thought to have a beneficial effect on DAn functionality and survival negatively impact internal metabolic processes in an sPD-specific manner. Under normal conditions, SHH signaling mediates cellular and neurochemical homeostasis within the nigrostriatal circuits, whereas diminished SHH signaling results in DAn degeneration[69]. Contrary, our sPD patient-

derived hNPCs already react to physiological SHH signaling unusually sensitive, initiating a cascade that ultimately results in an sPD-specific state of hypometabolism. The mechanisms underlying the development of primary cilia dysfunction and thus most metabolic alterations remain still unknown and have to be investigated in the future.

Based on these data, we present a model in which complex I deficiency and the SHH-mediated hypometabolism develop as two independent hits that negatively impact cellular energy supply as well as ROS homeostasis, but also each other, thereby creating a vicious cycle of self-destruction[66,70]. To interfere with both processes by firstly inhibiting the overactive SHH signaling and secondly by rescuing complex I deficiency might be a combinatorial neuroprotective therapy beneficial during early stages of sPD.

## Methods

### Ethical compliance

The use of patient-related material, information, and cell culture work was approved by local ethics committees (No. 4485, No. 4120, No. 17-259, FAU Erlangen-Nuernberg, Germany; and No 422-13 and 357/19 S, Technical University Munich, Germany) and all participants or their legal guardians gave written informed consent. Participants were not compensated. All related examinations, experiments, and methods were performed in accordance with relevant guidelines and regulations. Individuals were recruited independently of their sex/gender, race, ethnicity, or other socially relevant categories. The sex and gender was determined based on self-report.

sPD and Ctrl hiPSC lines were established, characterized, and provided by the ForIPS consortium[16,17]. hiPSC lines from 7 sPD patients and 5 Ctrls were used, in total 24 hiPSC lines with two clones per patient (Supplementary Data 1). To exchange selected sPD lines for research purposes the scientific board of the UKER biobank will consider each request.

### Cell culture hiPSCs

hiPSCs were maintained under feeder-free conditions on Geltrex (Thermo Fisher Scientific) coating and in mTesR1 medium (StemCell Technologies) at 37 °C, 5% $CO_2$, 21% $O_2$. At 70% confluency, cells were detached using StemMACS Passaging Solution XF (Miltenyi Biotec) for 6 min at 37 °C. StemMACS Passaging Solution XF was aspirated and hiPSC colonies were harvested in 1 ml mTeSR1 and chopped using a 1000 µl pipette tip. Harvested hiPSCs were diluted and seeded on Geltrex-coated plates. hiPSCs were cultured for ~200 days with passaging every 5–7 days (~60 passages). hiPSCs were screened regularly for pluripotency and stable karyotype[17].

### Neural precursor differentiation

hiPSCs were differentiated into human small molecule neural progenitor cells (hNPCs) by embryoid body (EB) formation as described previously[17]. At 70% confluency, hiPSC colonies were detached using 2 mg/ml collagenase type IV (Thermo Fisher Scientific) for 40 min at 37 °C, 5% $CO_2$, 21% $O_2$. Detached hiPSC colonies were harvested in knockout serum replacement medium (80% KnockOut DMEM (Life Technologies) supplemented with 20% Knockout serum replacement (Life Technologies), 10 µM SB431542 (Miltenyi Biotec), 0.5 µM purmorphamine (Tocris Bioscience), 1 µM dorsomorphin (Tocris Bioscience), 3 µM CHIR99021 (Tocris Bioscience), 4.44 nM FGF-8b (Miltenyi Biotec), 1% non-essential amino acids (Life Technologies), 1% L-glutamine (Life Technologies), 150 µM ascorbic acid 2-phosphate (Sigma), and 0.02% beta-mercaptoethanol (Life Technologies)). hiPSCs were cultivated for two days on an orbital shaker at 37 °C, 5% $CO_2$, 21% $O_2$ and 80 rpm with daily media changes. At day 3, EBs were transferred into neuronal precursor medium (50% DMEM/F12-GlutaMAX (Life Technologies), 50% Neurobasal (Life Technologies) supplemented with 1% B27 (minus vitamin A, 12587010, Life Technologies), 0.5% N2 (Life Technologies), 10 µM SB431542, 0.5 µM purmorphamine, 1 µM

dorsomorphin, 3 µM CHIR99021, 4.44 nM FGF-8b, 150 µM ascorbic acid 2-phosphate, and 0.02% beta-mercaptoethanol). EBs were cultivated for 4 days at 37 °C, 7% $CO_2$, 21% $O_2$, and 80 rpm with daily media changes. On day 6, EBs were transferred into neural precursor maintenance medium (neural precursor medium deprived of purmorphamine and CHIR99021). The following day, EBs were seeded unharmed on Geltrex-coated plates and expanded for 3–4 days in neural precursor maintenance medium at 37 °C, 7% $CO_2$, 21% $O_2$ with daily media changes. At day 10-11, hNPCs were dissociated using Accutase (Sigma) for 10 min at 37 °C, 7% $CO_2$, 21% $O_2$, chopped with a 1000 µl pipette tip and diluted in 5 ml neural precursor maintenance medium. hNPCs were harvested at 200 × g for 5 min, resuspended in 1 ml neural precursor maintenance medium and seeded on Geltrex-coated plates. Thereafter, hNPCs were cultivated at 37 °C, 7% $CO_2$, 21% $O_2$ with daily medium changes and passaged with Accutase at 80% confluency. After two passages, 0.5 µM purmorphamine was added to the maintenance medium.

### Dopaminergic neuron differentiation

DAn differentiation was done as described previously[17]. hNPCs were differentiated to DAns using 15 µg/mL poly-L-ornithine (Sigma) and 10 µg/mL laminin (Thermo Fisher Scientific) coating and differentiation steps were done precisely as described previously[17] using the following growth factors and supplements: 10 ng/mL human BDNF (Miltenyi Biotec); 10 ng/mL human GDNF (Miltenyi Biotec); 1 ng/mL human Tgf-beta3 (Miltenyi Biotec); 100 ng/mL human FGF-8b (Miltenyi Biotec); 200 µM ascorbic acid 2-phosphate (Sigma); 1 µM purmorphamine (Tocris); 500 µM dbcAMP (Sigma). DAns were differentiated at 37 °C, 7% $CO_2$, 21% $O_2$ for at least 50 days.

### Respiratory analysis

Respiratory analysis was done as described previously[17].

hiPSCs were maintained on Geltrex coating and in mTesR1 medium at 37 °C, 5% $CO_2$, 21% $O_2$. At 70% confluency, hiPSCs colonies were passaged using Accutase for 6–8 min at 37 °C, sheared to single cells by pipetting using a 1000 µl tip, and diluted in mTeSR medium containing 10 µM ROCK inhibitor Y27632. hiPSCs were harvested at 100 × g for 5 min and resuspended in 1 ml mTeSR medium containing 10 µM ROCK inhibitor Y27632. Cell number was determined and 5000 cells per well with at least 8 replicates per cell line were seeded in mTeSR medium containing 10 µM ROCK inhibitor Y27632 on Geltrex-coated XF96 cell culture microplates and incubated for 6 days with daily medium changes.

hNPCs were maintained on Geltrex coating and in neural precursor maintenance medium at 37 °C, 7% $CO_2$, 21% $O_2$. At 80% confluency, hNPCs were passaged, cell number was determined and 70,000 cells per well with at least 8 replicates per cell line were seeded on Geltrex-coated XF96 cell culture microplates and incubated for 48 h with daily medium changes.

DAns were passaged on day 9 of differentiation according to the differentiation protocol, cell number was determined and 5000 cells per well with at least 8 replicates per cell line were seeded on 15 µg/mL poly-L-ornithine and 10 µg/mL laminin coated XF96 cell culture microplates. DAns were matured for at least 30 days at 37 °C, 7% $CO_2$, 21% $O_2$ with medium changes every 2–5 days.

The respiratory analysis was performed using an XF96 Analyzer (Seahorse Bioscience) and XF Assay medium (Seahorse Bioscience) supplemented with 25 mM glucose (Sigma) or 5 mM pyruvate (Sigma). Prior to measurements, cells were washed once with XF Assay medium and at least 4 replicates per patient were incubated with XF Assay medium containing glucose or pyruvate for 1 h at 37 °C, 0% $CO_2$, 21% $O_2$. XFe96 Sensor Cartridges (Seahorse Bioscience) were hydrated with 200 µl Calibrant (Seahorse Bioscience) per well overnight and ports were loaded with (A) 10 µg/mL oligomycin (Sigma), (B) 5 µM carbonyl cyanide p-trifluoromethoxyphenylhydrazone (FCCP) (Sigma), (C)

50 μM rotenone (Sigma) and 20 μM antimycin A (Sigma), and (D) 1 M 2-deoxyglucose (Sigma). After equilibration, the analysis was performed with a total of four basal measuring points (mix for 1 min, time delay for 2 min, and measure for 3 min). Subsequent to port injections, respiratory and glycolytic flux analysis was performed with three measuring points. Data were analyzed using Wave 2.6.1. Data were normalized to DNA content analyzed on an equally seeded and grown plate (fibroblasts, hiPSCs, hNPCs) or the same plate (DANs) using the Quant-iT PicoGreen dsDNA Assay Kit (Thermo Fisher Scientific). The copy plate (medium aspirated) or the same plate was stored at −20 °C immediately after the Seahorse run was performed and thawed on ice for 30 min prior to analysis. For the copy plate, cells were lysed in 60 μl RIPA buffer (50 mM Tris-HCL (Sigma), 150 mM NaCl (Sigma), 1% Triton X-100, 0.5% sodium deoxycholate (Sigma), 0.1% SDS (Sigma), 3 mM EDTA (Sigma)) per well. For the same plate, 10 μl Proteinase K (20 mg/ml; Sigma-Aldrich) was added per well and cells were lysed at 37 °C for 1 h. The Quant-iT PicoGreen dsDNA assay was performed according to the manufacturer's instructions. DNA concentrations were calculated with a linear regression curve using Lambda DNA standards. For statistical analysis of basal and maximal mitochondrial respiration as well as proton leak, the mean OCR values per patient of measuring points prior to port A injection, after port B and prior to port C injection, or after port A and prior to port B injection, respectively, were used. The last two measuring points were used to subtract non-mitochondrial respiration or non-glycolytic acidification. For DANs, only the last measuring point was used to subtract the non-glycolytic acidification. Mitochondrial ATP production rates were calculated using the following equation with a P/O ratio of 2.75 pmol ATP/pmol O according to the manufacturer's instructions (https://www.agilent.com/cs/library/whitepaper/public/whitepaper-quantify-atp-production-rate-cell-analysis-5991-9303en-agilent.pdf).

$$
\begin{aligned}
&\text{mitoATP Production Rate (pmol ATP/ min)} \\
&= (\text{OCR}_{basal}\,(\text{pmol O2/ min}) - \text{OCR}_{proton\ leak}\,(\text{pmol O2/ min})) \quad (1) \\
&\quad * 2\,(\text{pmol O/pmol O2}) * \text{P/O}\,(\text{pmol ATP/pmol O})
\end{aligned}
$$

### Analysis of total ATP levels

At 80% confluency, hNPCs were passaged, cell number was determined, and 10,000 cells per well with at least three replicates per cell line were seeded on Geltrex-coated white 96-well plates and incubated for 48 h with daily medium changes at 37 °C, 7% CO₂, 21% O₂. Total ATP levels were analyzed using a Luminescent ATP Detection Assay Kit (Abcam) according to the manufacturer's instructions. Values were normalized to DNA content analyzed on an equally seeded and grown copy plate using the Quant-iT PicoGreen dsDNA Assay Kit (Thermo Fisher Scientific) as described above for Respiratory analysis.

### Analysis of total NADH/NAD⁺ levels

At 80% confluency, hNPCs were passaged, cell number was determined, and 5,000,000 cells per sample were resuspended in 100 μl lysis buffer. Total NADH and NAD⁺ levels were analyzed using a NAD/NADH Assay Kit (ab176723; Abcam) according to the manufacturer's instructions. Fluorescence intensity was quantified using a SpectraMax M5 (Molecular Devices). Values were normalized to genomic DNA content analyzed using additional 5,000,000 cells from the same cell solution. Cells were lysed in 100 μl RIPA buffer and genomic DNA was quantified using the Quant-iT PicoGreen dsDNA Assay Kit (Thermo Fisher Scientific) as described above for Respiratory analysis.

### Glucose uptake

The cellular glucose uptake was quantified using the Glucose Assay Kit (ab272532, Abcam) and the Glucose Uptake-Glo™ Assay (J1341, Promega) according to the manufacturer's instructions.

For the Glucose Assay Kit (ab272532, Abcam), hNPCs were passaged, cell number was determined, and 40,000 cells per well with at least 2 replicates per cell line were seeded on Geltrex-coated 96-well plates and incubated for 48 h with daily medium changes at 37 °C, 7% CO₂, 21% O₂. The medium was aspirated, and cells were incubated in fresh neural precursor maintenance medium for exactly 24 h. The medium was collected and diluted 1:2 in ddH₂O. In total, 2 μl of each dilution were mixed with 200 μl reagent and incubated at 95 °C for 8 min and at 8 °C for 4 min. The light absorbance at a wavelength of 630 nm of each sample was measured using a SpectraMax M (Molecular Devices). Glucose concentrations were calculated with a linear regression curve using the provided standards. To calculate the glucose uptake rates, average glucose concentrations per cell line were subtracted from the average blank value measured from two equally treated wells only containing medium.

For normalization of the glucose uptake rates, remaining cells were lysed in 60 μl RIPA buffer and genomic DNA was quantified using the Quant-iT PicoGreen dsDNA Assay Kit (Thermo Fisher Scientific) as described above for the Respiratory analysis.

For the Glucose Uptake-Glo™ Assay (J1341, Promega), hNPCs were passaged, cell number was determined, and 20,000 cells per well with at least two replicates per cell line were seeded on Geltrex-coated 96-well plates and incubated for 48 h with daily medium changes at 37 °C, 7% CO₂, 21% O₂. The medium was aspirated, cells were washed with PBS, and incubated in PBS containing 1 mM 2-deoxyglucose for 15 min at 37 °C. The 2-deoxyglucose uptake was assessed according to the manufacturer's instructions. Luminescence intensity was quantified using a SpectraMax M5 (Molecular Devices). Values were normalized to the DNA content of an equally seeded and treated copy plate quantified using the Quant-iT PicoGreen dsDNA Assay Kit (Thermo Fisher Scientific) as described above for the Respiratory analysis.

### Pyruvate dehydrogenase activity assay

At 80% confluency, hNPCs were passaged, cell number was determined, and 1,000,000 cells per sample were resuspended in 100 μl ice-cold assay buffer. The PDH activity was assessed using the pyruvate dehydrogenase Activity Assay Kit (MAK183, Sigma) according to the manufacturer's instructions. The absorbance at 450 nm was quantified every 5 min for 30 min using a SpectraMax M5 (Molecular Devices). Values were normalized to genomic DNA content analyzed using additional 1,000,000 cells from the same cell solution. Cells were lysed in 100 μl RIPA buffer and genomic DNA was quantified using the Quant-iT PicoGreen dsDNA Assay Kit (Thermo Fisher Scientific) as described above for Respiratory analysis.

### α-ketoglutarate dehydrogenase activity assay

At 80% confluency, hNPCs were passaged, cell number was determined, and 1,000,000 cells per sample were resuspended in 100 μl ice-cold assay buffer. The OGDHC activity was assessed using the alpha-Ketoglutarate Dehydrogenase Activity Assay Kit (Colorimetric) (ab185440, Abcam) according to the manufacturer's instructions. The absorbance at 450 nm was quantified every 5 min for 30 min using a SpectraMax M5 (Molecular Devices). Values were normalized to genomic DNA content analyzed using additional 1,000,000 cells from the same cell solution. Cells were lysed in 100 μl RIPA buffer and genomic DNA was quantified using the Quant-iT PicoGreen dsDNA Assay Kit (Thermo Fisher Scientific) as described above for Respiratory analysis.

### Quantification of human homocysteine levels

At 80% confluency, hNPCs were rinsed with PBS, collected using a cell scraper in 1 ml PBS, homogenized, and stored overnight at −20 °C. After two freeze-thaw cycles, the homogenates were centrifuged for 5 minutes at 5000×g, 4 °C. The supernatant was collected and the protein content was quantified using the Pierce BCA Protein Assay Kit (Thermo Fisher Scientific) according to the manufacturer's

instructions. Samples were diluted to obtain a 30 µg protein/100 µl PBS solution which was assayed immediately using the human homocysteine (HCY) ELISA kit (CSB-E13814h, Cusabio) according to the manufacturer's instructions. The absorbance at 450 nm and 540 nm was quantified within 5 min using a SpectraMax M5 (Molecular Devices). Absorbance at 540 nm was subtracted from the absorbance at 450 nm, and a standard curve was used to calculate the homocysteine concentration in nmol/ml.

### Isolation of RNA, cDNA synthesis, and quantitative real-time PCR

Gene expression was analyzed by RT-qPCR as described previously[17]. hNPCs were maintained on Geltrex-coated six-well plates and in neural precursor maintenance medium at 37 °C, 7% $CO_2$, 21% $O_2$ for at least 48 h. Total RNA was extracted using RNeasy Plus Mini Kit (Qiagen) according to the manufacturer's instructions and reverse transcribed to cDNA using the SuperScript VILO cDNA Synthesis Kit (Thermo Fisher Scientific). For RT-qPCR, 25 ng (278 ng/µl) cDNA was quantitatively amplified on a QuantStudio 7 Flex (Thermo Fisher Scientific) using TaqMan universal PCR MM no Ung (Thermo Fisher Scientific) and gene-specific TaqMan primers (Thermo Fisher Scientific): ACTB (Hs99999903_m1); SLC2A1 (Hs00892681_m1); SLC2A2 (Hs01096908_m1); SLC2A3 (Hs00359840_m1); SLC2A4 (Hs00168966_m1). The comparative Ct method was used to analyze differences in gene expression, values were normalized to ACTB.

### Immunostaining

Cells were cultured on Geltrex-coated 96-well plates or on glass coverslips for at least 72 h. Cells were fixed with 10% Formalin for 20 min at 37 °C, washed twice with PBS, and permeabilized/blocked with PBS containing 1% BSA (Sigma-Aldrich) and 0.3% Triton X-100 (Sigma-Aldrich) for 15 min at room temperature. Primary antibodies were diluted in PBS containing 1% BSA and 0.3% Triton X-100 and antibody incubation was performed at 4 °C overnight. Cells were washed twice with PBS and incubated with secondary antibodies diluted in PBS containing 1% BSA and 0.3% Triton X-100 for 2 h at room temperature. Nuclei were stained using a 0.1 µg/ml DAPI (Sigma)−PBS solution for 10 min at room temperature. Cells were washed twice with PBS and coverslips were mounted using Aqua-Poly/Mount (Polysciences Inc.) or Aqua-Poly/Mount was added per well. Primary antibodies were diluted as follows: ATP5F1A (ab14748, Abcam; 1:500), NES (Ma1110, Thermo Fisher Scientific; 1:250), Histone H3 (tri methyl K27) (ab6002, Abcam; 1:100), Histone H3 (tri methyl K9) (ab8898, Abcam; 1:500), SLC2A1 (MA5-31960, Invitrogen; 1:500), SLC2A3 (PA5-72331, Thermo Fisher Scientific; 1:500), SOX1 (Ab87775, Abcam; 1:500), SOX2 (sc17320, Santa Cruz; 1:500), TUBB3 (T5076, Sigma-Aldrich; 1:1000). Secondary antibodies were diluted as follows: donkey-anti-mouse IgG Alexa 488 (A21202, Thermo Fisher Scientific; 1:500), donkey-anti-rabbit IgG Alexa 488 (A21206, Thermo Fisher Scientific; 1:500). If necessary, images were processed using ImageJ 1.53c.

### Image quantification

**Quantification of histone levels and hNPC markers.** For cells grown on 96-well plates, images were acquired using a Cellinsight NXT platform (Thermo Fisher Scientific) with a 20 × 0.4 NA objective (field size of 454.41 by 454.41 µm) and analyzed using HCS Studio 2.0 (Thermo Fisher Scientific). The acquisition was configured individually for every analyzed antibody and chromophore. Nuclear DNA fluorescence intensity (DAPI dye) was assessed in channel 1 and a nuclear mask was created using the image analysis segmentation algorithm to identify viable cells as valid objects according to their object area, shape, and intensity. The nuclear mask was used to quantify stained proteins of valid objects in channel 2 or 3 with a fixed exposure time. A ring (thickness 13 pixels) was used to quantify the cytoplasmic markers with a fixed exposure time. The amount of analyzed valid objects per well

and cell line used for statistical analysis is stated in the corresponding figure legends. Violin plots were generated using the functions ggplot + geom_violin of the R package ggplot2 v3.4.2. For the statistical analysis, a linear mixed effects model (lm) was fit using the lmer function (R package lme4 Version 1.1-34), where unique cells were included but nested within donors (formula: Parameter measured per cell -disease state + 1 | disease state:Patients; REML = FALSE). *p*-values for lm were calculated using the Anova function (R package car Version 3.1-2).

**Quantification of mitochondrial abundance and morphology.** hNPCs were maintained on Geltrex-coated glass coverslips and in neural precursor maintenance medium at 37 °C, 7% $CO_2$, 21% $O_2$ for at least 48 h. The same amount of prewarmed neural precursor medium supplemented with 200 nM MitoTracker Deep Red (M22426, Thermo Fisher Scientific) was added per well and cells were stained for 20 min at 37 °C, 7% $CO_2$, 21% $O_2$. Cells were washed twice with PBS and fixed with 10% formalin (Sigma) for 20 min at 37 °C. ATP5F1A, TUBB3, and DAPI staining was performed as described above.

Images were taken using a Leica SP8 laser scanning confocal microscope equipped with 488, 561 and 633 nm lasers and a 63× objective. Images were processed using CellProfiler v4.2.5 (Broad Institute). Multiple images per well/cell line were loaded and assigned to the groups Patient and Disease state. Images were processed as described in ref. 70. In brief, fluorescence intensities of all channels were rescaled, illumination was corrected, primary and secondary (based on "Distance − B" or "Propagation") objects were identified, objects were filtered (based on shape and size), and tertiary objects were assigned. Mitochondrial speckles (MitoTracker or ATP5F1A staining) were enhanced and quantified as primary objects within the tertiary cell objects (based on 'Distance – B' or "Propagation"). A typical diameter of 3−300 pixel units was assumed for mitochondrial content and a threshold strategy with a threshold smoothing scale of 0, and a threshold correction factor of 2 was applied, as well as an object smoothing filter of 20. Next, the shape, area, and intensity of mitochondrial primary objects were quantified, and mitochondrial objects were related to tertiary cell objects. Additionally, the mitochondrial skeleton was quantified. For this purpose, the mitochondrial primary objects were converted to an image, and the morphological skeleton was identified. The lengths and branch points of the mitochondrial skeletons were quantified. Violin plots were generated using the functions ggplot + geom_violin of the R package ggplot2 v3.4.2. For the statistical analysis, a linear mixed effects model (lm) was fit using the lmer function (R package lme4 Version 1.1-34), where unique cells were included but nested within donors (formula: Parameter measured per cell -disease state + 1 | disease state:Patients; REML = FALSE). *p*-values for lm were calculated using the Anova function (R package car Version 3.1-2).

**Quantification of glucose transporters.** For cells grown on 96-well plates, images were acquired as described for the quantification of histone marks. Acquired images were exported from HCS Studio 2.0 and analyzed using CellProfiler v4.2.5 (Broad Institute). Multiple images per well/cell line were loaded and assigned to the groups Patient and Disease state. Images were processed as described for the quantification of mitochondrial abundance and morphology. The distribution of glucose transporters/fluorescence intensity within the identified tertiary objects (based on "Propagation") was measured using MeasureObjectIntensityDistribution in ten bins around the center of the corresponding primary object with bin 1 being close to the center and bin 10 close to the edges. Violin plots were generated using the functions ggplot + geom_violin of the R package ggplot2 v3.4.2. For the statistical analysis, a linear mixed effects model (lm) was fit using the lmer function (R package lme4 Version 1.1-34), where unique cells were included but nested within donors (formula: Parameter measured per cell -disease state + 1 | disease state:Patients;

REML = FALSE). *p*-values for lm were calculated using the Anova function (R package car Version 3.1-2).

## Western blot

hNPCs were maintained on Geltrex coating and in neural precursor maintenance medium at 37 °C, 7% CO$_2$, 21% O$_2$ for at least 72 h. Approximately $2 \times 10^7$ cells were lysed in RIPA buffer and subsequently stored at −80 °C. Protein concentration was quantified using the Pierce BCA Protein Assay Kit (Thermo Fisher Scientific) according to the manufacturer's instructions. 10 µg protein extract was diluted in RIPA and NuPAGE (Novex) and incubated for 5 min at 95 °C. Protein extracts from 5 Ctrl and 7 sPD hNPC lines and Protein Marker VI (AppliChem) were separated on a Criterion XT Bis-Tris Gel, 4–12% (Bio-Rad) using Tris/Glycine Buffer (Bio-Rad) and a Criterion Vertical Electrophoresis Cell (Bio-Rad) at 120 V for 70 min. Proteins were blotted on methanol-activated Immobilon−P Membranes (Millipore) using XT MOPS buffer (Bio-Rad) and a Criterion Blotter (Bio-Rad) at 20 V, 4 °C overnight. Membranes were blocked for 1 h with TBS containing 0.01% Tween (TBST) and 5% milk. Primary antibodies NDUFB8 (459210, Novex; 1:500), Complex II- Subunit 30 (SDHB) (459230, Thermo Fisher Scientific; 1:500), UQCRC2 (ab14745, Abcam; 1:2500), MT-CO2 (ab110258, Abcam; 1:1000), ATP5F1A (ab14748, Abcam; 1:4000), DRP1 (5391, Cell Signaling; 1:1000), DRP1 phospho-Ser616 (4494, Cell Signaling; 1:1000), TUBA (GTX628802, Genetex; 1:20,000), ACTB (ABO145-200, OriGene; 1:2000), OGDHL (17110-1-AP, Proteintech, 1:5000), GAPDH (GTX627408 peroxidase coupled, Genetex, 1:20,000), SLC2A1 (MA5-31960, Invitrogen, 1:5000), SLC2A3 (PA5-72331, Thermo Fisher Scientific, 1:5000) were incubated in blocking buffer overnight at 4 °C. Membrane was washed with TBST and secondary antibodies rabbit-anti-mouse IgG peroxidase (GTX213112-01, GeneTex; 1:10,000), goat-anti-rabbit IgG peroxidase (111-035-003, Dianova; 1:10,000), and rabbit-anti-goat IgG peroxidase (305-035-003, Dianova; 1:10,000) were incubated in blocking buffer at room temperature for 2 h, respectively. Subsequently, the membrane was washed with TBST and incubated with ECL substrate (GE Healthcare) for 1 min. Protein bands were visualized using a ChemiDoc Imager (Bio-Rad) and quantified using Image Lab (Bio-Rad). Quantified expression levels were normalized to ACTB or TUBA levels.

## Analysis of complex I activity

Complex I activity was quantified using the complex I enzymatic activity microplate assay kit (Abcam, ab109721) according to the manufacturer's instructions and as described previously[17]. Kinetics were measured on a SpectraMax M5 (Molecular Devices) for 45 min. Values were normalized to total protein concentrations and average Ctrl levels.

## Proteome analysis

**Sample preparation for total hNPC proteome analysis.** hNPCs were maintained on Geltrex coating and in neural precursor maintenance medium at 37 °C, 7% CO$_2$, 21% O$_2$ for at least 72 h. At 80% confluency, hNPCs were passaged using Accutase, cell number was determined, and 3,000,000 cells per replicate with three replicates per cell clone were collected. Cells were harvested at $200 \times g$ for 5 min, medium was aspirated and cell pellets were snap frozen in liquid nitrogen and stored at −80 °C. Frozen cell pellets were lysed in SDC Buffer (1% sodium deoxycholate (wt/vol) in 100 mM Tris pH 8.5) and boiled for 5 min at 95 °C. Lysates were then cooled on ice for 5 min and sonicated using the Bioruptor sonication device for 30 min. Reduction and alkylation was performed by adding Tris(2-carboxyethyl)phosphine (TCEP) and 2-Chloracetamide (CAA) at the final concentrations of 10 mM and 40 mM, respectively, and incubating them for 5 min at 45 °C. Samples were digested overnight by the addition of 1:50 LysC (1:50 wt/wt: Wako) and Trypsin (1:50 wt/wt: Sigma-Aldrich) overnight at 37 °C with agitation (1500 rpm) on an Eppendorf Thermomixer C.

The next day, peptides were desalted using SDB-RPS (Empore) Stage-Tips. Briefly, samples were tenfold diluted using 1% TFA in isopropanol and then loaded onto the StageTips, which were subsequently washed with 200 µL of 1% TFA in isopropanol and then with 0.2% TFA/2% acetonitrile (ACN) twice. Peptides were eluted using 75 µL of 80% ACN/1.25% NH$_4$OH and dried using a SpeedVac centrifuge (Concentrator Plus; Eppendorf) for 1 h at 30 °C. Peptides were resuspended in 0.2% TFA/2% ACN and peptide concentration was determined using the Nanodrop 2000 (Thermo Scientific). In total, 200 ng of peptides were subjected to LC-MS/MS analysis.

**Data-independent acquisition LC-MS analysis for total hNPC proteome.** Peptides were loaded on a 50 cm reversed-phase column (75 µm inner diameter, packed in-house with ReproSil-Pur C18-AQ 1.9-µm resin). No trap column was used. To maintain a column temperature of 60 °C, we used a homemade column oven. An EASY-nLC 1200 system (Thermo Fisher Scientific) was connected online with a mass spectrometer (Orbitrap Exploris 480, Thermo Fisher Scientific) via nano-electrospray source. Peptides were separated using a binary buffer system consisting of buffer A (0.1% formic acid (FA)) and buffer B (80% ACN, 0.1% FA). We used a constant flow rate of 300 nl/min. We loaded 200 ng of peptides and eluted them with a 100 min gradient. The gradient starts with 5% buffer B and increases consistently to 30% in 80 min, until it reaches 95% in 88 min and remains constant for another 4 min. At the end, Buffer B decreases to 5% in 96 min and remains constant for another 4 min. The MS data was acquired using a data-independent acquisition (DIA) mode with a full scan range of 300−1650 *m/z* at 120,000 resolution, automatic gain control (AGC) of 3e6 and a maximum injection time of 60 ms. The stepped higher-energy collision dissociation (HCD) was set to 25.5, 27.30. Each full scan was followed by 33 DIA scans which were performed at a 30,000 resolution, an AGC of 1e6 and the maximum injection time set to auto. Information regarding m/z separation and number of windows are provided as Supplementary Data 3.

**Data processing and bioinformatic analysis of total hNPC proteome.** DIA raw files were analyzed using directDIA in Spectronaut version 15 (Biognosys). The search was done against UniProt human proteome of canonical and isoform sequences (downloaded July 2019) with 20,383 entries for final protein identification and quantification. Enzyme specificity was set to trypsin with up to two missed cleavages. Maximum and minimum peptide length was set to 52 and 7, respectively. The search included carbamidomethylation as a fixed modification and oxidation of methionine and N-terminal acetylation of proteins as variable modifications. A protein and precursor FDR of 1% were used for filtering and subsequent reporting in samples (*q*-value mode with no imputation).

For bioinformatic analyses, intensities were log2-transformed. Next, the dataset was filtered by a minimum of three valid values in at least one experimental group and subsequently imputed using a Gaussian normal distribution (width = 0.3 and downshift = 1.8) using Perseus version 1.6.1.3[71]. Further analysis was performed using R v4.1.0. PCA was performed using the prcomp function and visualized using the package ggbiplot v0.55 with probability ellipses (0.68 of normal probability). The Pearson correlation between samples was calculated using the cor function within R, and correlations were plotted in a heatmap using the pheatmap function of the R package pheatmap v1.0.12. Agglomerative hierarchical clustering by the hclust function (method = complete) was applied to group samples.

Hypothesis testing was performed using the package DESeq2 v1.36.0 and the design formula design = ~samples.n + samples.n:replicate + condition with condition being either Ctrl or sPD, samples.n being a unique number per patient-derived cell line (1 to 12), and replicate indicating the different replicates per cell line (1, 2, or 3). *P* values were adjusted for multiple testing within DESeq2 by Benjamini

and Hochberg[72]. Proteins with a p.adjust-value (*q*-value) <0.05 were considered significantly altered in sPD.

Based on the complete set of DEPs a volcano plot and heatmap was produced. For the volcano plot, log2(fold change) was plotted versus the -log10(p.adjust-value) on the *x*- and *y*-axis, respectively. The Volcano plot was generated using the EnhancedVolcano function of the R package EnhancedVolcano v1.12.0 with pCutoff = 0.05 and FCcutoff=0.26. The heatmap was generated by using the heatmap.2 function within the gplots v3.1.1 package. Agglomerative hierarchical clustering by the hclust function (method = complete) was applied to group samples or proteins. Log2(fold changes) of DEPs were scaled and represented as z-score.

### Transcriptome–Proteome correlation

Bulk-like DEG[17] and DEP lists were filtered for common genes/proteins yielding a list of 1250 DEG-DEP pairs. This list was further subset according to pairs exhibiting a positive correlation (downregulate DEG and downregulated DEP - 553 pairs; upregulated DEG and upregulated DEP - 33 pairs) or a negative correlation (downregulate DEG and upregulated DEP - 582 pairs; upregulated DEG and downregulated DEP - 82 pairs). The correlation between DEGs and DEPs was assessed using Pearson's correlation within the base R function cor.test. Plots were generated using ggplot2 v3.4.0 and trend lines were added using geom_smooth(method = lm).

### Non-targeted metabolomics analysis

**Cell preparation.** hNPCs were seeded at a density of 700,000 cells/well in six replicates in Geltrex-coated six-well plates and cultured in neural precursor maintenance medium with daily medium changes at 37 °C, 7% $CO_2$, 21% $O_2$ for at least 72 h to achieve a desired cell number of 1,000,000–1,500,000 cells per well. One replicate was used to determine the cell number. hNPCs were dissociated using Accutase for 10 min at 37 °C, 7% $CO_2$, 21% $O_2$, chopped with a 1000 μl pipette tip, and diluted in 5 ml neural precursor maintenance medium. Cell number was determined using a Neubauer-improved cell counting chamber. The remaining replicates were used for metabolite extraction. Cells were washed twice with warm PBS and metabolism was quenched by adding precooled extraction solvent (80% methanol (AppliChem)) which contained 4 standard compounds to monitor the extraction efficiency. The amount of extraction solvent used was adjusted to the cell count, 1 ml solvent was used for 1,000,000 cells. Cells were harvested by scraping them with the extraction solvent, were collected in precooled microtubes (Sarstedt), and stored at −80 °C.

For Ctrl samples, Geltrex-coated six-well plates containing neural precursor maintenance medium were incubated at 37 °C, 7% $CO_2$, 21% $O_2$ for at least 72 h. Medium was changed daily. After 72 h, medium was collected in precooled microtubes and stored at −80 °C. Geltrex-coated wells were washed twice with warm PBS and 1 ml extraction solvent was added. After scraping, the Geltrex–extraction solvent mixture was collected in precooled microtubes and stored at −80 °C.

For analysis, 80 mg glass beads (0.5 mm; VK-05, PeqLab) were added to the samples and cells were homogenized using a Preccellys24 (PeqLab) for two times 25 s at 5500 rpm, 4 °C. The homogenates were used for fluorometric DNA quantification and for metabolomic analysis.

**Fluorometric DNA quantification.** Fluorometric DNA quantification in homogenates was performed as described by[73]. In brief, 80 μl Hoechst 33342 (20 μg/ml in PBS; Thermo Fisher Scientific) were mixed with 20 μl of the homogenate or 88% Methanol (blank) in a black 96-well plate (Thermo Fisher Scientific) and incubated in the dark for 30 min at room temperature. Fluorescence intensity was quantified using a GloMax Multi Detection System (Promega) with an UV filter ($\lambda_{EX}$ 365 nm, $\lambda_{Em}$ 410–460 nm; Promega). Values were normalized to blank

levels and the average of 4 replicates per sample was used for normalization of metabolite levels.

**LC-MS/MS-based metabolomics analysis.** The cell homogenates were centrifuged for 5 min at 3250× *g*, 4 °C. Eight aliquots of 50 μl of the supernatant were loaded onto four 96-well microplates. Two (i.e. early and late eluting compounds) aliquots for analysis by ultra-high performance liquid chromatography-tandem mass spectrometry (UPLC-MS/MS) in positive ion mode electrospray ionization (ESI), one for analysis by UPLC-MS/MS in negative ion mode ESI, and one for analysis by (HILIC)/UPLC-MS/MS in negative ion mode ESI. Three types of quality control samples were included into each plate: samples generated from a pool of human plasma, samples generated from a small portion of each experimental samples served as technical replicate throughout the dataset, and extracted water samples served as process blanks. Experimental samples and controls were randomized across the metabolomics analysis. The samples were dried on a TurboVap 96 (Zymark).

Prior to UPLC-MS/MS analysis, the dried samples were reconstituted in acidic or basic LC-compatible solvents, each of which contained a series of standard compounds at fixed concentrations to ensure injection and chromatographic consistency. The UPLC-MS/MS platform utilized a Waters Acquity UPLC with Waters UPLC BEH C18-2.1 × 100 mm, 1.7-μm columns and a Thermo Scientific Q-Exactive high resolution/accurate mass spectrometer interfaced with a heated electrospray ionization (HESI-II) source and Orbitrap mass analyzer operated at 35,000 mass resolution. One aliquot reconstituted in acidic positive ion conditions, chromatographically optimized for more hydrophilic compounds. In this method, the extract was gradient eluted from a C18 column (Waters UPLC BEH C18-2.1 × 100 mm, 1.7 μm) using water and methanol, containing 0.05% perfluoropentanoic acid (PFPA) and 0.1% formic acid (FA). Another aliquot was also analyzed using acidic positive ion conditions; however it was chromatographically optimized for more hydrophobic compounds. In this method, the extract was gradient eluted from the same aforementioned C18 column using methanol, acetonitrile, water, 0.05% PFPA and 0.01% FA and was operated at an overall higher organic content. Another aliquot was analyzed using basic negative ion optimized conditions using a separate dedicated C18 column. The basic extracts were gradient eluted from the column using methanol and water, however with 6.5 mM Ammonium Bicarbonate at pH 8. The fourth aliquot was analyzed via negative ionization following elution from a HILIC column (Waters UPLC BEH Amide 2.1 × 150 mm, 1.7 μm) using a gradient consisting of water and acetonitrile with 10 mM Ammonium Formate, pH 10.8. The MS analysis alternated between MS and data-dependent MSn scans using dynamic exclusion. The scan range varied slighted between methods but covered 70–1000 m/z.

Raw data was extracted, peak-identified and QC processed using Metabolon's hardware and software (Metabolon, Inc., North Carolina, USA). Compounds were identified by comparison to library entries of purified standards or recurrent unknown entities, based on three criteria: retention index within a narrow RI window of the proposed identification, accurate mass match to the library +/− 10 ppm, and the MS/MS forward and reverse scores between the experimental data and authentic standards. The MS/MS scores are based on a comparison of the ions present in the experimental spectrum to the ions present in the library spectrum. While there may be similarities between these molecules based on one of these factors, the use of all three data points can be utilized to distinguish and differentiate biochemicals.

**Data analysis.** OrigScale values were median normalized, adjusted to DNA levels and imported into R v4.1.0. Only metabolites detected in more than 30% of the samples were used for further analysis. To approximate normal distribution, values were log2-transformed.

Unsupervised PCA was used to discover differential variation features and confirmed close relationship between replicates of samples. PCA was performed using the prcomp function in R and visualized using the R package ggbiplot v0.55 with probability ellipses (0.68 of normal probability). Thus, average values of all replicates per sample and metabolite were used for statistical analysis. Hypothesis testing was performed using Student's *t*-test (t.test function in R) and *p*-values were adjusted for multiple testing using the R package fdrtool v1.2.15 (statistic = pvalue, cutoff.method = ptc0). Metabolites with a *q*-value < 0.05 were considered significantly altered in sPD. A volcano plot was produced using the R package EnhancedVolcano v1.12.0 by plotting the log2(fold change) versus the −log10(*p*-value).

### Isotopic labeling

**Cell preparation.** hNPCs were seeded on Geltrex-coated 6-well plates containing neural precursor maintenance medium at a density of 1,000,000 cells/well with six replicates per cell line. hNPCs were cultured at 37 °C, 7% $CO_2$, 21% $O_2$ for at least 72 h. Medium was replaced by a labeling medium (50% DMEM/F12-GlutaMAX (L0091500, Biowest), 50% Neurobasal (A2477501, Life Technologies) supplemented with 1% B27 (minus vitamin A, 12587010, Life Technologies), 0.5% N2 (Life Technologies), 10 μM SB431542, 0.5 μM purmorphamine, 1 μM dorsomorphin, 4.44 nM FGF-8b, 150 μM ascorbic acid 2-phosphate, 0.02% beta-mercaptoethanol, 0.125 mM sodium pyruvate (Sigma), 21.25 mM glucose (Sigma), and 1.25 mM L-glutamine (Sigma)) containing the respective stable-isotope tracer instead of its unlabeled variant. Cells were cultured with either 1.25 mM [U-13C]-glutamine or 21.25 mM [U-13C]-glucose (all tracers: Cambride Isotope Laboratories, USA) for 24 h. Subsequently, cell culture supernatant was stored for profiling the extracellular metabolome. Three replicates per cell line and blanks were washed with 0.9% NaCl and quenched with ice-cold methanol and ice-cold ddH$_2$O (containing 1 μg/ml D6-glutaric acid as internal standard). Cells were scraped and extracts were added into tubes containing ice-cold chloroform. Following vortexing at 1400 rpm for 20 min at 4 °C and centrifugation at 17,000× *g* for 5 min at 4 °C, 300 μl of the polar phase were transferred into GC glass vials with microinsert and dried under vacuum at 4 °C.

**Fluorometric DNA quantification.** The other three replicates per cell line were used for the quantification of genomic DNA using the Quant-iT PicoGreen dsDNA Assay Kit (Thermo Fisher Scientific). Cells were lysed in 200 μl RIPA buffer (50 mM Tris-HCL (Sigma), 150 mM NaCl (Sigma), 1% Triton X-100, 0.5% sodium deoxycholate (Sigma), 0.1% SDS (Sigma), 3 mM EDTA (Sigma)) per well. The Quant-iT PicoGreen dsDNA assay was performed according to the manufacturer's instructions. DNA concentrations were calculated with a linear regression curve using Lambda DNA standards. Average values per replicates were calculated and used for normalization.

**GC-MS-based metabolomics analysis.** GC-MS measurement of isotopic enrichment and relative metabolite abundance was performed as previously published[74]. Briefly, dried extracts were derivatized using equal amounts of methoxyamine (20 mg/ml in pyridine) and MTBSTFA and injected into an Agilent 7890B gas chromatograph equipped with a 30 m DB-35 ms and 5 m Duruguard capillary column. Metabolites were detected in selected ion mode by an Agilent 5977 MSD system. The MetaboliteDetector software was used to analyze chromatograms, calculate mass isotopomer distributions (MIDs) and perform relative comparison of metabolite levels[75].

**Quantification of medium concentrations.** Concentrations of glucose, lactate, glutamine, and glutamate in cell culture supernatants and control/blank media were determined using a YSI 2950D Biochemistry Analyzer. Quantification was performed by measuring respective reference compounds. Uptake rates were calculated by subtracting sample concentrations from the measured blank concentrations. Only for blank glutamine, the medium glutamine concentration of 1.25 mM was assumed, as the measurements were erroneous. Uptake rates were normalized to the genomic DNA content measured per sample.

**Data analysis.** Selected MIDs, as well as uptake rates, and total cellular metabolite concentrations normalized to the genomic DNA content per sample were used for statistical analysis in R v4.1.0. Hypothesis testing was performed for selected MIDs using the package lme4 v1.1-31 and the formula values ~condition + (1 | condition:replicate) with condition being either Ctrl or sPD and replicate indicating the different replicates per patient-derived cell line (1, 2, or 3). *p*-values were calculated using the Anova function (R package car Version 3.0-10). Selected MIDs, uptake rates, or cellular metabolites with a *p*-value < 0.05 were considered significantly altered in sPD. The fractional contribution of glucose- or glutamine-derived carbons to the total abundance of metabolite carbon was calculated by dividing the sum of MID fractions (except M0) by the number of metabolite carbons.

### Growth curve

hNPCs were seeded on Geltrex-coated 24-well plates containing neural precursor maintenance medium at a density of 100,000 cells/well with 15 replicates per cell line. hNPCs were cultured at 37 °C, 7% $CO_2$, 21% $O_2$ for at least 5 days with daily medium changes. Each day, three replicates per cell line were used for genomic DNA preparation. Medium was aspirated, cells were washed with PBS once and lysed in 100 μl RIPA buffer. Genomic DNA was quantified using the Quant-iT PicoGreen dsDNA Assay Kit (Thermo Fisher Scientific) according to the manufacturer's instructions. DNA concentrations were calculated with a linear regression curve using Lambda DNA standards. If assuming an exponential cell growth, a growth rate of 0.001312/h was determined. Both Ctrl and sPD hNPCs exhibited a comparable proliferation rate[17].

### Metabolic flux analysis

Estimation of metabolic fluxes was performed using isotopic nonstationary $^{13}$C-metabolic flux analysis (MFA) based on a metabolic network including the main pathways of the central carbon metabolism—glycolysis, citric acid cycle, and amino acid metabolism. The network reactions and carbon transitions are listed in Supplementary Data 10. To estimate the metabolic fluxes, corrected MIDs calculated from [U-13C]-glutamine and [U-13C]-glucose labeling experiments, as well as the extracellular exchange rates (for glucose, lactate, glutamine, and glutamate), and the effluxes for biomass formation (estimated from the cellular growth rates as described in ref. 76 were integrated into the metabolic network. Most extracellular exchange rates were considered reversible to allow balancing extracellular pools, only uptake rates for essential amino acids and glucose were considered irreversible.

Non-stationary $^{13}$C-MFA was performed using INCA v2.1[47,77] within MATLAB R2018a (installed packages: Statistics and Optimization Toolbox) as described in ref. 78. In brief, INCA estimated pathway fluxes using the elementary metabolite unit method. Fluxes were estimated by minimizing the variance-weighted sum of squared residuals between experimental and simulated data by least-squares regression. To find the global optimum, model prediction was repeated for at least 5 times. The predictions were subjected to qui-square statistical testing to evaluate the goodness-of-fit. Parameter continuation was used to calculate the 95% confidence intervals. A redundancy analysis was automatically performed by INCA. Data overfitting was avoided by limiting the metabolic network to experimentally measured reactions/metabolites, thus to the minimum of reactions necessary to accurately simulate measured cellular carbon flows.

## Pathway enrichment analysis

DEGs/DEPs were investigated for enrichment in KEGG, KEGG-Module and WikiPathway (WP) terms using the enrichKEGG, enrichMKEGG, or enrichWP function of the R package clusterProfiler v4.2.2 (one-sided hypergeometric test). Enrichment of Reactome terms was assessed using the enrichPathway function of the R package ReactomePA v1.38.0 (one-sided hypergeometric test). $p$-values were adjusted for multiple testing by Benjamini and Hochberg. Enrichment maps were generated using the treeplot function of clusterProfiler v4.2.2.

Integrated pathway enrichment analysis of previous transcriptome (bulk-like DEGs)[16] or present proteome data (DEPs) and present metabolome (metabolites with $q$-value < 0.05) data was performed in MetaboAnalyst v5.0 [https://www.metaboanalyst.ca/MetaboAnalyst/home.xhtml] (Enrichment analysis = Hypergeometric Test, Topology measure = Degree Centrality, Integration method = Combine queries) using KEGG (Oct2019) metabolic pathways.

The 'citric acid cycle' pathway and the 'Electron Transport Chain (OXPHOS system in mitochondria)' pathway (source: WikiPathways) were visualized and annotated using cytoscape v3.9.1 (installed apps: WikiPathways v3.3.10, CyTargetLinker v4.1.0, stringApp v1.7.0, and enhancedGraphics v1.5.4) and the R package RCy3 v2.14.2.

## Prediction of transcription factor binding sites

The human OGDHL promoter region (GXP_638603) was obtained from Genomatix by using the ElDorado database version ElDorado 04-2021 (genome build GRCh38). Transcription factor binding sites for GLI2 (MA0734.1; MA0734.2), GLI3 (MA1491.1; MA1491.2), and FOXA2 (MA0047.3) within this promoter region and with a minimal relative profile score of 80% were predicted using JASPAR, release 9 (2022). The human OGDHL locus (genome build GRCh38) was visualized using the UCSC genome browser. The alignment between the human and mouse (genome build GRCm38/mm10) genome was obtained from the UCSC genome browser track Vertebrate Multiz Alignment & Conservation (downloaded 11.2022).

## Correlation of candidate experimental variables with patient-derived clinical parameters

Key experimental variables per patient- or Ctrl-derived cell lines as well as clinical information related to disease progression were collected. This yielded a dataset of 77 parameters describing the patients or Ctrl individuals. As no information about disease progression can be available for Ctrl individuals, the dataset was also separated based on the disease state. The Pearson correlation between parameters for the whole dataset or the sPD dataset was performed using the function rcorr of the R package Hmisc v5.1-0. A correlogram showing the Pearson correlation coefficients was generated using the function corrplot of the R package corrplot v0.92. Parameters were ordered based on hierarchical clustering. Scatter plots with linear regression lines and 95% confidence intervals were generated using the function ggscatter of the R package ggpubr v0.6.0. Pearson correlation coefficients with $p$-values were added using the function stat_cor. Parameter pairs with $p$-values < 0.1 were considered as significant correlations.

## Multiple-factor analysis

In total 5 groups of variables were used for the multiple-factor analysis: the top 100 DEGs, the top 100 DEPs, the significantly altered metabolites (45 metabolites), as well as the experimental variables used for the correlation analysis (61 functional parameters), and as a supplementary group, the information about the disease state (Ctrl or sPD). The multiple-factor analysis in the sense of Escofier-Pages with a supplementary group of variables was performed using the function MFA of the R package FactoMineR v2.8. The multiple-factor analysis was visualized with confidence ellipses using the function fviz_mfa_ind of the R package factoextra v1.0.7. The contribution of groups to

dimensions was visualized using the function fviz_contrib of the R package factoextra v1.0.7.

## Statistics and reproducibility

No statistical methods were used to predetermine the sample size. If not stated otherwise, every analysis (except the omics analysis) was performed thrice using independently collected material from 5 Ctrl and 7 sPD patient-derived cell clones e.g. from three different passages (fibroblasts and hiPSCs) or three independent differentiation approaches (hNPCs and DANs). Each data point represents the average of these three replicates for each individual. If two clones per patient were used for experiments, first the average value of these clones per individual and replicate was calculated. Average values per patients were used for any further statistical analysis performed using GraphPad Prism 6. For two-group comparisons and in case of normal distribution, unpaired, two-tailed $t$-test was applied. In the case of non-Gaussian distribution, two-tailed Mann–Whitney $U$ test was applied. For the comparison of multiple groups, one-way ANOVA with Sidak's multiple comparison test was performed. Data displaying a measurement progression are shown with mean ± standard error of the mean (SEM). Boxplots are displayed from min to max values with all data points shown.

If not stated otherwise, R version 4.2.2 and RStudio 2022.12.0 Build 353 was used for further analysis. Violin plots were used to visualize the distributions of repeated measurements per individual cell line (e.g. if multiple cells per cell line were analyzed). Violin plots were generated using the functions ggplot + geom_violin of the R package ggplot2 v3.4.2. For the statistical analysis, a linear mixed effects model (lm) was fit using the lmer function (R package lme4 Version 1.1-31), where unique cells were included but nested within donors (formula: Parameter measured per cell ~disease state + 1 | disease state:Patients; REML = FALSE). $p$-values for lm were calculated using the Anova function (R package car Version 3.0-10). Boxplots summarizing measurements per individual cell line were generated using the functions ggplot + geom_boxplot of the R package ggplot2 v3.4.2. These boxplots display the median and range from the 25th to 75th percentile. Whiskers extend to the most extreme data point which is no more than 1.5 times the interquartile range.

If not stated otherwise, $p$-values or $q$-values below 0.05 were considered significant. Differences with a $p$-value or $q$-value between 0.05 and 0.1 were considered as trends (#). Not-significant differences are not indicated.

## Reporting summary

Further information on research design is available in the Nature Portfolio Reporting Summary linked to this article.

# Data availability

All data produced in this study are archived internally. Further information and requests for resources and reagents should be directed to and will be fulfilled by the corresponding authors. The mass spectrometry proteomics data have been deposited to the ProteomeXchange Consortium via the PRIDE partner repository with the dataset identifier PXD038399. [13]C labeling data have been deposited in the Metabolomics Workbench under project ID PR001536 and study ID ST002389[79]. The following databases were used within this study: KEGG pathways [https://www.genome.jp/kegg/pathway.html] (downloaded on 11.2022), WikiPathway pathways [https://www.wikipathways.org/index.php/WikiPathways] (downloaded on 11.2022), Reactome pathways [https://reactome.org/] (downloaded on 11.2022), Genomatix ElDorado database version 04-2021 (genome build GRCh38) [https://mygga.genomatix.de/online_help//help_eldorado/GenomeBrowser.html] (downloaded on 04.2021), UCSC genome browser [https://genome.ucsc.edu/] (accessed 11.2022), KEGG pathways used with MetaboAnalyst v5.0 [https://www.metaboanalyst.ca/MetaboAnalyst/home.xhtml] (downloaded 10.2019). Source data are provided with this paper.

## Code availability

Analysis code and data are freely available at Github [https://github.com/sebischmidt/A-reversible-state-of-hypometabolism-in-a-cellular-model-of-sporadic-Parkinson-s-disease] and have been deposited in Zenodo[80].

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

## Acknowledgements
We thank Franz Marxreiter (University Erlangen) for providing baseline clinical information; Annerose Kurz-Drexler, Tanja Orschmann, Susanne Badeke, Esther Álvarez Sánchez, Jessica Häußler, and Laura Snaidr for excellent technical assistance; Alessandra Moretti (TUM), Antje Gertes (HZM), Günter Höglinger (LMU), and Heiko Lickert (HZM) for discussions and comments. This work was supported in part by the Bavarian Ministry of Science and the Arts in the framework of the ForIPS consortium (C.S., W.W., J.W., and B.W.), the ForInter consortium (B.W., J.W., and F.J.T.), by the German Federal Ministry of Education and Research (BMBF) through ACS_iIMMUNE (no. 01EO2105 to M.R.) and the Integrated Network MitoPD (Mitochondrial endophenotypes of Morbus Parkinson), under the auspices of the e:Med Programme (grant 031A430E to W.W.), through the Joint Project HIT-Tau (High Throughput Approaches for the Individualized Therapy of Tau-Related Diseases—TP2: Grant 01EK1605C (to W.W. and D.T.), by the AMPro project "Aging and Metabolic Programming", as well as by "ExNet-0041-Phase2-3 (SyNergy-HZM)" through the Initiative and Network Fund of the Helmholtz Association (to F.G. and W.W.), through the Niedersächsisches Vorab (grant VWZN3266 to K.H.), through the Deutsche Forschungsgemeinschaft (DFG, German

Research Foundation) project HI1400/3-1 (K.H.), CRU5024 WI 3567/4-1 (B.W.) and WI 1620/4-1 (J.W.). Open Access funding enabled and organized by Projekt DEAL.

## Author contributions

This study was designed by W.W., S.S., D.M.V.W., F.G., C.S., M.J., and G.G.W. and coordinated by S.S. and W.W. J.W. and B.W. provided the sPD hiPSC clones. J.W. and M.R. examined patients and provided data for disease progression. C.S. and S.S. performed the Seahorse XF analysis in hiPSCs. A.H., M.Z.N., S.K., K.H., and S.S. performed the 13C labeling experiments. A.H., and K.H. quantified metabolite levels in cell culture media. D.T.V., O.K., M.M., and S.S. performed the proteome analysis. A.A., J.A., and S.S. performed the nontargeted metabolomics analysis. L.S. and S.S. quantified histone modifications. Bioinformatic analysis was done by S.S. under the supervision of D.T. M.D.L. performed the computational analysis of the scRNA-seq under the supervision of F.J.T. The remaining experiments were performed by S.S. W.W., S.S., F.G., D.M.V.W., D.T., G.G.W., and S.H. wrote the manuscript.

## Competing interests

F.J.T. consults for Immunai Inc., Singularity Bio B.V., CytoReason Ltd, and Omniscope Ltd, and has ownership interest in Dermagnostix GmbH and Cellarity. The remaining authors declare no competing interests.

## Additional information

[1]Institute of Developmental Genetics, Helmholtz Zentrum München, Neuherberg, Germany. [2]Munich Institute of Biomedical Engineering, Department of Chemistry, Technical University of Munich, Munich, Germany. [3]Department for Proteomics and Signal Transduction, Max-Planck Institute of Biochemistry, Martinsried, Germany. [4]Department of Bioinformatics and Biochemistry and Braunschweig Integrated Center of Systems Biology (BRICS), Technische Universität Braunschweig, Braunschweig, Germany. [5]Department of Molecular Neurology, University Hospital Erlangen, Friedrich-Alexander-Universität Erlangen-Nürnberg (FAU), Erlangen, Germany. [6]Institute of Metabolism and Cell Death, Helmholtz Zentrum München, Neuherberg, Germany. [7]Research Unit Molecular Endocrinology and Metabolism, Helmholtz Zentrum München, Neuherberg, Germany. [8]Department of Stem Cell Biology, University Hospital Erlangen, Friedrich-Alexander-Universität Erlangen-Nürnberg (FAU), Erlangen, Germany. [9]Department of Molecular Biosciences, The Wenner-Gren Institute, Stockholm University, Stockholm, Sweden. [10]Institute of Computational Biology, Helmholtz Zentrum München, Neuherberg, Germany. [11]Department of Mathematics, Technische Universität München, Garching bei München, Germany. [12]Institute for Synthetic Biomedicine, Helmholtz Zentrum München, Neuherberg, Germany. [13]Institute of Experimental Genetics, Helmholtz Zentrum München, German Research Center for Environmental Health, Neuherberg, Germany. [14]Department of Biochemistry, Yong Loo Lin School of Medicine, National University of Singapore, Singapore, Singapore. [15]Institute of Biochemistry, Faculty of Medicine, University of Ljubljana, Ljubljana, Slovenia. [16]NNF Center for Protein Research, Faculty of Health Sciences, University of Copenhagen, Copenhagen, Denmark. [17]Chair of Developmental Genetics, Munich School of Life Sciences Weihenstephan, Technical University of Munich, Freising, Germany. [18]Munich Cluster of Systems Neurology (SyNergy), Munich, Germany. [19]German Center for Neurodegenerative Diseases (DZNE) site Munich, Munich, Germany. ✉e-mail: sebastian.schmidt@helmholtz-munich.de; wolfgang.wurst@helmholtz-munich.de

