## [Peer Review File · Nature Communications]

A reversible state of hypometabolism in a human cellular model of sporadic Parkinson's diseaseREVIEWER COMMENTS

Reviewer #1 (Remarks to the Author):

The manuscript by Schmidt et al. presents findings that support the role of hypometabolism in sporadic Parkinson's disease (sPD). The authors used human induced pluripotent stem cells (hiPSCs) from late-onset sPD patients and healthy individuals for disease modeling. They utilized a multilayered omics approach, including transcriptomics, proteomics, and metabolomics, to evaluate metabolic properties in sPD neuronal progenitor cells (NPCs). The authors suggest that mitochondrial dysfunction and alterations in glucose metabolism are prominent in sPD NPCs. Furthermore, they described alterations in primary cilia function, which may be connected to PD pathologies. In my opinion, this study is interesting and impactful but rather correlative and descriptive in nature. Although the molecular mechanistic analysis is not the strength of this manuscript, the large dataset provided by transcriptomics, proteomics, and metabolomics analysis would be valuable for the field, and the results presented have the potential to open up new avenues of research. This manuscript could be considered for publication in Nature Communications after addressing the following comments:

Figure 1: sPD hiPSCs display reduced maximal respiration. Technically speaking, maximal respiration cannot be lower than baseline respiration. This is usually an indication of reduced cell density and suboptimal FCCP concentration. This point should be addressed carefully by repeating these experiments with similar cell numbers across conditions. The addition of an ATP-linked respiration graph and the display of ECAR data would allow readers to interpret these results. It is well known that as progenitor cells develop into mature neurons, metabolism shifts from glycolysis to oxidative phosphorylation. In order to interpret the observed differences, especially in DANs, the authors should include images with differentiation markers to confirm that the control vs. sPD DANs populations are similar in terms of neuronal differentiation. For example, do these neurons fire action potentials? Do they display similar neuronal processes?

Figure 2: To evaluate mitochondrial content, the authors used mitotracker staining, which is sensitive to mitochondrial membrane potential. This staining will exclude a sub-population of "unhealthy" depolarized mitochondria. Either mitochondrial antibody staining or a combination of membrane potential sensitive and insensitive mitochondrial stains will be necessary to evaluate mitochondrial content accurately. High-resolution zoomed-in images of both control and sPD hNPCs should be included to interpret the results. It is also not clear what the n values represent. This analysis should be performed per cell, and the total cell number should be increased for accurate statistical analysis. Downregulation of mRNA levels in sPD cells but no change in mitochondrial protein level may indicate alterations in protein turnover rate, which is related to total mitochondrial use – also addressed for Figure 1. Thus, further analysis of the NPC stages should be analyzed in control vs. sPD.

Figures 3/4: The differences in transcriptome and proteome observations are valuable for the field, and very few studies address this. This adds strength to the manuscript. This reviewer acknowledges that while it may be challenging, validation of this data with DANs (in addition to NPSs) would really enhance the impact of this manuscript for the field. Considering the authors' previous publication (Ref 16), which indicates that mitochondrial dysfunction develops in sPD upon dopaminergic neuron differentiation, a detailed analysis of mitochondrial health will be important to explain citric acid cycle defects.

Figure 5: The most accurate method to assess glucose uptake would be to utilize genetically encoded glucose sensors. This is because the gene expression level of glucose transporters may not represent the protein level or total protein present on the plasma membrane responsible for glucose transport. The level of 2DG also depends on HK1 activity. Increased glucose lactate conversion may also indicate an enhanced glycolysis rate in sPD (as shown in Fig 5e and Supp Fig 3c), which is common when mitochondria are dysfunctional. Therefore, ECAR graphs in Figure 1 will be significant. Designing seahorse measurements to specifically evaluate glycolysis rate would also address this issue. To further elaborate the changes in glutamine metabolism, the author should conduct experiments in the presence of an MPC inhibitor as previously described by Divakaruni A. et al. in JCB 2017.

Figure 6: In the opinion of this reviewer, the connection between SHH signaling/PC and

hypometabolism is highly suggestive, but the data presented is still preliminary and represents the weakest point of this manuscript. This section could be removed (Fig 6a-b and Supp Fig 5 can be combined as a supplementary figure) and discussed in the discussion sections. Once more conclusive experiments are performed, this figure could be the beginning of an intriguing new manuscript.

Reviewer #2 (Remarks to the Author):

Schmidt et al, present a manuscript which outlines studies in human neural progenitor cells from sporadic PD patients and controls, investigating the metabolic phenotype using multiple methods linking this phenotype to changes in SHH signalling. This manuscript builds on the previous paper from the same group which already showed metabolic abnormalities and anomalies with SHH signalling. Here the authors use multiple omics measurements to build a story around a key enzyme defect in the citric acid cycle, and restoring SHH signalling by reducing overactive SHH signalling the metabolic abnormalities are also restored. Furthermore, the authors present findings that this pathway is independent of the complex I defect in sPD and therefore propose these are two independent hits of pathology. The study aims to answer crucial questions about the order in which the pathologies of sPD build up and therefore where to begin with therapeutic interventions. The authors propose a model based upon their findings; however some points need addressing. This study is based upon NPCs from 7 sPD patients (which have already been published on with a metabolic deficit and alterations in SHH signalling), it is clear from much of the data presented that there is variability in the sPD patient lines, this is the case for the respiration measures, the uptake experiments and indeed many others. In addition, the treatment effect of cyc is also very variable in the sPD lines, having the restorative effect the authors state in some lines but in others, it is clear they have not responded to the treatment. This raises the fundamental question, that with such a heterogeneous disease as sporadic PD, are the authors describing a metabolic pathway of dysfunction and repair which is of limited relevance to the whole sPD population. As the authors have published many of the metabolic and SHH phenotypes previously in these same lines, it begs the question how common this sequence of pathologies is in other sPD lines and cohorts. Furthermore, the response to cyc treatment is variable in this small sPD cohort requiring the need to study in a larger cohort of sPD patient lines. To make such generalisable statements as the authors do, more lines would need to be included in the study.

In addition, the manuscript relies mainly on the use of NPCs rather than iDANs and although the authors present very limited data in iDANs, the metabolic status of the NPCs may be (and has been shown to be by others) fundamentally different from fully differentiated cells. Therefore, this calls into question the relevance of the sequence of events and metabolic rewiring shown in the NPC cell model, in addition to the connection to SHH signalling. Both of these pathways may well be fundamentally different in fully differentiated cells, indeed the link with complex I deficiency may be direct in iDANs and it is not effected in the NPCs as they fundamentally are more metabolically flexible and less OXPHOS dependent.

Some of the data presented in this manuscript appears to have already been published in the same lines in the previous publication from this group, so there is a lot of repetition of results from their already published study; this is at times difficult to disentangle from the new data presented in this manuscript. The respiration data in Figure 1 with 25mM glucose or pyruvate, it would crucial for the authors to explain how this data differs or is the same culture conditions as the respiration data presented in their previous paper Figure 1 (2022), likewise for the complex I defect. In addition, the authors previously showed that cyc treatment could restore the respiration of sPD NPCs in their previous publication of 2022, therefore, the data presented in Figure 6 in this manuscript appears in part to be a direct repeat of the same experiment in the same lines. This is also the case for the transcriptomic analysis the authors present and refer to. It is critical the authors clearly define the novel data in this manuscript.

In Figure 6 the imaging quantification for the H3K9me3 is stated 600 cells per cell clone, have the authors imaged 600 cells in total across 3 differentiations of NPCs or iDANs or passages of fibroblasts? In the plots are all 600 cell levels indicated as individual data points? This could massively skew the statistics if each individual cell is being counted as an individual data point rather than the mean levels of the cells from each repeat of the experiment.

Reviewer #3 (Remarks to the Author):

Using a multilayered omics analysis of hiPSCs from sPD patients and healthy individuals as disease models, the authors have elucidated that the α -ketoglutarate dehydrogenase complex (OGDHC) in the citric acid cycle is the bottleneck in sPD metabolism. They previously reported that the dysfunction of PC signaling pathways and especially SHH signaling is a molecular pathway underlying PD development. In the current study, they demonstrated that the alterations in cellular metabolism and the OGDHC activity were restored by interfering the enhanced SHH signal transduction in PC function. The authors presented a human cellular system model of sPD that combines reduced glucose metabolism, reduced mitochondrial respiration, ATP production, OGDHC and complex I deficiency. The model would provide clues to the understanding of the molecular basis of this disease and expected to be used for development a neuroprotective therapy of sPD. Specific points are as follows.

Major points

Page 3 Result: Are the cell lines used in this study the same as those used in previous studies (Ref. 16)? If not, to confirm that the cells used in this study are appropriate models for sPD cells, the morphology and expression of NPC and DA markers should be shown, or references should be provided.

Page 4 Proteomics session and Page 10 Discussion: Proteomics data shows the decrease in OGDHL level and less change in OGDH in sPD hiPSC (Supplementary Data 3). Why didn't the authors mention the decrease in PGDHL in the proteomics session? When the metabolic categories (citric acid cycle) were enriched in the pathway enrichment analysis, the authors would have found a decrease in OGDHL prior to the metabolomic study. Comparing the amounts of OGDHL and the proteins discussed here, such as complex1 assembly, PC signaling proteins, among the seven model cells may yield results that support their model.

The authors concluded that complex I deficiency and the SHH mediated hypometabolism develop as two independent hits that negatively impact cellular energy supply. Is this a conclusion reached by averaging the omics data from the seven sPD hiPSC, or is it a common dysregulation across all seven model cells? In other words, does the PD model presented by the authors always include both dysregulations, or can it be one or the other?

Minor point

1. Supplementary Data 4: Data of mKEGG is missing.
2. Page 5 line 1: I cannot find the term of primary cilia (PC) in Fig. 3d, Supplementary Fig. 2d-c, and Supplementary Data 4). Is PC referring intraflagellar transport here?
3. Page 5 line 27; TIMMDC1 is not shown in Fig. 3.
4. Page10 line 14 and Fig 6: Fig 6i is a mistake for Fig. 6l.
5. Fig. 6: A brief explanation regarding Fig.6l would be helpful. I think it is not easy to understand this figure in the explanation on page 10 lines12-14.
6. Page19, LC/MS: Just to confirm, was the 50 cm column used without a trap column?
7. Page 19, DIA-MS: If m/z range was fractionated, information on m/z of separation, number of windows, whether overlapping windows were acquired should be provided.
8. Page19 line 39: Database download date is required.
9. Please show the full name of DEG, FCCP, ROS, and SHH.

Dear Reviewers,

We thank you all for your valuable comments on our manuscript which helped to improve the manuscript substantially.

Within the revised manuscript, we highlighted additional passages, which were requested by the reviewers, in yellow. Passages which we eliminated due to the restructuring of the manuscript are also indicated (crossed out).

A point-by-point response to each reviewers' comments is indicated below. For better readability of this part of our response, we highlighted each reviewers' comments in a different color, whereas our responses are always kept in black.

Thank you once again for your valuable input and specifically the time taken to help us to improve our manuscript.

Best regards,

Wolfgang Wurst

REVIEWER COMMENTS

1. Reviewer #1 (Remarks to the Author):

The manuscript by Schmidt et al. presents findings that support the role of hypometabolism in sporadic Parkinson's disease (sPD). The authors used human induced pluripotent stem cells (hiPSCs) from late-onset sPD patients and healthy individuals for disease modeling. They utilized a multilayered omics approach, including transcriptomics, proteomics, and metabolomics, to evaluate metabolic properties in sPD neuronal progenitor cells (NPCs). The authors suggest that mitochondrial dysfunction and alterations in glucose metabolism are prominent in sPD NPCs. Furthermore, they described alterations in primary cilia function, which may be connected to PD pathologies. In my opinion, this study is interesting and impactful but rather correlative and descriptive in nature. Although the molecular mechanistic analysis is not the strength of this manuscript, the large dataset provided by transcriptomics, proteomics, and metabolomics analysis would be valuable for the field, and the results presented have the potential to open up new avenues of research. This manuscript could be considered for publication in Nature Communications after addressing the following comments:

Figure 1:

- 1.1. sPD hiPSCs display reduced maximal respiration. Technically speaking, maximal respiration cannot be lower than baseline respiration. This is usually an indication of reduced cell density and suboptimal FCCP concentration. This point should be addressed carefully by repeating these experiments with similar cell numbers across conditions.

We repeated the Seahorse XF experiments with hiPSCs using different cell densities and on a new machine, however, always monitored a similar drift in our data. The decline of cellular respiration with medium injected at the specific timepoints instead of inhibitors and FCCP is depicted in **ReFig. 1a**. Interestingly, this drift was repeatedly measured with these hiPSCs, but disappeared upon differentiating the hiPSCs into neuronal cells (**Fig.1b,c** - see **chapter 1.2**). We discussed this issue with an expert in the field (Martin Jastroch -) – now also co-author of the manuscript – and he suggested the following procedure which is accepted in the field.

To correct for this drift in our datasets and to circumvent this technical limitation, we regressed out the decline in cellular respiration over time. For a direct comparison, the original dataset is plotted next to the corrected one in **ReFig. 1b,c**. This did not interfere with the conclusions presented in the manuscript, but allowed to correct for technical limitations that resulted in a lower maximal than baseline respiration.

Response to reviewer's comments

ReFig. 1 | Drift in cellular respiration of hiPSC. (a) Mitochondrial stress test performed in hiPSCs using a Seahorse XFe96 Extracellular Flux Analyzer. Cells were measured in Seahorse XF assay medium supplemented with 25 mM glucose or 5 mM pyruvate. Injected was assay medium without any compounds at the original timepoints. (b) Mitochondrial stress test performed in hiPSCs. Injected were (A) Oligomycin (1 μg/ml), (B) Carbonyl cyanide p-trifluoro-methoxyphenyl hydrazone (FCCP; 0.5 μM), (C) Rotenone (5 μM)/Antimycin A (2 μM), and (D) 2-Deoxyglucose (2-DG; 100 mM). (c) Mitochondrial respiration depicted in ReFig. 1b was corrected by the decline in respiration over time depicted in ReFig. 1a.

Measurement progression is shown with means ± standard error of the mean (SEM). Boxplots display the median and range from the 25th to 75th percentile. Whiskers extend from the min to max value. Each dot represents one patient. n = 5 Ctrl and 7 sPD patient-derived cell clones, in triplicates. p-values were determined by one-way ANOVA with Sidak's Post-hoc test. *, p < 0.05; **, p < 0.01; ***, p < 0.001.

Response to reviewer's comments

1.2. The addition of an ATP-linked respiration graph and the display of ECAR data would allow readers to interpret these results.

The ATP-linked respiration graphs have been included in **Fig. 1** (see below).

Response to reviewer's comments

The ECAR data have been included as **Supplementary Fig. 2a,b,c** (see below). The “glycolytic flux” analysis has been performed simultaneously to the original respiration analysis. As the ECAR mainly correlates with lactate/H⁺ secretion and can be masked by various other cellular processes, it only offers a rough overview of glycolytic rates and has to be interpreted with caution. Due to this limitation, we initially decided not to report the ECAR data. This warning is now also stated in the results part together with the discussion of the ECAR results on page 4, lines 10-30:

“To get a more comprehensive overview of cellular metabolism, the glycolytic flux based on the extracellular acidification rate (ECAR) was also assessed. The glycolytic flux was only analyzed in cells supplied with glucose as an energy substrate. The oxidation of glucose during glycolysis depends on ATP hydrolysis and results in the production of protons, pyruvate, and often lactate. ECAR mainly correlates with lactate/H⁺ secretion and can be masked by various other cellular processes leading to acidification. Thus, it only offers a rough overview of glycolytic rates and has to be interpreted with caution². The glycolytic flux analysis did not show significant differences in the ECAR between sPD and Ctrl hiPSCs, hNPCs, and DANs (**Supplementary Fig. 2a,b,c**), supporting our hypothesis of a defect downstream of glycolysis as indicated by the OCR measurements using pyruvate as a substrate. It might, however, also indicate that reduced mitochondrial ATP production in sPD is not compensated by an increased glycolytic flux and conversion of pyruvate to lactate. To further elucidate the impact of glycolytic flux and mitochondrial respiration to energy production, we calculated the ECAR to OCR ratio and visualized the total levels by plotting the ECAR against the OCR (**Supplementary Fig. 2d,e,f**). The higher the ratio, the lower the proportion of mitochondrial respiration for energy production should be. Indicative for the well-known glycolytic switch during neuronal differentiation, the ECAR to OCR ratio is declining during this process with the highest values in hiPSCs and the lowest ones in DANs. Furthermore, the ratio is significantly increased in sPD hNPCs and tends to be increased in hiPSC and DANs. This indicates a decreased proportion of mitochondrial respiration to total energy production in sPD.”

Response to reviewer's comments

Supplementary Fig. 2 | Glycolytic flux analysis of Ctrl and sPD cell lines with glucose as energy substrate. (a) Extracellular acidification rate (ECAR) analyzed in hiPSCs, **(b)** thereof differentiated hNPCs, and **(c)** DANs using a Seahorse XFe96 Extracellular Flux Analyzer. Cells were measured in Seahorse XF assay medium supplemented with 25 mM glucose. Injected were (A) Oligomycin (1 μ g/ml), (B) Carbonyl cyanide p-trifluoro-methoxyphenyl hydrazone (FCCP; 0.5 μ M), (C) Rotenone (5 μ M)/Antimycin A (2 μ M), and (D) 2-Deoxyglucose (2-DG; 100 mM). **(d)** (left) The ECAR to OCR ratio was calculated by dividing the basal glycolytic flux by the basal mitochondrial respiration measured using glucose as energy substrate (**Fig. 1b**) and reflects the cell's tendencies for ATP production. (right) The separation between Control and sPD lines is visualized by plotting the basal mitochondrial respiration (OCR) against the basal glycolytic flux (ECAR). Measurement progression is shown with means \pm standard error of the mean (SEM). Boxplots display the median and range from the 25th to 75th percentile. Whiskers extend from the min to max value. Each dot represents one patient. n = 5 Ctrl and 7 sPD patient-derived cell clones, in triplicates. p-values were determined by two-sided t-test **a** (max), **b**, **d**, **e**; two-sided Mann-Whitney-U test **a** (basal), **c**, **f**. *, p < 0.05; **, p < 0.01; ***, p < 0.001.

Response to reviewer's comments

1.3. It is well known that as progenitor cells develop into mature neurons, metabolism shifts from glycolysis to oxidative phosphorylation. In order to interpret the observed differences, especially in DANs, the authors should include images with differentiation markers to confirm that the control vs. sPD DANs populations are similar in terms of neuronal differentiation. For example, do these neurons fire action potentials? Do they display similar neuronal processes?

Immunohistochemical staining of differentiation markers have been initially reported in our previous publication¹ as Fig. 1 (shown here as **ReFig. 2**). To verify that DAN differentiation and maturation hasn't been affected in sPD clones, neurons/DANs have been counted and DAN morphology was assessed.

ReFig. 2 | Taken from ¹ (a) Immunostainings exemplarily shown for O3H-R1-003. hiPSC pluripotency staining for markers OCT4, NANOG, SOX2. Scale bar=200 μ m. hNPC staining for markers SOX1, SOX2, NESTIN, PAX6. Scale bar=100 μ m. Neuron staining for markers TUBB3 and DAN marker TH. Scale bar=100 μ m. Astrocyte staining for markers GFAP and SLC1A3. Scale bar=100 μ m. (b) Quantification of RBFOX3 (synonym: NeuN) positive as well as TH / RBFOX3 double-positive cells in DAN populations. $n = 5$ Ctrl and 7 sPD clones, in triplicates. (c) Characterization of neurite morphologies of DANs. Boxplots show the average number of neurites emerging from TH positive cell bodies, their average number of branch points and their average length. $n = 5$ Ctrl and 7 sPD clones, in triplicates. Boxplots display the median and range from the 25th to 75th percentile. Whiskers extend from the min to max value. Each dot represents one patient. P -values were determined by two-sided t -test **b** (right), **c**; two-sided Mann–Whitney-U test **b** (left). * $p < 0.05$, ** $p < 0.01$, *** $p < 0.001$.

Response to reviewer's comments

Additionally, we now also included a quantification of the hNPC marks SOX1, SOX2, and NESTIN as **Supplementary Fig. 1 (see below)** to validate that the differentiation stage is comparable between Ctrl and sPD clones.

This is now explicitly stated at the beginning of the results part on page 3, lines 17-27:

“Differentiation stages were confirmed using immunohistochemical staining for characteristic markers such as the precursor markers NESTIN, SOX2, and SOX1 (**Supplementary Fig. 1a**) or the DAN marker TUBB3, RBFOX3 (synonym: NeuN), and TH¹. Expression of these differentiation markers was not affected in sPD hNPCs (**Supplementary Fig. 2b**), nor was the abundance and morphology of DANs derived thereof¹. This indicates that the DAN differentiation process assessed at various stages was comparable between Ctrl and sPD cells.”

Supplementary Fig. 1 | Characterization of hiPSC-derived hNPCs. (a) Immunostainings are exemplarily shown for clone O3H-R1-003 using antibodies against the NPC markers SOX1, SOX2, NESTIN. Scale bar=100 μ m. **(b)** Violin plots show the mean cytosolic fluorescence intensity of NESTIN, or **(c)** the mean nuclear fluorescence intensity of SOX1, or **(d)** SOX2. (left) Samples are pooled for a Ctrl-sPD comparison, or (right) values are plotted per patient-derived clone. The dashed line indicates the median fluorescence intensity level of Ctrl clones. Boxplots display the median and range from the 25th to 75th percentile. Whiskers extend to the most extreme data point which is no more than 1.5 times the interquartile range. 600 cells per clone were analyzed from n = 5 Ctrl and 7 sPD clones, in triplicates. p-values for the Ctrl-sPD comparison were determined by linear mixed effects model. *, p < 0.05; **, p < 0.01; ***, p < 0.001.

Figure 2:

- 1.4. To evaluate mitochondrial content, the authors used mitotracker staining, which is sensitive to mitochondrial membrane potential. This staining will exclude a sub-population of "unhealthy" depolarized mitochondria. Either mitochondrial antibody staining or a combination of membrane potential sensitive and insensitive mitochondrial stains will be necessary to evaluate mitochondrial content accurately.

The analysis of mitochondrial mass and morphology has been repeated in hNPCs and is **now also included for the DANs derived thereof**. In addition to the mitoTracker staining to visualize active/functional mitochondria, we now also included immunohistochemical staining of the ATP synthase using an antibody against ATP5F1A to visualize the total pool of mitochondria per cell. In addition to the rough quantification of mitochondrial mass by quantifying the fluorescence intensity per cell, we now also included a more detailed analysis of mitochondrial morphology at higher resolution. In total five measures of mitochondrial morphology were assessed: The number of mitochondria per cell, the mean area and fluorescence intensity per cell, as well as the mean length of the morphological skeleton and its number of branch points per cell. This is now included as **Fig. 2c,d (see below)**.

Additionally, we quantified the expression and post-translational modifications of components of the fusion and fission machinery to get a more detailed insight into mitochondrial dynamics. This is now included as **Fig. 2 e,f,g (see below)**.

This is now explicitly stated and discussed from page 4, line 32 to page 5, line 26:

"Mitochondrial health is not affected in neural cells derived from sPD patients"

Aiming to identify the underlying causes of reduced mitochondrial respiration, we first investigated mitochondrial mass and morphology. Alterations in both have been previously described in various PD models³⁻⁶.

The amount of total mitochondrial numbers and morphology was determined using two different independent stainings. On one hand, we stained the ATP synthase with an antibody against ATP5F1A in hNPCs (**Fig. 2a**) and DANs derived thereof (**Fig. 2b**). To specifically visualize functional/active mitochondria, we used MitoTracker which accumulates specifically in mitochondria with intact membrane potential (**Fig. 2a,b**). In total, five characteristics of mitochondrial morphology were assessed: The number of mitochondria per cell, the mean area and fluorescence intensity of the mitochondria per cell, as well as the mean length of their morphological skeleton and its number of branch points per cell. However, neither in hNPCs (**Fig. 2c**) nor in DANs (**Fig. 2d**) derived thereof any significant differences between Ctrl and sPD clones could be observed, indicating that mitochondrial content or general health is not impaired in neuronal cells derived from sPD hiPSCs.

Additionally, we characterized the abundance and post-translational modifications of the mitochondrial fusion and fission machinery which is essential for mitochondrial quality control and functioning. These processes allow cells to adapt the mitochondrial morphology to cellular metabolic demands and substrate supply⁷ and is a measure for mitochondrial stress. Fusion of the outer and inner mitochondrial membrane is mainly facilitated by the mitofusions (MFN1 and MFN2), as well as OPA1, respectively, and mitochondrial fission is mainly mediated by DRP1⁷. As fusion is regulated by the expression of the corresponding components, expression levels of *MFN1*, *MFN2*, and *OPA1* were assessed on mRNA level using RT-qPCR. Contrary, fission is mainly regulated by post-translational modifications which were quantified on protein level using western blots. For example, phosphorylation of Serine at position 616 is thought to activate fission activity^{7,8}. As mitochondrial function highly relies on substrate availability, cells were supplied with glucose or pyruvate as during the mitochondrial respiration analysis to unmask possible deficits in adjusting mitochondrial morphology to cellular demands in sPD clones. However, no significant differences in the expression of *MFN1*, *MFN2*, or *OPA1*, (**Fig. 2e**) as well as in the abundance of DRP1, or DRP1-pSerine⁶¹⁶ (**Fig. 2f,g**) could be identified in sPD hNPC neither with glucose nor with pyruvate as energy

substrate. Upon changing the substrate supply from glucose to pyruvate, a significant upregulation in the expression of *MFN2* and thus the fusion machinery could be observed for both Ctrl and sPD clones, possibly as a reaction to the loss of glycolytic energy production. These results further validate our previous results and excludes a decrease in mitochondrial mass or alterations in mitochondrial dynamics as an explanation for the respiratory deficiency that was observed in Seahorse analysis."

Fig. 2 | Analysis of mitochondrial abundance and morphologies. (a) Quantification of total mitochondrial mass and functional mitochondria in hNPCs or (b) DANs. Total mitochondrial mass was visualized using immunostainings with an antibody against the ATP synthase (ATP5F1A). Functional mitochondria with an active membrane potential were visualized using a MitoTracker probe (200 nM for 20 min). Images are exemplarily shown for hNPCs of clone i1E4-R1-003 (Ctrl), and iR66-R1-007 (sPD); for DANs of clone i1E4-R1-003 (mitoTracker – Ctrl), iR66-R1-007 (mitoTracker – sPD), i1JF-R1-018 (ATP5F1A – Ctrl), and iJ2C-R1-015 (ATP5F1A – sPD). Scale bar = 20 μ m. (c) Violin plots highlight some morphological characteristics of mitochondria in hNPCs or (d) DANs: Number of mitochondria (count), mitochondrial area, skeleton length, number of branch points, and mean

Response to reviewer's comments

intensity. On average, 112 (hNPCs – ATP5F1A), 160 (hNPCs – mitoTracker), 370 (DAnS – ATP5F1A), 360 (DAnS – mitoTracker) cells per clone were analyzed from n = 5 Ctrl and 7 sPD clones, in triplicates. **(e)** To assess alterations in the mitochondrial fusion machinery, expression of the mitofusions *MFN1*, *MFN2*, and *OPA1* was quantified in hNPCs by RT-qPCR. Cells were cultivated on the energy substrates used in the Seahorse XF analysis (25mM glucose or 5mM pyruvate). **(f)** To assess alterations in the mitochondrial fission machinery, expression of *DRP1* and its phosphorylation on Ser⁶¹⁶ was quantified in hNPCs by western blot. Protein levels were normalized to ACTB. **(g)** Western blots are exemplary shown for some hNPC clones. Boxplots display the median and range from the 25th to 75th percentile. Whiskers extend from the min to max value or to the most extreme data point which is no more than 1.5 times the interquartile range (for **c,d**). Each dot represents one patient. p-values were determined by linear mixed effects model **c, d**; one-way ANOVA with Sidak's Post-hoc test **e, f**. *, p < 0.05; **, p < 0.01; ***, p < 0.001.

- 1.5. High-resolution zoomed-in images of both control and sPD hNPCs should be included to interpret the results.

High-resolution images of both control and sPD hNPCs and DAnS have been included as **Fig. 2a,b** (see **chapter 1.4**).

- 1.6. It is also not clear what the n values represent. This analysis should be performed per cell, and the total cell number should be increased for accurate statistical analysis.

The outdated original analysis was performed per cell for roughly 600 cells per clone. For visualization purpose, the average value per clone/patient has been calculated and reported.

After updating this analysis, we now reported the data as violin plots (density plot + box plot) in **Fig. 2c,d** (see **chapter 1.4**) summarizing all measured cells per condition. The number of cells analyzed per condition or patient has been listed in the figure legend, but is also reported in the analysis notebook deposited in GitHub next to the underlying measurements for each cell.

- 1.7. Downregulation of mRNA levels in sPD cells but no change in mitochondrial protein level may indicate alterations in protein turnover rate, which is related to total mitochondrial use – also addressed for Figure 1. Thus, further analysis of the NPC stages should be analyzed in control vs. sPD.

(See **chapter 1.3**) We now included a quantification of the hNPC marks *SOX1*, *SOX2*, and *NESTIN* as **Supplementary Fig. 1** to validate that the differentiation stage is comparable between Ctrl and sPD clones. This is now explicitly stated at the beginning of the results part on page 3, lines 17-27.

Figures 3/4:

- 1.8. The differences in transcriptome and proteome observations are valuable for the field, and very few studies address this. This adds strength to the manuscript. This reviewer acknowledges that while it may be challenging, validation of this data with DANs (in addition to NPSs) would really enhance the impact of this manuscript for the field.

We agree that additional transcriptome and proteome data of DANs would be quite interesting to monitor alterations in expression throughout different cell types and during differentiation processes. These experiments are currently planned, however, in our opinion out of the scope of this manuscript.

- 1.9. Considering the authors' previous publication (Ref 16), which indicates that mitochondrial dysfunction develops in sPD upon dopaminergic neuron differentiation, a detailed analysis of mitochondrial health will be important to explain citric acid cycle defects.

We thank the reviewer for this valuable comment. Indeed, a further analysis of mitochondrial health including a more detailed analysis of mitochondrial abundance, morphology, and dynamics is crucial to interpret the findings described in manuscript. Thus, we have invested a lot to include these data into the manuscript which are now shown in **Fig. 2 (see chapter 1.4)** and **Fig. 3 (see below)**.

This has been mentioned already in previous answers but is also summarized here. The analysis of mitochondrial mass and morphology has been repeated in hNPCs and is now also included for the DANs derived thereof. In addition to the mitoTracker staining to visualize active/functional mitochondria, we now also included immunohistochemical staining of the ATP synthase using an antibody against ATP5F1A to visualize the total pool of mitochondria per cell. In addition to the rough quantification of mitochondrial mass by quantifying the fluorescence intensity per cell, we now also included a more detailed analysis of mitochondrial morphology. In total five measures of mitochondrial morphology were assessed: The number of mitochondria per cell, the mean area and fluorescence intensity per cell, as well as the mean length of the morphological skeleton and its number of branch points per cell. This is now included as **Fig. 2c,d (see chapter 1.4)**.

Additionally, we quantified the expression and post-translational modifications of components of the fusion and fission machinery to get a more detailed insight into mitochondrial dynamics. This is now included as **Fig. 2 e,f,g**.

This is now explicitly stated and discussed from page 4, line 32 to page 5, line 26 (**see chapter 1.4**).

Revision of manuscript NCOMMS-23-06769-T - Molecular mechanisms underlying a reversible state of hypometabolism in sporadic Parkinson's disease
Response to reviewer's comments

Fig. 3 | Abundance of the electron transport chain complexes is not affected in sPD. (a) Metabolic KEGG pathways enriched in sPD hNPCs. Single-cell transcriptome data (bulk-like) were previously published¹. FDR corrected p-values are represented by q-values. (b) Visualization of the 'Electron Transport Chain (OXPHOS system in mitochondria)' pathway with manual annotations. Color intensities for up- (red) and downregulated (blue) genes are proportional to the fold change. Not significantly altered genes are colored in white. (c) The abundance of mitochondrial complexes I - IV was quantified by Western blot with antibodies against the labile subunits NDUFB8 (complex I), SDHB (complex II), UQCRC2 (Complex III), MT-CO2 (Complex IV) and ATP5F1A (Complex V). Expression levels were normalized to ACTB or α-Tubulin levels. Western blots are exemplary shown for some hNPC clones. Quantifications of protein levels are shown for (d) hNPCs and (e) DANs. Boxplots display the median and range from the 25th to 75th percentile. Whiskers extend from the min to max value. Each dot represents one patient. p-values were determined by one-sided hypergeometric tests a, two-sided t-test d, e. *, p < 0.05; **, p < 0.01; ***, p < 0.001.

Figure 5:

- 1.10. The most accurate method to assess glucose uptake would be to utilize genetically encoded glucose sensors.

This option has been discussed when designing these experiments. Due to the following reasons we decided to choose the commercially available Glucose Assay Kit over genetically encoded glucose sensors.

The Glucose Assay Kit allows to detect glucose easily and robustly in the medium with a well-defined range from 39 μM – 16,600 μM glucose. Thus, it is very sensitive and would allow to calculate absolute glucose quantities or the absolute glucose uptake of cells after a defined period of time. Furthermore, variabilities in cell culture e.g. due to numbers of cells can be easily corrected for by e.g. normalizing the data to total cell numbers.

In contrast, the genetically encoded glucose sensor only allows to quantify total glucose levels in cells. Thus, it is more static and has a very limited usability to monitor the dynamic glucose uptake and metabolization of cells. The total level might be similar while a proportion of glucose is metabolized and new glucose is taken up. Another disadvantage of genetically encoded sensors is the problem with deliverability and expression. Neither transient overexpression nor stable integration of glucose transporters can be used for an easy and robust comparison of glucose levels within multiple cell lines. Amongst others, the quantification might be limited due to 1) different transfection rates between clones; 2) different expression rates of transiently delivered DNA/mRNA; 3) different locus accessibility and thus expression between cell clones for stable integrations. Thus, many factors exist that are introducing variability which would need to be corrected for.

- 1.11. This is because the gene expression level of glucose transporters may not represent the protein level or total protein present on the plasma membrane responsible for glucose transport.

This is especially relevant in the context of the transcriptome-proteome discrepancies reported in our manuscript. We thank the reviewer for pointing this out.

Thus, the expression levels of the main glucose transporters SLC2A1 and SLC2A3 have been analyzed on the protein level using Western blots in hNPCs as well as in DANs derived thereof. These data are in line with the mRNA expression data. Furthermore, the radial distribution of the glucose transporters has been assessed to quantify their cellular localization (membrane vs cytosol).

This is now included as **Fig. 6e,f,g,h,i** and is stated and discussed on page 9, lines 24-28:

“Also on protein level, the expression of the most abundant glucose transporters SLC2A3 and SLC2A1 was not affected, neither in sPD hNPCs (**Fig. 6e,f**) nor in DANs (**Fig. 6g**). Additionally, we quantified the cellular and membrane localization of these glucose transporters using immunohistochemical staining in hNPCs (**Fig. 6h**). Again, the mean fluorescence intensity as well as the radial distribution of SLC2A1 and SLC2A3 was not altered in sPD hNPCs (**Fig. 6i**).”

Besides quantifying the abundance and cellular localization of glucose transporters, we already have included a functional assessment of glucose transporters and glucose phosphorylation by HK1 in the originally submitted version (now **Fig. 6j**). This also indicates that glucose uptake and phosphorylation are not the limiting metabolic steps in sPD clones. This is stated on page 9, lines 29-36:

“Still, to assess the overall function of glucose transporters and hexokinase-1 (HK1) which mediates the first step of glycolysis to capture glucose within cells, the uptake and phosphorylation of the glucose analog

Response to reviewer's comments

2-deoxyglucose was quantified. 2-deoxyglucose is phosphorylated by HK1, but inhibits further glycolytic processing and thus accumulates within cells⁹."

Fig. 6 | Metabolite uptake and secretion rates indicate a state of hypometabolism in sPD. (a) Uptake/secretion rates of main exchange metabolites of hNPCs. Levels of metabolites were measured in the growth medium of hNPCs after a 24h incubation period using a YSI biochemistry analyzer. **(b)** Alterations in glucose uptake were validated independently in hNPCs and **(c)** DANs. Quantified were glucose levels in the growth medium after a 24h incubation period. Values were subtracted from blank values and normalized to mean levels of Ctrl hNPCs or DANs, respectively. **(d)** Normalized gene expression levels of the glucose transporters *SLC2A1*, *SLC2A2*, *SLC2A3*, and *SLC2A4* analyzed on mRNA level by RT-qPCR in hNPCs. **(e)** The abundance of glucose transporters on protein level was quantified by Western blot with antibodies against *SLC2A1* and *SLC2A3*. Expression levels were normalized to *GAPDH*. Western blots are shown exemplarily for some hNPC clones. Quantification of protein levels is shown for **(f)** hNPCs and **(g)** DANs. **(h)** The cellular localization of glucose transporters was visualized using immunostainings with antibodies against *SLC2A1* and *SLC2A3*. Immunostainings are exemplarily shown for hNPCs of clone i11JF-R1-018 (*SLC2A1* – Ctrl), iM89-R1-005 (*SLC2A1* – sPD), iO3H-R1-005 (*SLC2A3* – Ctrl), and iAY6-R1-003 (*SLC2A3* – sPD). Scale bar = 20 μ m. **(i)** To quantify the radial distribution of glucose transporters in hNPCs, the cytoplasmic area was separated into 10 rings around the center of the corresponding nuclei. Violin plots show the cytoplasmic area, mean fluorescence intensity of the cytoplasm as well as of the outermost rings 9 and 10. On average, 3960 (*SLC2A1*), 4062 (*SLC2A3*) cells per clone were analyzed from n = 5 Ctrl and 7 sPD clones, in triplicates. **(j)** Quantification of cellular 2-deoxyglucose (2DG) levels of hNPCs after a 15min incubation period indicated that glucose transporters and hexokinase 1 activity are not the bottlenecks in sPD metabolism. Boxplots display the median and range from the 25th to 75th percentile. Whiskers extend from the min to max value or to the most extreme data point which is no more than 1.5 times the interquartile range (for i). Each dot represents one patient. n = 5 Ctrl and 7 sPD patient-derived cell clones, in triplicates or 6 replicates for the non-targeted metabolome analysis. p-values were determined by p-values were determined by linear mixed effects model **a**, **i**; two-sided Mann–Whitney-U test **b**, **c**, **d** (*SLC2A2*); two-sided t-test **d** (*SLC2A1*, *SLC2A3*, *SLC2A4*), **f**, **g**, **j**. *, p < 0.05; **, p < 0.01; ***, p < 0.001.

Response to reviewer's comments

1.12. The level of 2DG also depends on HK1 activity.

We thank the reviewer for this correction. This has been updated on page 9, lines 29-31 (**see chapter 1.11**).

1.13. Increased glucose lactate conversion may also indicate an enhanced glycolysis rate in sPD (as shown in Fig 5e and Supp Fig 3c), which is common when mitochondria are dysfunctional. Therefore, ECAR graphs in Figure 1 will be significant. Designing seahorse measurements to specifically evaluate glycolysis rate would also address this issue.

The ECAR data have been included as Supplementary **Fig. 2a,b,c** (**see chapter 1.2**). The “glycolytic flux” analysis has been performed simultaneously to the original respiration analysis. As already discussed above, the ECAR mainly correlates with lactate/H⁺ secretion and can be masked by various other cellular processes. Thus, it only offers a rough overview of glycolytic rates and has to be interpreted with caution. Due to this limitation, we initially decided not to report the ECAR data. This warning is now also stated in the results part together with the discussion of the ECAR results on page 4, lines 10-21 (**see chapter 1.2**).

Next to including the ECAR data, we now also plotted the ECAR over the OCR and reported the ECAR-to-OCR ratio for hiPSC, hNPCs, and DANs. This is now included in **Supplementary Fig. 2d,e,f** (**see chapter 1.2**).

Although total glycolytic rates are not increased in sPD clones, the ECAR-to-OCR ratio is (trend in DANs; significantly in hNPCs) increased due to the deficit in mitochondrial respiration. This is now stated in the manuscript on page 4, lines 22-30 (**see chapter 1.2**).

1.14. To further elaborate the changes in glutamine metabolism, the author should conduct experiments in the presence of an MPC inhibitor as previously described by Divakaruni A. et al. in JCB 2017.

Stable isotope tracing experiments with either [U-¹³C]Glutamine or [U-¹³C]Glucose in the presence of the mitochondrial pyruvate carrier (MPC) inhibitor UK5099 (as described by ¹⁰) have been performed in hNPCs (**ReFig. 3a**). As shown by the mass isotopomer distributions (MIDs) in **ReFig. 3e,f**, the MPC inhibition was successful. Following MPC inhibition, the α -ketoglutarate dehydrogenase complex (OGDHC) was still reduced in sPD. The activity was quantified based on the ratio of M4-succinate to M5- α -ketoglutarate (**ReFig. 3b**). This further validates our main finding. However, these additional experiments did not provide further insights into metabolic alterations in sPD under basal condition and thus were not included in the manuscript.

ReFig. 3 | Inhibition of the mitochondrial pyruvate carrier. Comparison of untreated samples with MPC inhibitor treated samples (UK5099; 100 μ M for 24h). **(a)** Overview of carbon transitions for $[U-^{13}C]$ Glutamine and $[U-^{13}C]$ Glucose metabolism following MPC inhibition. **(b)** Quantification of the α -ketoglutarate dehydrogenase complex activity for carbons originating from $[U-^{13}C]$ Glutamine. Ratio of M4-succinate over M5- α -ketoglutarate. **(c)** Quantification of the oxidative glutamine flux of carbons originating from $[U-^{13}C]$ Glutamine. Ratio of M3- over M5-glutamate. **(d)** Quantification of the reductive glutamine flux of carbons originating from $[U-^{13}C]$ Glutamine. Ratio of M5-citrate over M5-glutamate. **(e)** Mass isotopomer distributions (MIDs) determined by isotopic tracing with $[U-^{13}C]$ Glucose (Glc) or **(f)** $[U-^{13}C]$ Glutamine (Gln) and GC-MS measurement.

Boxplots display the median and range from the 25th to the 75th percentile. Whiskers extend from the min to max value. Each dot represents one patient. $n = 5$ Ctrl and 7 sPD patient-derived cell clones, in triplicates. p-values were determined by linear mixed effects model **b, c, d**; no statistical test applied **e, f**. *, $p < 0.05$; **, $p < 0.01$; ***, $p < 0.001$.

Figure 6:

- 1.15. In the opinion of this reviewer, the connection between SHH signaling/PC and hypometabolism is highly suggestive, but the data presented is still preliminary and represents the weakest point of this manuscript. This section could be removed (Fig 6a-b and Supp Fig 5 can be combined as a supplementary figure) and discussed in the discussion sections. Once more conclusive experiments are performed, this figure could be the beginning of an intriguing new manuscript.

The fact that alterations in basal metabolism (hypometabolism) and enzyme activity (OGDHC) were regulated in a signaling dependent manner specifically in sPD, is a novel finding, and in our opinion of the greatest interest for the community. Especially as SHH signaling is thought to have a beneficial effect on DAN functionality and survival, its correct dose dependent and timely integration into the cellular context is highly important also for potential therapeutic interventions. Furthermore, these experiments allow for the first time to discriminate between different groups of metabolic alterations in sPD. The SHH "inducible" factors such as OGDHC deficiency and hypometabolism as well as "non-inducible" factors such as complex I deficiency might display two hits that have to be targeted in potential neuroprotective therapies independently.

These assumptions are further strengthened by the correlation analysis displayed in **Fig. 9**. The nuclear levels of SHH transcription factors (GLI3-FL/GLI3-R) as a result of the SHH signal transduction significantly correlated with OGDHL protein levels as well as OGDHC activity. Thus, with an increased nuclear activator/repressor ratio the OGDHL levels are increasing as well as total OGDHC activity. Or vice versa, with increased nuclear GLI3-R (repressor) levels as mediated by the overactive SHH signal transduction in sPD, the OGDHL protein levels and total complex activity is decreasing. This is now stated in the manuscript on page 13, lines 26-35:

"Interestingly, the ciliary capacity to transduce SHH signaling correlated with metabolic parameters. The capacity was assessed by monitoring the nuclear levels of GLI3-Full length (GLI3-FL) which is thought to function as weak transcriptional enhancer and the truncated GLI3-Repressor (GLI3-R) form. An increase of the GLI3-FL/GLI3-R (enhancer/repressor) ratio positively correlated with the protein abundance of the predicted SHH target OGDHL in hNPCs (R= 0.98; p-value= 0.00047). OGDHL protein levels thereby positively correlate with overall OGDHC activity (R=0.73; p-value= 0.098). These findings together with the correlations of disease progression with OGDHC activity and OGDHL protein levels ($\Delta H\&Y$ (R=-0.82; p-value=0.046) further strengthen the observed impact of ciliary mediated SHH signaling on sPD onset and progression."

Thus, in this manuscript we provide first evidence that 1) the OGDHC deficiency and the hypometabolism observed in sPD is triggered by alterations in SHH signal translation possibly via 2) a direct misregulation of the brain-specific OGDHC subunit OGDHL. Due to the importance of these findings for the overall statement of the manuscript and further research, we decided to display and discuss these results in the manuscript. It is noteworthy to state, however, that these findings will be further elucidated in a new manuscript in the future.

Revision of manuscript NCOMMS-23-06769-T - Molecular mechanisms underlying a reversible state of hypometabolism in sporadic Parkinson's disease
Response to reviewer's comments

Fig. 9 | OGDHC activity and metabolic alterations correlate with disease progression in sPD patients. (a) Multiple factor analysis using DEGs (bulk-like; top 100) (Fig. 3 and ¹), DEPs (top 100) (Fig. 4), significantly altered metabolites (45 metabolites) (Fig. 5), as well as the key experimental variables displayed in Fig. 1- 7 and ¹ describing alterations in sPD (61). The group points and lines represent the patient coordinates conditioned by the corresponding group variables. **(b)** Contributions of quantitative variables to the dimension 1 and 2 of the multiple factor analysis. **(c)** Correlogram visualizing the Pearson correlation coefficients of variables measured in sPD hNPCs and DAns, as well as parameters associated with disease progression in the sPD patients over a period of 9-12 years (Table 1). Disease progression markers are highlighted in green. Colors and dot sizes are proportional to the correlation coefficients. Negative correlations are colored in blue, positive correlations in red. **(d)** Visualization of interesting dependencies within measured variables and parameters of disease progression in sPD patients. Each point represents one patient. Linear regressions were calculated for each group and are displayed with a 95% confidence interval. The Pearson correlation coefficient (R) is displayed next to the corresponding p-value (p). Correlations with p-values < 0.1 were considered significant. **(e)** Graphical overview summarizing the main findings.

2. Reviewer #2 (Remarks to the Author):

Schmidt et al, present a manuscript which outlines studies in human neural progenitor cells from sporadic PD patients and controls, investigating the metabolic phenotype using multiple methods linking this phenotype to changes in SHH signalling. This manuscript builds on the previous paper from the same group which already showed metabolic abnormalities and anomalies with SHH signalling. Here the authors use multiple omics measurements to build a story around a key enzyme defect in the citric acid cycle, and restoring SHH signalling by reducing overactive SHH signalling the metabolic abnormalities are also restored. Furthermore, the authors present findings that this pathway is independent of the complex I defect in sPD and therefore propose these are two independent hits of pathology. The study aims to answer crucial questions about the order in which the pathologies of sPD build up and therefore where to begin with therapeutic interventions. The authors propose a model based upon their findings; however some points need addressing.

- 2.1. This study is based upon NPCs from 7 sPD patients (which have already been published on with a metabolic deficit and alterations in SHH signalling), it is clear from much of the data presented that **there is variability in the sPD patient lines**, this is the case for the respiration measures, the uptake experiments and indeed many others.

In addition, the treatment effect of cyc is also very variable in the sPD lines, having the restorative effect the authors state in some lines but in others, it is clear they have not responded to the treatment. This raises the fundamental question, that with such a heterogeneous disease as sporadic PD, **are the authors describing a metabolic pathway of dysfunction and repair which is of limited relevance to the whole sPD population.**

As the authors have published many of the metabolic and SHH phenotypes previously in these same lines, it begs the question how **common this sequence of pathologies is in other sPD lines and cohorts.**

Furthermore, the response to cyc treatment is variable in this small sPD cohort requiring the need to study in a larger cohort of sPD patient lines. **To make such generalisable statements as the authors do, more lines would need to be included in the study.**

We thank the reviewer for raising these points about variability in our cell lines (both control and sPD). We want to answer all these comments in the following way:

1. Not many have studied sporadic Parkinson's disease using hiPSC. Out of 385 PD hiPSC lines available in 2020, only 7% (~27 hiPSC lines) are originating from sporadic PD patients. The vast majority of studies is focusing on modeling PD using LRRK2-G2019S, PRKN exon deletions, PINK1 Q456X, SNCA triplication, and GBA N370S hiPSC lines. In average, these current hiPSC-PD studies used five hiPSC lines in total (two Ctrl hiPSC vs. 3 patient derived hiPSC lines)¹¹. Numbers tend to be higher for sporadic PD studies. Recent studies are summarized in **ReTable 1**. Thus, with hiPSC lines derived from 5 Ctrl and 7 sPD patients this study is already in the upper range of recent literature.

ReTable 1: Summary of recently published Parkinson's disease paper working with hiPSCs.

Study	Year	Journal	Number of sPD hiPSC lines	Number of control hiPSC lines
Rosh et al. ¹²	2023	bioRxiv	4	3
Chlebanowska et al. ¹³	2023	bioRxiv	7	5
Corenblum et al. ¹⁴	2023	Progress in Neurobiology	4	4
Stern et al. ¹⁵	2022	NPJ Parkinsons Dis.	4	3
Vlasov et al. ¹⁶	2021	Cells	2	2
Badanjak et al. ¹⁷	2021	Front Cell Dev Biol.	1	1
Schulze et al. ¹⁸	2018	acta neuropathol commun	8	6
Burbulla et al. ¹⁹	2017	Science	2	2
Fernández-Santiago et al. ²⁰	2015	EMBO Mol Med	6	4

2. Given the kind of experiments done in this study it is – in our modest opinion – beyond the scope of such a timely study to include at this point more sPD lines due to the following reasons:
- There are no further hiPSCs reprogrammed in the same way as the ones we have already analyzed available to us right now. Thus, we would have to get access to new patient material (including the ethical approvals), reprogram the fibroblasts, and differentiate the hiPSCs before being able to analyze them. For this analysis, we would have to rerun all potential new lines together with the old ones again – in order to reduce technical errors inevitably introduced by omics studies. Thus, including more lines into our study would request – in order to produce reliable and scientifically sound results – to repeat all experiments (specifically the omics studies) for all lines – not only the potential new ones – again.
 - We are convinced that adding even about 5 or 10 lines (both from controls and patients) to this study this would not reduce variability and will not mitigate this argument. It has been estimated that sample sizes of 10–30 individuals per hiPSC study are required to achieve the necessary statistical power ¹¹. Based on this estimation, our study (containing 12 hiPSC lines) is within the required range and thus we are convinced that our results are of the highest relevance for the field. This conviction is also based on two additional arguments:
 - We are able to reproduce clinically relevant disease phenotypes. Concerning our main finding of a sPD specific state of hypometabolism, similar observations have been reported in sPD patients using positron emission tomography (PET) imaging ^{21–23}. Additionally, alterations in OGDHC activity have been implicated in multiple neurodegenerative diseases such as Huntington's disease, Alzheimer's disease (reduction of ~30-90% ²⁴), and Parkinson's disease (reduction of ~50% ^{25,26}) to various degrees. Also, OGDHC abundance is reduced in postmortem brain regions of PD patients ²⁷. But as the reviewer correctly notes, due to the known heterogeneity and complexity of the disease - this may not necessarily be the case for all PD patients. However, to interpret these observations made in patients and to tackle these phenotypes in therapeutic approaches, a more detailed understanding is essential which we aim to provide using our cellular models.
 - In a translational effort, we compared the core results based on the analysis of our cellular model system (hiPSC and hNPCs, DANs differentiated thereof) with clinical patient data on disease progression. The sPD patients that donated the original

Response to reviewer's comments

skin fibroblasts used for hiPSC generation were characterized at the timepoint of biopsy and were assessed again after 9-12 years. Disease progression was assessed by monitoring changes in PD-associated scores determined according to the Hoehn & Yahr scale (H&Y), part III (motor phenotypes) of the Unified Parkinson's Disease Rating Scale (UPDRS), as well as the activities of daily living (ADL) scale (Table 1). Amongst others, the OGDHC activity as well as glucose consumption measured in hNPCs and DANs thereby significantly correlated with disease progression in patients (Fig. 9c,d). Furthermore, we combined our experimental data (assays, DEGs, DEPs, and significantly dysregulated metabolites) in a mixed effects model. This allowed to order the cell lines hierarchically according to the disease progression observed in the original sPD patients solely based on the experimental variables measured in the cellular model. Thereby the observed variability in the experimental data is matching the variability in the disease progression in the sPD patients. This is included in the manuscript as Fig. 9 and stated in the manuscript on page 12, line 24 to page 13, line 41:

Multiple factor analysis based on the characterization of hNPCs and DANs allows to stratify patients.

In a next step, the experimental data were used to stratify the patients according to the severity of their molecular alterations. To do so, a multiple factor analysis (MFA) was performed including detailed information gained from the scRNA-seq, proteome analysis, and nontargeted metabolomics analysis. In total 5 groups of variables were used: the top 100 DEGs, the top 100 DEPs, the significantly altered metabolites (45 metabolites), as well as the key metabolic and ciliary parameters (61 functional parameters), and as a supplementary group, the information about the disease state (Ctrl or sPD – Table 1).

The results show that dimensions 1 and 2 explain together about 42 % of the variability observed in patient derived cells (Fig. 9a). The first dimension represents mainly the functional parameters and altered metabolites, whereas DEGs and DEPS mainly contribute to dimension 2 (Fig. 9b). Of particular interest is the separation of Ctrl and sPD patients by dimension 1 (Fig. 9a). The larger the distance between a sPD patient and the average of Ctrl patients, the larger is the molecular deviations which might reflect the severity of the disease state.

To validate this hypothesis, the sPD patients that donated the skin fibroblasts used for hiPSCs generation were clinically examined at the timepoint of biopsy and were assessed again after a mean of 10.7 years (range 9-12 years). Both at baseline and at follow-up, all PD patients fulfilled the clinical diagnostic criteria for PD²⁸ and were defined as having sPD by the absence of known PD-causing familial mutations (PARK 1-18) and a negative family history of PD²⁹. Disease progression was assessed by monitoring changes in PD-associated scores determined according to the Hoehn & Yahr scale (H&Y), part III (motor symptoms) of the Unified Parkinson's Disease Rating Scale (UPDRS), as well as the activities of daily living (ADL) scale (Table 1).

Indeed, dimension 1 can also be used to separate subgroups of sPD patients with slow versus fast disease progression (see also Table 1). As patient R66 who was lost to follow-up clustered together with the fast progression group, it may be possible that R66 also exhibited a faster disease progression.

OGDHC deficiency and metabolic alterations correlate with disease progression in sPD patients

In a further translational effort we correlated our metabolic *in vitro* findings with disease progression of the sPD patients that donated the skin fibroblasts used for hiPSCs generation. A subset of variables is displayed in Fig. 9c or as scatter plots in Fig. 9d.

In line with our observations regarding the impact of OGDHC deficiency in sPD, a strong link existed between the rate of disease progression and OGDHC activity. In both, hNPCs and DANs, OGDHC activity correlated with disease progression (Δ H&Y (hNPCs – $R=-0.93$; $p=0.006$) (DANs – $R=-0.96$; $p=0.009$), Δ ADL (hNPCs – $R=0.95$; $p=0.004$) (DANs – $R=0.89$; $p=0.046$), and in parts Δ UPDRS III (hNPCs – $R=-0.89$; $p=0.017$)). Furthermore, in patients but not in Ctrl OGDHC activity correlated with cellular glucose consumption

Response to reviewer's comments

indicating a link between reduced glucose uptake and the reduced activity of the enzyme complex. Complementary, also the cellular glucose uptake was linked to disease progression (e.g. hNPCs – $R = 0.9$; $p = 0.015$).

In addition, OGDHC deficiency in sPD remained stable during the differentiation from hNPCs to DANs ($R = 0.82$; $p = 0.043$). This again validates the usability of hNPCs for disease modeling.

Interestingly, the ciliary capacity to transduce SHH signaling correlated with metabolic parameters. The capacity was assessed by monitoring the nuclear levels of GLI3-Full length (GLI3-FL) which is thought to function as weak transcriptional enhancer and the truncated GLI3-Repressor (GLI3-R) form. An increase of the GLI3-FL/GLI3-R (enhancer/repressor) ratio positively correlated with the protein abundance of the predicted SHH target OGDHL in hNPCs ($R = 0.98$; p -value= 0.00047). OGDHL protein levels thereby positively correlate with overall OGDHC activity ($R = 0.73$; p -value= 0.098). These findings together with the correlations of disease progression with OGDHC activity and OGDHL protein levels ($\Delta H\&Y$ ($R = -0.82$; p -value=0.046) further strengthen the observed impact of ciliary mediated SHH signaling on sPD onset and progression.

A similar separation of patients versus controls as by dimension 1 (Fig. 9a,b) could be achieved by only plotting the OGDHC activity measured in hNPCs against the respective complex I activity (Fig. 9d). Although activity levels of both enzymes did not significantly correlate in sPD ($R = -0.64$; p -value=0.12) or Ctrl ($R = -0.66$; p -value=0.22) samples, they clearly separated the individuals by the disease state. Furthermore, the separation was clearer in this case than if parameters would have been considered alone (Fig. 7g and ¹).

In sum, we present here a human cellular model system that combines – and allows to model – most of the known sporadic PD-associated metabolic alterations. Based on our findings, we propose a mechanism in which PC dysfunction underlies the onset of most of these metabolic alterations (Fig. 9e). Dysfunctional PC thereby affect SHH signal transduction which in turn results in altered gene expression patterns amongst others of the OGDHC. This creates a bottleneck within the citric acid cycle and thus reduces the flux through the main metabolic routes of glycolysis, citric acid cycle, and OXPHOS resulting in a sPD specific state of hypometabolism. Contrary, complex I deficiency seems to evolve independently of the PC mediated metabolic alterations. Thus, we present a model in which complex I deficiency and the SHH mediated hypometabolism develop as two independent hits that negatively impact cellular energy supply and could be used to predict disease progression in sPD patients.”

Table 1 | Disease progression in sPD patients within 10 years after biopsy. (left) Description of sPD patients at the timepoint of skin biopsy (Gender, and time in years between sPD diagnosis and tissue biopsy). **(right)** Long term history of sPD patients. Changes in the Hoehn&Yahr scale ($\Delta H\&Y$), in motor examinations (Part III) according to the Unified Parkinson Disease Rating Scale ($\Delta UPDRS$ III), medication requirement (ΔL -Dopa equivalent), and "activities of daily living" (ΔADL) monitored within 9-12 years after skin biopsy.

Time of biopsy			Clinical changes ~10 years after biopsy			
Patient ID	Gender	Years of illness	$\Delta H\&Y$	$\Delta UPDRS$ III [points]	ΔL -Dopa equivalent [mg]	ΔADL
J2C	m	3	3	24	400	-0.6
M89	m	3	3	53	1563	-0.6
C99	m	7	2.5	36	610	-0.5
R66	m	3	Follow-up not available			
AY6	m	4	1	-1	800	-0.2
PX7	m	1	1	0	1900	-0.1

Fig. 9 | OGDHC activity and metabolic alterations correlate with disease progression in sPD patients. (a) Multiple factor analysis using DEGs (bulk-like; top 100) (Fig. 3 and ¹), DEPs (top 100) (Fig. 4), significantly altered metabolites (45 metabolites) (Fig. 5), as well as the key experimental variables displayed in Fig. 1- 7 and ¹ describing alterations in sPD (61). The group points and lines represent the patient coordinates conditioned by the corresponding group variables. (b) Contributions of quantitative variables to the dimension 1 and 2 of the multiple factor analysis. (c) Correlogram visualizing the Pearson correlation coefficients of variables measured in sPD hNPCs and DANs, as well as parameters associated with disease progression in the sPD patients over a period of 9-12 years (Table 1). Disease progression markers are highlighted in green. Colors and dot sizes are proportional to the correlation coefficients. Negative correlations are colored in blue, positive correlations in red. (d) Visualization of interesting dependencies within measured variables and parameters of disease progression in sPD patients. Each point represents one patient. Linear regressions were calculated for each group and are displayed with a 95% confidence interval. The Pearson correlation coefficient (R) is displayed next to the corresponding p-value (p). Correlations with p-values < 0.1 were considered significant. (e) Graphical overview summarizing the main findings.

Response to reviewer's comments

- 2.2. In addition, the manuscript relies mainly on the use of NPCs rather than iDANs and although the authors present very limited data in iDANs, the metabolic status of the NPCs may be (and has been shown to be by others) fundamentally different from fully differentiated cells. Therefore, this calls into question the relevance of the sequence of events and metabolic rewiring shown in the NPC cell model, in addition to the connection to SHH signalling. Both of these pathways may well be fundamentally different in fully differentiated cells, indeed the link with complex I deficiency may be direct in iDANs and it is not effected in the NPCs as they fundamentally are more metabolically flexible and less OXPHOS dependent.

NPCs already underwent the metabolic switch from mainly glycolytic energy supply to OXPHOS. This assumption is based on our Seahorse XF experiments (**Fig. 1; Supplementary Fig. 2**) where the respiration behavior as well as the ECAR/OCR ratio of hNPCs is more similar to DANs than to hiPSC. This has also been previously described by ³⁰. But indeed, the NPC metabolism is in parts different from the DAN metabolism. Thus, to further highlight the relevance of our findings, we 1) repeated the key measurements in DANs, 2) compared our results with disease progression in the SPD patients that donated the skin fibroblasts used for hiPSC reprogramming (**see chapter 2.1**), and 3) compared our findings to already published clinical parameter assessed in SPD patients (**see chapter 2.1**).

- 1) To highlight the conservation of SPD specific deficiencies from the hNPC to DAN state, we validated our key findings in DANs and included these new data at the respective sites within the revised manuscript. We now provide a more detailed analysis of mitochondrial health in DANs. A quantification of mitochondrial mass and morphology is shown in **Fig. 2** and a quantification of the OXPHOS complexes is shown in **Fig. 3**. Next, the expression of glucose transporters as well as total glucose consumption was quantified in DANs to validate the PD specific state of hypometabolism. Indeed, also in DANs glucose uptake and metabolism was reduced in SPD to a similar extent as in hNPCs (**Fig. 1 and 6, Supplementary Fig. 2**). Additionally, we also assessed the overall OGDHC activity in DANs which was again reduced in SPD to a similar extent as in hNPCs (**Fig. 7**). To further characterize a possible conservation of the phenotypes from the hNPC to the DAN state, we correlated the hNPC OGDHC activity with the DAN OGDHC activity per cell line (**Fig. 9**). In SPD lines, OGDHC activity significantly correlated between hNPCs and DANs, meaning that the level of activity in the cell lines was similar in both states. However, this was not the case in Ctrl lines which exhibited a greater flexibility. This further highlights the relevance of OGDHC deficiency in SPD.

Altogether, our observations validate the usability of cellular models - also of hNPCs - to further investigate the alterations observed in SPD patients in a more detailed manner.

Response to reviewer's comments

2.3. Some of the data presented in this manuscript appears to have already been published in the same lines in the previous publication from this group, so there is a lot of repetition of results from their already published study; this is at times difficult to disentangle from the new data presented in this manuscript. The respiration data in Figure 1 with 25mM glucose or pyruvate, it would be crucial for the authors to explain how this data differs or is the same culture conditions as the respiration data presented in their previous paper Figure 1 (2022), likewise for the complex I defect. In addition, the authors previously showed that cyc treatment could restore the respiration of sPD NPCs in their previous publication of 2022, therefore, the data presented in Figure 6 in this manuscript appears in part to be a direct repeat of the same experiment in the same lines. This is also the case for the transcriptomic analysis the authors present and refer to. It is critical the authors clearly define the novel data in this manuscript.

We strongly disagree at this point, that we mainly present here data that has been already published by us previously¹. The only data that has been taken from the previous manuscript are the OCR data of cells consuming pyruvate. However, this has been and is clearly indicated in the manuscript in the respective legend of **Fig. 1**. We have deliberately opted to do so in order to allow a direct comparison between the OCR and ECAR data with glucose as energy substrate to the previously published data with pyruvate. This comparison is essential to demonstrate the metabolic alterations occurring in sPD lines which is the basis of our manuscript.

Data for the complex I deficiency is not shown in the present paper at all. However, our previous paper was cited in the manuscript body according to scientific standards:

“Indeed, a reduced complex I activity by approximately 30 % has been described for these sPD hNPCs¹. This corresponds well with literature describing complex I deficiency in different PD models as well as in post-mortem brain tissues of PD patients^{31,32}.”

(revised manuscript page 6, lines 7-8)

For a better visualization of a comparison, the respiration data following cycloamine treatment from the previous manuscript have been integrated. Again, it has been clearly stated in the body of the manuscript as well as the respective figure legend, that these data were “taken from”. In the present version of the manuscript due to rearrangements, this subfigure has been eliminated. In the manuscript body this result is still cited according to scientific standards:

“By repressing the overactive SHH signal transduction in sPD hNPCs to similar levels as in Ctrl, we previously showed that mitochondrial respiration analyzed by Seahorse XF was restored in sPD to levels similar as in Ctrl¹.”

(revised manuscript page 11, lines 34-36)

Regarding the transcriptome data, we present here a further specialized analysis focusing on solely metabolic parameters based on the previously published list of differentially expressed genes (Fig. 3a,b). Furthermore, in the respective figure legend, the manuscript body, and the methods section we cited the origin of the dataset e.g.:

“However, a specific analysis of recently published single-cell transcriptome data (bulk-like) of these hNPCs¹ for genes involved in bioenergetic processes identified an enrichment of dysregulated genes between sPD patients and Ctrl in pathways associated with the respiratory chain (Fig. 2b).”

(original manuscript page 4, lines 8-11)

Response to reviewer's comments

“To get insights into the molecular underpinnings for the observed respiratory deficiency in sPD hNPCs - which showed the most pronounced phenotype - we first performed a pathway enrichment analysis using our recently published single-cell transcriptome data (bulk-like) of these cells¹.”

(revised manuscript page 5, lines 30-33)

In order to do a multi-omics analysis, we also needed to integrate these published data with the proteome and metabolome data generated. Thus, we are following scientific standards in reusing published datasets. This specific transcriptome dataset is available not only to us, but also to others since its publication in 2022, to build on and generate or refute new hypotheses.

Response to reviewer's comments

2.4. In Figure 6 the imaging quantification for the H3K9me3 is stated 600 cells per cell clone, have the authors imaged 600 cells in total across 3 differentiations of NPCs or iDANs or passages of fibroblasts? In the plots are all 600 cell levels indicated as individual data points? This could massively skew the statistics if each individual cell is being counted as an individual data point rather than the mean levels of the cells from each repeat of the experiment.

Indeed, in the figure legends / the analysis scripts on GitHub we always stated how many individual cells per patient derived cell line were analyzed.

“(i) Immunostainings for aging-associated histone markers H3K9me3 and H3K27me3. Immunostainings are exemplarily shown for hNPCs of clone O3H-R1-003. Scale bar = 100 μ m. (j) Violin plots show the fluorescence intensity per condition. 600 cells per clone were analyzed from n = 5 Ctrl and 7 sPD clones, in triplicates. Boxplots display the median and range from the 25th to 75th percentile. Whiskers extend from the min to max value or to the most extreme data point which is no more than 1.5 times the interquartile range (for j). Each dot Represents one patient. n = 5 Ctrl and 7 sPD patient-derived cell Clones, in triplicates. p-values were determined by linear mixed effects model a, c, d, e, f, j; two-sided t-test g. *, p < 0.05; **, p < 0.01; ***, p < 0.001.”

(revised manuscript page 49, line 15 to page 50, line7)

As we had to perform these analyses on the cellular level instead of the population level, we decided not to use the mean values to not lose the additional information about variation in the cellular populations. Consequently, we used an appropriate statistical test that takes into account the dependencies within the dataset, meaning that it contains repeated measurements per sample. As stated in the corresponding figure legends and the Material and Methods section “Statistics”, we performed a nested ANOVA by nesting the repeated measurements per donor in a linear mixed effects model with the formula “Parameter measured per cell” ~ disease state + 1 | disease state:Patients.

“If not stated otherwise, R version 4.2.2 and RStudio 2022.12.0 Build 353 was used for further analysis. Violin plots were used to visualize the distributions of repeated measurements per individual cell line (e.g. if multiple cells per cell line were analyzed). Violin plots were generated using the functions *ggplot + geom_violin* of the R package *ggplot2* v3.4.2. For the statistical analysis, a linear mixed effects model (lm) was fit using the *lmer* function (R package “*lme4*” Version 1.1-34), where unique cells were included but nested within donors (formula: Parameter measured per cell ~ disease state + 1 | disease state:Patients; REML = FALSE). P values for lm were calculated using the *Anova* function (R package “*car*” Version 3.1-2). Boxplots summarizing measurements per individual cell line were generated using the functions *ggplot + geom_boxplot* of the R package *ggplot2* v3.4.2. These boxplots display the median and range from the 25th to 75th percentile. Whiskers extend to the most extreme data point which is no more than 1.5 times the interquartile range.”

(revised manuscript page 31, lines 25-36)

3. Reviewer #3 (Remarks to the Author):

Using a multilayered omics analysis of hiPSCs from sPD patients and healthy individuals as disease models, the authors have elucidated that the α -ketoglutarate dehydrogenase complex (OGDHC) in the citric acid cycle is the bottleneck in sPD metabolism. They previously reported that the dysfunction of PC signaling pathways and especially SHH signaling is a molecular pathway underlying PD development. In the current study, they demonstrated that the alterations in cellular metabolism and the OGDHC activity were restored by interfering the enhanced SHH signal transduction in PC function. The authors presented a human cellular system model of sPD that combines reduced glucose metabolism, reduced mitochondrial respiration, ATP production, OGDHC and complex I deficiency. The model would provide clues to the understanding of the molecular basis of this disease and expected to be used for development a neuroprotective therapy of sPD. Specific points are as follows.

Major points

- 3.1. Page 3 Result: Are the cell lines used in this study the same as those used in previous studies (Ref. 16)? If not, to confirm that the cells used in this study are appropriate models for sPD cells, the morphology and expression of NPC and DA markers should be shown, or references should be provided.

Indeed, the cell lines used in this study are exactly the same as described and characterized in Ref16 (1). Immunohistochemical staining of differentiation marks have been reported (shown here as **ReFig. 2**). To verify that DAn differentiation and maturation hasn't been affected in sPD clones, neurons/DAn have been counted and DAn morphology was assessed.

Additionally, we now also included a quantification of the hNPC marks SOX1, SOX2, and NESTIN as **Supplementary Fig. 1** to validate that the differentiation stage is comparable between Ctrl and sPD clones.

This is now also explicitly stated at the beginning of the results part on page 3, lines 11-27:

"In this study, we used hiPSCs derived from fibroblasts from 7 sPD patients and 5 age- and sex-matched Ctrl individuals, which were cultivated *in vitro* for approximately 60 passages. sPD patients were clinically examined and screened for the absence of known PD-causing familial mutations (PARK1-18)¹⁶. hiPSCs were repeatedly characterized for copy number variations and their differentiation potential¹. To establish a model system for sPD, hiPSCs were differentiated into human neuronal precursor cells (hNPCs) and further into dopaminergic neurons (DAn), which are vulnerable to degeneration in PD³³. Differentiation stages were confirmed using immunohistochemical staining for characteristic markers such as the precursor markers NESTIN, SOX2, and SOX1 (**Supplementary Fig. 1a**) or the DAn marker TUBB3, RBFOX3 (synonym: NeuN), and TH¹. Expression of these differentiation markers was not affected in sPD hNPCs (**Supplementary Fig. 2b**), nor was the abundance and morphology of DAn derived thereof¹. This indicates that the DAn differentiation process assessed at various stages was comparable between Ctrl and sPD cells."

Supplementary Fig. 1 | Characterization of hiPSC-derived hNPCs. (a) Immunostainings are exemplarily shown for clone O3H-R1-003 using antibodies against the NPC markers SOX1, SOX2, NESTIN. Scale bar=100 μ m. (b) Violin plots show the mean cytosolic fluorescence intensity of NESTIN, or (c) the mean nuclear fluorescence intensity of SOX1, or (d) SOX2. (left) Samples are pooled for a Ctrl-sPD comparison, or (right) values are plotted per patient-derived clone. The dashed line indicates the median fluorescence intensity level of Ctrl clones. Boxplots display the median and range from the 25th to 75th percentile. Whiskers extend to the most extreme data point which is no more than 1.5 times the interquartile range. 600 cells per clone were analyzed from n = 5 Ctrl and 7 sPD clones, in triplicates. p-values for the Ctrl-sPD comparison were determined by linear mixed effects model. *, p < 0.05; **, p < 0.01; ***, p < 0.001.

3.2. Page 4 Proteomics session and Page 10 Discussion: Proteomics data shows the decrease in OGDHL level and less change in OGDH in sPD hiPSC (Supplementary Data 3). Why didn't the authors mention the decrease in OGDHL in the proteomics session? When the metabolic categories (citric acid cycle) were enriched in the pathway enrichment analysis, the authors would have found a decrease in OGDHL prior to the metabolomic study.

We wrote the manuscript according to the chronological order of the experiments: Starting with broad omics gradually going deeper until identifying the metabolic bottleneck. Indeed, we could have focused on the OGDHL earlier. However, not only OGDHL was misexpressed on the transcriptome or proteome level, which made it somewhat difficult to identify OGDHL or the OGDHC as the bottleneck at that point. Several steps in the citric acid cycle would have been candidates for this. That's why we had to do additional experiments to narrow it down further. In addition, we did not want to speculate too far at that time and distract from other interesting results, such as the transcriptome-proteome differences or the expression of complex I subunits.

Response to reviewer's comments

3.3. Comparing the amounts of OGDHL and the proteins discussed here, such as complex1 assembly, PC signaling proteins, among the seven model cells may yield results that support their model.

The authors concluded that complex I deficiency and the SHH mediated hypometabolism develop as two independent hits that negatively impact cellular energy supply. Is this a conclusion reached by averaging the omics data from the seven sPD hiPSC, or is it a common dysregulation across all seven model cells? In other words, does the PD model presented by the authors always include both dysregulations, or can it be one or the other?

We thank the reviewer for these very thoughtful comments that, in our opinion, significantly improved the impact of our manuscript. As suggested, we correlated the OGDHL protein levels with the nuclear abundance of the transcription factors mediating SHH signaling as well as OGDHC activity for both groups, Ctrl and sPD (**Fig. 9c,d**). As OGDHL levels positively correlated with both the nuclear SHH transcription factor (GLI3) levels as well as OGDHC activity, this further strengthens our hypothesis of the impact of PC mediated SHH signal transduction on the onset of metabolic phenotypes associated with sPD.

Additionally, we also correlated the OGDHC activity with complex I activity to learn more about the relationship of these two important deficiencies associated with PD (**Fig. 9c,d**). Activity levels of both enzymes did not significantly correlate in sPD ($R=-0.64$; $p\text{-value}=0.12$) or Ctrl ($R=-0.66$; $p\text{-value}=0.22$) samples. However, blotting the OGDHC activity against the complex I activity clearly separated the individuals by the disease state. Furthermore, the separation was clearer in this case than if parameters would have been considered alone (**Fig. 7g and 1**). Although the two parameters did not significantly correlate, it seemed that the higher the Complex I activity was, the lower the OGDHC activity and vice versa.

In a translational effort, we also correlated the core results based on the analysis of our cellular model system (hiPSC and hNPCs, DANs differentiated thereof) with clinical patient data on disease progression. The sPD patients that donated the original skin fibroblasts used for hiPSC generation were characterized at the timepoint of biopsy and were assessed again after 9-12 years. Disease progression was assessed by monitoring changes in PD-associated scores determined according to the Hoehn & Yahr scale (H&Y), part III (motor phenotypes) of the Unified Parkinson's Disease Rating Scale (UPDRS), as well as the activities of daily living (ADL) scale (**Table 1**). Interestingly, OGDHC activity measured in hNPCs and DANs thereby significantly correlated with disease progression in patients (**Fig. 9c,d**).

This is included in the manuscript as **Fig. 9** and stated in the manuscript on page 13, lines 11-41:

“OGDHC deficiency and metabolic alterations correlate with disease progression in sPD patients

In a further translational effort we correlated our metabolic *in vitro* findings with disease progression of the sPD patients that donated the skin fibroblasts used for hiPSCs generation. A subset of variables is displayed in **Fig. 9c** or as scatter plots in **Fig. 9d**.

In line with our observations regarding the impact of OGDHC deficiency in sPD, a strong link existed between the rate of disease progression and OGDHC activity. In both, hNPCs and DANs, OGDHC activity correlated with disease progression ($\Delta\text{H\&Y}$ (hNPCs – $R=-0.93$; $p=0.006$) (DANs – $R=-0.96$; $p=0.009$), ΔADL (hNPCs – $R=0.95$; $p=0.004$) (DANs – $R=0.89$; $p=0.046$), and in parts $\Delta\text{UPDRS III}$ (hNPCs – $R=-0.89$; $p=0.017$)). Furthermore, in patients but not in Ctrl OGDHC activity correlated with cellular glucose consumption indicating a link between reduced glucose uptake and the reduced activity of the enzyme complex. Complementary, also the cellular glucose uptake was linked to disease progression (e.g. hNPCs – $R = 0.9$; $p = 0.015$).

Response to reviewer's comments

In addition, OGDHC deficiency in sPD remained stable during the differentiation from hNPCs to DANs ($R = 0.82$; $p = 0.043$). This again validates the usability of hNPCs for disease modeling.

Interestingly, the ciliary capacity to transduce SHH signaling correlated with metabolic parameters. The capacity was assessed by monitoring the nuclear levels of GLI3-Full length (GLI3-FL) which is thought to function as weak transcriptional enhancer and the truncated GLI3-Repressor (GLI3-R) form. An increase of the GLI3-FL/GLI3-R (enhancer/repressor) ratio positively correlated with the protein abundance of the predicted SHH target OGDHL in hNPCs ($R = 0.98$; $p\text{-value} = 0.00047$). OGDHL protein levels thereby positively correlate with overall OGDHC activity ($R = 0.73$; $p\text{-value} = 0.098$). These findings together with the correlations of disease progression with OGDHC activity and OGDHL protein levels ($\Delta H\&Y$ ($R = -0.82$; $p\text{-value} = 0.046$)) further strengthen the observed impact of ciliary mediated SHH signaling on sPD onset and progression.

A similar separation of patients versus controls as by dimension 1 (Fig. 9a,b) could be achieved by only plotting the OGDHC activity measured in hNPCs against the respective complex I activity (Fig. 9d). Although activity levels of both enzymes did not significantly correlate in sPD ($R = -0.64$; $p\text{-value} = 0.12$) or Ctrl ($R = -0.66$; $p\text{-value} = 0.22$) samples, they clearly separated the individuals by the disease state. Furthermore, the separation was clearer in this case than if parameters would have been considered alone (Fig. 7g and ¹).

Table 1 | Disease progression in sPD patients within 10 years after biopsy. (left) Description of sPD patients at the timepoint of skin biopsy (Gender, and time in years between sPD diagnosis and tissue biopsy). **(right)** Long term history of sPD patients. Changes in the Hoehn&Yahr scale (H&Y), in motor examinations (Part III) according to the Unified Parkinson Disease Rating Scale (UPDRS III), medication requirement (L-Dopa equivalent), and "activities of daily living" (ADL) monitored within 9-12 years after skin biopsy.

Time of biopsy			Clinical changes ~10 years after biopsy			
Patient ID	Gender	Years of illness	Δ H&Y	Δ UPDRS III [points]	Δ L-Dopa equivalent [mg]	Δ ADL
J2C	m	3	3	24	400	-0.6
M89	m	3	3	53	1563	-0.6
C99	m	7	2.5	36	610	-0.5
R66	m	3	Follow-up not available			
AY6	m	4	1	-1	800	-0.2
PX7	m	1	1	0	1900	-0.1
88H	f	6	2	7	640	-0.3

Revision of manuscript NCOMMS-23-06769-T - Molecular mechanisms underlying a reversible state of hypometabolism in sporadic Parkinson's disease
Response to reviewer's comments

Fig. 9 | OGDHC activity and metabolic alterations correlate with disease progression in sPD patients. (a) Multiple factor analysis using DEGs (bulk-like; top 100) (Fig. 3 and ¹), DEPs (top 100) (Fig. 4), significantly altered metabolites (45 metabolites) (Fig. 5), as well as the key experimental variables displayed in Fig. 1- 7 and ¹ describing alterations in sPD (61). The group points and lines represent the patient coordinates conditioned by the corresponding group variables. (b) Contributions of quantitative variables to the dimension 1 and 2 of the multiple factor analysis. (c) Correlogram visualizing the Pearson correlation coefficients of variables measured in sPD hNPCs and DANs, as well as parameters associated with disease progression in the sPD patients over a period of 9-12 years (Table 1). Disease progression markers are highlighted in green. Colors and dot sizes are proportional to the correlation coefficients. Negative correlations are colored in blue, positive correlations in red. (d) Visualization of interesting dependencies within measured variables and parameters of disease progression in sPD patients. Each point represents one patient. Linear regressions were calculated for each group and are displayed with a 95% confidence interval. The Pearson correlation coefficient (R) is displayed next to the corresponding p-value (p). Correlations with p-values < 0.1 were considered significant. (e) Graphical overview summarizing the main findings.

Minor point

3.4. Supplementary Data 4: Data of mKEGG is missing.

The table just summarizes significantly enriched terms. Unfortunately for mKEGG, not a single term was significantly enriched. These data are not missing, there was just nothing to report.

3.5. Page 5 line 1: I cannot find the term of primary cilia (PC) in Fig. 3d, Supplementary Fig. 2d-c, and Supplementary Data 4). Is PC referring intraflagellar transport here?

Yes indeed, intraflagellar transport refers to the microtubule dependent transport processes within PC. This has been further clarified in the corresponding figure (now **Fig. 4d**).

Revision of manuscript NCOMMS-23-06769-T - Molecular mechanisms underlying a reversible state of hypometabolism in sporadic Parkinson's disease

Response to reviewer's comments

Fig. 4 | Proteome analysis points towards primary cilia and citric acid cycle defects in sPD. (a) Principal component analysis visualizes the variability within technical replicates and conditions. **(b)** Proteome analysis identified 1,667 (out of 7,943 proteins) which were dysregulated in sPD hNPCs. Dotted line indicates the significance threshold ($q < 0.05$). **(c)** Heatmap showing \log_2 transformed fold changes (FC) with columns scaled by z-score for differentially expressed proteins (DEPs). **(d)** Enriched Reactome terms based on all DEPs. See also **Supplementary Fig. 3**. **(e)** Overlap of genome-wide predicted GLI3 target genes using the MatInspector (Genomatix) with DEPs. **(f)** Correlation between DEGs and DEPs. FC for DEGs are plotted on the x-axis, FC of DEPs on the y-axis. Linear regressions between upregulated DEGs and DEPs (red), downregulated DEGs and DEPs (blue), upregulated DEGs and downregulated DEPs (orange), downregulated DEGs and upregulated DEPs (light blue) are displayed together with their respective Pearson correlation coefficients and p-values. **(g)** Enriched Reactome terms for DEG-DEP pairs with a negative correlation (downregulated DEG and upregulated DEP; upregulated DEG and downregulated DEP) between transcriptome and proteome level as well as a **(h)** positive correlation (downregulated DEG and DEP; upregulated DEG and DEP). $n = 5$ Ctrl and 7 sPD patient-derived cell clones, in Triplicates. p-values were determined by one-sided hypergeometric tests **d, g, h**; one-sided Fisher's Exact Test **e**; two-sided t -test **f**. p-values corrected for multiplicity are represented by q-values. *, $p < 0.05$; **, $p < 0.01$; ***, $p < 0.001$. See also **Supplementary Fig. 3; Supplementary Data 3, 4, and 5**.

3.6. Page 5 line 27; TIMMDC1 is not shown in Fig. 3.

This has been corrected. TIMMDC1 is now highlighted in the corresponding figure (now **Fig. 4b** – see **chapter 3.5**).

3.7. Page 10 line 14 and Fig 6: Fig 6i is a mistake for Fig. 6l.

Yes, meant was Fig. 6l. This has been corrected.

3.8. Fig. 6: A brief explanation regarding Fig. 6l would be helpful. I think it is not easy to understand this figure in the explanation on page 10 lines 12-14.

The corresponding paragraph has been updated to provide a more detailed explanation of the summary sketch (now **Fig. 9e**). This is now stated on page 14, lines 1-11:

“In sum, we present here a human cellular model system that combines – and allows to model – most of the known sporadic PD-associated metabolic alterations. Based on our findings, we propose a mechanism in which PC dysfunction underlies the onset of most of these metabolic alterations (**Fig. 9e**). Dysfunctional PC thereby affect SHH signal transduction which in turn results in altered gene expression patterns amongst others of the OGDHC. This creates a bottleneck within the citric acid cycle and thus reduces the flux through the main metabolic routes of glycolysis, citric acid cycle, and OXPHOS resulting in a sPD specific state of hypometabolism. Contrary, complex I deficiency seems to evolve independently of the PC mediated metabolic alterations. Thus, we present a model in which complex I deficiency and the SHH mediated hypometabolism develop as two independent hits that negatively impact cellular energy supply and could be used to predict disease progression in sPD patients.”

3.9. Page 19, LC/MS: Just to confirm, was the 50 cm column used without a trap column?

Yes, no trap column was used. This has been clarified also in the corresponding methods section on page 24, lines 22-23:

“Peptides were loaded on a 50 cm reversed phase column (75 µm inner diameter, packed in house with ReproSil-Pur C18-AQ 1.9 µm resin). **No trap column was used.**”

3.10. Page 19, DIA-MS: If m/z range was fractionated, information on m/z of separation, number of windows, whether overlapping windows were acquired should be provided.

Information regarding m/z separation and number of windows are now provided as **Supplementary Data 10**. This has been also clarified on page 24, lines 32-37:

“The MS data was acquired using a data independent acquisition (DIA) mode with a full scan range of 300–1650 m/z at 120,000 resolution, automatic gain control (AGC) of 3e6 and a maximum injection time of 60 ms. The stepped higher-energy collision dissociation (HCD) was set to 25.5, 27.30. Each full scan was followed by 33 DIA scans which were performed at a 30,000 resolution, an AGC of 1e6 and the maximum injection time set to “auto”. **Information regarding m/z separation and number of windows are provided as Supplementary Data 10.**”

3.11. Page19 line 39: Database download date is required.

The UniProt database was downloaded July 2019. This has been also clarified on page 24, line 39-41:

"DIA raw files were analyzed using directDIA in Spectronaut version 15 (Biognosys). The search was done against UniProt human proteome of canonical and isoform sequences (downloaded July 2019) with 20,383 entries for final protein identification and quantification."

3.12. Please show the full name of DEG, FCCP, ROS, and SHH.

The full names are now included.

- For DEGs on page 6, lines 29-30:

"Similarly to the transcriptome ¹, levels of most DEPs were only slightly altered, however, up- and downregulation of DEPs was more balanced (Fig. 4b) compared to the transcriptome level where ~88% of differentially expressed genes (DEGs) were downregulated."

- For FCCP in the figure legend of Fig. 1.

Fig. 1 | Respiratory characterization of Ctrl and sPD cell lines using different energy substrates. (a) Mitochondrial stress test performed in hiPSCs, **(b)** thereof differentiated hNPCs, and **(c)** DANs using a Seahorse XFe96 Extracellular Flux Analyzer. Cells were measured in Seahorse XF assay medium supplemented with 25 mM glucose or 5 mM pyruvate (as shown by ¹). Injected were (A) Oligomycin (1 µg/ml), (B) carbonyl cyanide p-trifluoro-methoxyphenyl hydrazone (FCCP; 0.5 µM), (C) Rotenone (5 µM)/Antimycin A (2 µM), and (D) 2-Deoxyglucose (2-DG; 100 mM). Measurement progression is shown with means ± standard error of the mean (SEM). Boxplots display the median and range from the 25th to 75th percentile. Whiskers extend from the min to max value. Each dot represents one patient. n = 5 Ctrl and 7 sPD patient-derived cell clones, in triplicates. p-values were determined by one-way ANOVA with Sidak's Post-hoc test. *, p < 0.05; **, p < 0.01; ***, p < 0.001."

- For ROS on page 15, lines 6-7:

"As a consequence of altered OGDHC function in sPD, also several OGDHC-dependent cellular processes were affected as well such as α-ketoglutarate metabolism or cellular reactive oxygen species (ROS) signaling."

- For SHH on page 3, line 4.

"Thereby, alterations in sPD metabolism were introduced by enhanced primary cilia (PC)-mediated sonic hedgehog (SHH) signal transduction, as alterations in cellular metabolism could be rescued in our cellular model of sPD by interfering with SHH signaling."

References

1. Schmidt, S. *et al.* Primary cilia and SHH signaling impairments in human and mouse models of Parkinson's disease. *Nature communications* **13**, 4819; 10.1038/s41467-022-32229-9 (2022).
2. Schmidt, C. A., Fisher-Wellman, K. H. & Neuffer, P. D. From OCR and ECAR to energy: Perspectives on the design and interpretation of bioenergetics studies. *The Journal of biological chemistry* **297**, 101140; 10.1016/j.jbc.2021.101140 (2021).
3. Little, D. *et al.* A single cell high content assay detects mitochondrial dysfunction in iPSC-derived neurons with mutations in SNCA. *Scientific reports* **8**, 9033; 10.1038/s41598-018-27058-0 (2018).
4. Norris, K. L. *et al.* Convergence of Parkin, PINK1, and α -Synuclein on Stress-induced Mitochondrial Morphological Remodeling. *The Journal of biological chemistry* **290**, 13862–13874; 10.1074/jbc.M114.634063 (2015).
5. Mortiboys, H., Johansen, K. K., Aasly, J. O. & Bandmann, O. Mitochondrial impairment in patients with Parkinson disease with the G2019S mutation in LRRK2. *Neurology* **75**, 2017–2020; 10.1212/WNL.0b013e3181ff9685 (2010).
6. Mortiboys, H. *et al.* Mitochondrial function and morphology are impaired in parkin-mutant fibroblasts. *Annals of neurology* **64**, 555–565; 10.1002/ana.21492 (2008).
7. van der Blik, A. M., Shen, Q. & Kawajiri, S. Mechanisms of mitochondrial fission and fusion. *Cold Spring Harbor perspectives in biology* **5**; 10.1101/cshperspect.a011072 (2013).
8. Knott, A. B., Perkins, G., Schwarzenbacher, R. & Bossy-Wetzell, E. Mitochondrial fragmentation in neurodegeneration. *Nature reviews. Neuroscience* **9**, 505–518; 10.1038/nrn2417 (2008).
9. Pajak, B. *et al.* 2-Deoxy-d-Glucose and Its Analogs: From Diagnostic to Therapeutic Agents. *International Journal of Molecular Sciences* **21**; 10.3390/ijms21010234 (2019).
10. Divakaruni, A. S. *et al.* Inhibition of the mitochondrial pyruvate carrier protects from excitotoxic neuronal death. *The Journal of cell biology* **216**, 1091–1105; 10.1083/jcb.201612067 (2017).
11. Tran, J., Anastacio, H. & Bardy, C. Genetic predispositions of Parkinson's disease revealed in patient-derived brain cells. *NPJ Parkinson's disease* **6**, 8; 10.1038/s41531-020-0110-8 (2020).
12. Rosh, I. *et al.* Synaptic dysfunction and dysregulation of extracellular matrix-related genes in dopaminergic neurons derived from Parkinson's disease sporadic patients and with GBA1 mutations (2023).
13. Chlebanowska, P. *et al.* Mitochondrial fitness influences neuronal excitability of dopaminergic neurons from patients with idiopathic form of Parkinson's disease (2023).
14. Corenblum, M. J. *et al.* Parallel Neurodegenerative Phenotypes in Sporadic Parkinson's Disease Fibroblasts and Midbrain Dopamine Neurons. *bioRxiv*; 10.1101/2023.02.10.527867 (2023).
15. Stern, S. *et al.* Reduced synaptic activity and dysregulated extracellular matrix pathways in midbrain neurons from Parkinson's disease patients. *NPJ Parkinson's disease* **8**, 103; 10.1038/s41531-022-00366-z (2022).
16. Vlasov, I. N. *et al.* Transcriptome Analysis of Induced Pluripotent Stem Cells and Neuronal Progenitor Cells, Derived from Discordant Monozygotic Twins with Parkinson's Disease. *Cells* **10**; 10.3390/cells10123478 (2021).

Response to reviewer's comments

17. Badanjak, K. *et al.* iPSC-Derived Microglia as a Model to Study Inflammation in Idiopathic Parkinson's Disease. *Frontiers in Cell and Developmental Biology* **9**, 740758; 10.3389/fcell.2021.740758 (2021).
18. Schulze, M. *et al.* Sporadic Parkinson's disease derived neuronal cells show disease-specific mRNA and small RNA signatures with abundant deregulation of piRNAs. *Acta neuropathologica communications* **6**, 58; 10.1186/s40478-018-0561-x (2018).
19. Burbulla, L. F. *et al.* Dopamine oxidation mediates mitochondrial and lysosomal dysfunction in Parkinson's disease. *Science (New York, N.Y.)* **357**, 1255–1261; 10.1126/science.aam9080 (2017).
20. Fernández-Santiago, R. *et al.* Aberrant epigenome in iPSC-derived dopaminergic neurons from Parkinson's disease patients. *EMBO molecular medicine* **7**, 1529–1546; 10.15252/emmm.201505439 (2015).
21. Steidel, K. *et al.* Longitudinal trimodal imaging of midbrain-associated network degeneration in Parkinson's disease. *NPJ Parkinson's disease* **8**, 79; 10.1038/s41531-022-00341-8 (2022).
22. Yoon, E. J. *et al.* Brain Metabolism Related to Mild Cognitive Impairment and Phenoconversion in Patients With Isolated REM Sleep Behavior Disorder. *Neurology* **98**, e2413-e2424; 10.1212/WNL.0000000000200326 (2022).
23. Albrecht, F., Ballarini, T., Neumann, J. & Schroeter, M. L. FDG-PET hypometabolism is more sensitive than MRI atrophy in Parkinson's disease: A whole-brain multimodal imaging meta-analysis. *NeuroImage. Clinical* **21**, 101594; 10.1016/j.nicl.2018.11.004 (2019).
24. Hansen, G. E. & Gibson, G. E. The α -Ketoglutarate Dehydrogenase Complex as a Hub of Plasticity in Neurodegeneration and Regeneration. *IJMS* **23**; 10.3390/ijms232012403 (2022).
25. Gibson, G. *et al.* Deficits in a tricarboxylic acid cycle enzyme in brains from patients with Parkinson's disease. *Neurochemistry International* **43**, 129–135; 10.1016/S0197-0186(02)00225-5 (2003).
26. Berndt, N., Bulik, S. & Holzhütter, H.-G. Kinetic Modeling of the Mitochondrial Energy Metabolism of Neuronal Cells: The Impact of Reduced α -Ketoglutarate Dehydrogenase Activities on ATP Production and Generation of Reactive Oxygen Species. *International journal of cell biology* **2012**, 757594; 10.1155/2012/757594 (2012).
27. Mizuno, Y. *et al.* Role of mitochondria in the etiology and pathogenesis of Parkinson's disease. *Biochimica et Biophysica Acta (BBA) - Molecular Basis of Disease* **1271**, 265–274; 10.1016/0925-4439(95)00038-6 (1995).
28. Postuma, R. B. *et al.* MDS clinical diagnostic criteria for Parkinson's disease. *Movement disorders : official journal of the Movement Disorder Society* **30**, 1591–1601; 10.1002/mds.26424 (2015).
29. Popp, B. *et al.* Need for high-resolution Genetic Analysis in iPSC: Results and Lessons from the ForIPS Consortium. *Scientific reports* **8**, 17201; 10.1038/s41598-018-35506-0 (2018).
30. Lorenz, C. *et al.* Human iPSC-Derived Neural Progenitors Are an Effective Drug Discovery Model for Neurological mtDNA Disorders. *Cell Stem Cell* **20**, 659-674.e9; 10.1016/j.stem.2016.12.013 (2017).
31. Schapira, A. H. *et al.* Anatomic and disease specificity of NADH CoQ1 reductase (complex I) deficiency in Parkinson's disease. *Journal of neurochemistry* **55**, 2142–2145; 10.1111/j.1471-4159.1990.tb05809.x (1990).

Revision of manuscript NCOMMS-23-06769-T - Molecular mechanisms underlying a reversible state of hypometabolism in sporadic Parkinson's disease

Response to reviewer's comments

32. Vos, M. Mitochondrial Complex I deficiency: guilty in Parkinson's disease. *Signal transduction and targeted therapy* **7**, 136; 10.1038/s41392-022-00983-3 (2022).
33. Fahn, S. Description of Parkinson's disease as a clinical syndrome. *Annals of the New York Academy of Sciences* **991**, 1–14; 10.1111/j.1749-6632.2003.tb07458.x (2003).

REVIEWERS' COMMENTS

Reviewer #1 (Remarks to the Author):

In this revised manuscript authors satisfactorily addressed a series of points raised by different reviewers. The authors added a considerable amount of new data that significantly improved their work. I recommend the paper for publication.

Reviewer #2 (Remarks to the Author):

Thank you to the authors to considering and addressing the reviewers comments. The authors have addressed most comments made and in my opinion made a good case for not including some of the additional data requested.

The authors have clearly added much more information about the phenotype to the manuscript, clearly delineated what data has been previously reported and what is novel in this manuscript and the additional translational correlations strengthen the impact of the manuscript.

Dear Reviewers,

We thank you all for your valuable comments on our manuscript which helped to improve the manuscript substantially!

A point-by-point response to each reviewers' comment is indicated below. For better readability of this part of our response, we highlighted each reviewers' comments in a different color, whereas our responses are always kept in black.

Thank you once again for your valuable input and specifically the time taken to help us to improve our manuscript.

Best regards,

Wolfgang Wurst

Revision of manuscript NCOMMS-23-06769A- A reversible state of hypometabolism in a human cellular model of sporadic Parkinson's disease

Response to reviewer's comments

REVIEWER COMMENTS

Reviewer #1 (Remarks to the Author):

In this revised manuscript authors satisfactorily addressed a series of points raised by different reviewers. The authors added a considerable amount of new data that significantly improved their work. I recommend the paper for publication.

We thank the reviewer for his/her comments and appreciate the positive evaluation of our manuscript.

Reviewer #2 (Remarks to the Author):

Thank you to the authors to considering and addressing the reviewers comments. The authors have addressed most comments made and in my opinion made a good case for not including some of the additional data requested.

The authors have clearly added much more information about the phenotype to the manuscript, clearly delineated what data has been previously reported and what is novel in this manuscript and the additional translational correlations strengthen the impact of the manuscript.

We thank the reviewer for his/her comments and appreciate the positive evaluation of our manuscript.